# Tackling Fake Forgetting through Uncertainty Quantification

**Yingdan Shi** [1]  **Sijia Liu** [2]  **Kaize Ding** [3]  **Ren Wang** [1]

## Abstract

Machine unlearning seeks to remove the influence of specified data from a trained model. While the unlearning accuracy is a widely used metric for assessing unlearning performance, it falls short in assessing the reliability of forgetting. In this paper, we find that the forget data points misclassified by unlearning accuracy still have their ground truth labels included in the conformal prediction set from the uncertainty quantification perspective, leading to a phenomenon we term fake forgetting. To address this issue, we propose a novel metric CR, inspired by conformal prediction, that offers a more reliable assessment of forgetting quality. Building on these insights, we further propose an unlearning framework CPU that incorporates conformal prediction into the Carlini & Wagner adversarial attack loss, enabling the ground truth label to be effectively removed from the conformal prediction set. Through extensive experiments on image classification tasks, we demonstrate both the effectiveness of our proposed metric and the superior forgetting quality achieved by our framework. Code is available at https://github.com/TIML-Group/Conformal-Prediction-Unlearning.

## 1. Introduction

Machine unlearning has become essential for data privacy, particularly under legal regulations such as the General Data Protection Regulation (GDPR) (Bourtoule et al., 2021). These regulations emphasize the right for individuals to have their data removed or forgotten, creating a demand for machine unlearning methods that can enable machine learning models to behave as if specific forget data were never used in the training stage. Beyond privacy, unlearning also serves as a tool for mitigating harmful biases and stereotypes in

| Class Name | Forget Data | Original Model | Finetune Method | Our Method |
|---|---|---|---|---|
| **Wok** | | | | |
| **Swimming Trunks** | | | | |
| **Classification** | – | ✓ | ✗ | ✗ |
| **In Set** | – | ✓ | ✓ | ✗ |

*Figure 1.* Grad-CAM maps of one original model and two corresponding unlearning models in Tiny ImageNet with ViT. The **Classification** row indicates whether the model correctly predicts the image's true label, while the **In Set** row represents whether the true label is included in the prediction set. Although the Finetune method can misclassify the forgetting data, Grad-CAM can still highlight key features of the object since the true label is included in the prediction set. In contrast, when our unlearning framework CPU removes the true label from the set, activation regions shift significantly away from the object's key features. This confirms that the forgetting quality improves if the true label can be excluded from the prediction set.

models. The existing post hoc machine unlearning methods can be categorized into training-based (Graves et al., 2021; Tarun et al., 2023; Thudi et al., 2022; Warnecke et al., 2021; Shi & Wang, 2025b) and training-free (Foster et al., 2024; Golatkar et al., 2021; 2020; Guo et al., 2019; Nguyen et al., 2020; Sekhari et al., 2021; Sun et al., 2024) approaches, depending on whether they require any model training steps during the unlearning process (Foster et al., 2024).

To measure the forgetting quality and predictive performance of an unlearning model, several unlearning metrics have been proposed (Hayes et al., 2025; Cao & Yang, 2015; Chen et al., 2021; Kashef, 2021; Shokri et al., 2017). However, existing unlearning metric unlearning accuracy (**UA**), falls short in fully evaluating forgetting reliability. UA primarily focuses on whether models can predict forget data accurately **without sufficiently considering uncertainty and confidence level**. In a nutshell, misclassifying the forget data does not mean that the model has completely forgotten it.

To verify this view, conformal prediction (Lei & Wasserman, 2014; Papadopoulos et al., 2002) as an uncertainty quantification technique, is applied in our work to recover the misclassified data in UA. Through extensive experiments,

[1]Illinois Institute of Technology, Chicago, IL, USA [2]Michigan State University, East Lansing, MI, USA [3]Northwestern University, Evanston, IL, USA. Correspondence to: Ren Wang <rwang74@iit.edu>.

*Proceedings of the 43rd International Conference on Machine Learning*, Seoul, South Korea. PMLR 306, 2026. Copyright 2026 by the author(s).

we find that although the model misclassifies part of the forget data from the UA perspectives, **over** $50\%$ **of these misclassified data instances still appear in the conformal prediction set and can be easily recovered, which exposes a fake forgetting issue**. As shown in Figure 1, the important features of prediction visualize this fake forgetting issue by using Grad-CAM (Selvaraju et al., 2017). Despite the Finetune method misclassifying the forget data, the Grad-CAM maps still focus heavily on the important features of the object itself since the true label is included in the prediction set. In contrast, when our unlearning framework removes the true label from the set, activation regions shift significantly away from the object's key features. This confirms that forgetting quality improves if the true label can be excluded from the prediction set.

Based on the above insights, we design a novel metric **C**onformal **R**atio (**CR**) that more effectively captures the uncertainty and robustness of unlearning performance inspired by conformal prediction to tackle the fake forgetting issue. Additionally, motivated by conformal prediction insights about fake forgetting and Carlini & Wagner (C&W) attack loss (Carlini & Wagner, 2017), we propose a general **C**onformal **P**rediction **U**nlearning framework (**CPU**), which can improve existing training-based unlearning methods and promote reliable forgetting. Grad-CAM maps of our method in Figure 1 reveal that **once the true label no longer falls within the conformal prediction set, the activation regions shift significantly**. To sum up, our contributions are as follows:

- Our analysis reveals that conformal prediction can recover a substantial portion of data previously classified as forgotten by the UA metric. This fake forgetting issue underscores critical limitations in the UA metric.

- We design a novel metric CR, motivated by conformal prediction, to address the limitations of UA.

- We propose an unlearning framework CPU motivated by conformal prediction and C&W loss, which can enhance the unlearning quality of training-based unlearning methods over both existing and our metrics.

- Extensive experiments demonstrate the effectiveness of our proposed metric and validate the performance of our unlearning framework under both our metric and existing evaluation metrics.

## 2. Related Work

Machine unlearning has emerged as a vital research topic due to several privacy, regulatory, and ethical concerns associated with machine learning models. It refers to the process of selectively removing specific data points from a trained machine learning model. Generally, post-hoc machine unlearning can be divided into training-based (Graves et al., 2021; Tarun et al., 2023; Thudi et al., 2022; Warnecke et al., 2021; Shi & Wang, 2025a) and training-free approaches (Foster et al., 2024; Golatkar et al., 2021; 2020; Guo et al., 2019; Nguyen et al., 2020; Sekhari et al., 2021).

To evaluate unlearning methods, UA (Brophy & Lowd, 2021; Foster et al., 2024) is one of the most commonly used metric for assessing unlearning performance. In addition, several works, such as MIA (Chen et al., 2021; Shokri et al., 2017) and U-MIA (Hayes et al., 2025), have been proposed as auxiliary evaluation metrics. Our work focuses on addressing UA's limitation: it does not account for the confidence of model predictions. To overcome this, we adopt conformal prediction (Angelopoulos & Bates, 2021), a model-agnostic framework that transforms any black-box model's outputs into well-calibrated prediction sets. Its versatility has led to numerous specialized conformal prediction methods tailored to specific prediction problems (Lei et al., 2018; Lei & Wasserman, 2014; Papadopoulos et al., 2002; Romano et al., 2020a).

Becker et al. (Becker & Liebig, 2022) is the most closely related work to ours, as it explicitly studies uncertainty in the context of machine unlearning. However, it studies machine unlearning from the perspective of parameter-level uncertainty, quantifying the sensitivity of model parameters to specific training data using the Fisher Information Matrix. While this approach provides valuable insights into how individual samples influence model parameters, it does not directly address the confidence of model predictions after unlearning, which is crucial for evaluating whether information has been truly forgotten at the output level.

A more recent concurrent work (Spartalis et al., 2025) employs a knowledge distillation framework with dynamic logits tempering and Gumbel noise to smooth prediction probabilities. Their uncertainty adjustments mainly serve as a training heuristic to improve the distillation process. Our approach, by contrast, leverages conformal prediction not only to provide a principled and well-calibrated evaluation metric, but also to enhance the unlearning process itself through a simple, model-agnostic loss term. This distinction highlights both the evaluation reliability and the general applicability of our framework.

## 3. Enhancing UA Metric Based on Conformal Prediction

### 3.1. Preliminaries and Notations

**Machine Unlearning.** In our work, we focus on the image classification task, which is widely used in prior literature (Shen et al., 2024; Zhao et al., 2024). Two forgetting scenarios are mainly considered in this work: (i) *random data*

*forgetting* focuses on randomly forgetting specific data instances within the training data, and (ii) *class-wise forgetting* aims to remove all data information associated with an entire class. We also report the results of the worst-case and subclass-wise forgetting scenarios in Tables 11 and 12 in Appendix G. Let $\mathcal{D}_{train}$ denote the original training data used to obtain an original model $\boldsymbol{\theta}_o$. We split the whole training data $\mathcal{D}_{train}$ into two subsets, forget data $\mathcal{D}_f$ and retain data $\mathcal{D}_r = \mathcal{D}_{train} \setminus \mathcal{D}_f$. Let $\mathcal{D}_{test}$ represent test data. $\theta_u$ denotes the model after the unlearning process.

**Conformal Prediction.** Conformal prediction (CP) is proposed to quantify uncertainty, providing prediction sets that contain the ground truth label with a theoretically guaranteed probability (Angelopoulos & Bates, 2021). Among the various types of conformal prediction, this work mainly focuses on split conformal prediction (SCP)[1] since it is the most straightforward and easy-to-implement approach. We also report results of other conformal prediction techniques in Appendix F.1. To construct a conformal prediction set, SCP involves four steps on the unlearning model:

1. *Calibration Set*. SCP first chooses unseen data as a calibration set, which must be held out from both the training and test sets to ensure independence.

2. *Non-conformity Score*. In our work, we follow the conventional choice and set the non-conformity score as

$$S(\boldsymbol{x}, y_i) = 1 - p_i(\boldsymbol{x}), \qquad (1)$$

where $p_i(\boldsymbol{x})$ represents the probability of different class $y_i$.

3. *Quantile Computation*. Given a target miscoverage rate $\alpha \in [0, 1]$, SCP obtains threshold $\hat{q}$ by taking the $1 - \alpha$ quantile of the non-conformity score of the ground truth labels $y_t$ on the calibration data $(\boldsymbol{x}, y_t) \in \mathcal{D}_c$,

$$\hat{q} = \text{Quantile}_{1-\alpha}\big(S(\boldsymbol{x}, y_t)\big). \qquad (2)$$

4. *Prediction Set*. For the data point $\boldsymbol{x}$ that needs to be tested, labels with non-conformity scores lower than the threshold $\hat{q}$ are selected for the final prediction set:

$$\mathbb{C}(\boldsymbol{x}) = \{y_i : S(\boldsymbol{x}, y_i) \leq \hat{q}\}, \qquad (3)$$

For the effects of calibration set size and the confidence level $(1 - \alpha)$, we provide an ablation study in Section 5.3.

---

[1]Note that while the goal is to remove the influence of the forget data so that it behaves similarly to the calibration data, the exchangeability property may not always hold in machine unlearning settings. Here, we are directly leveraging the concept of conformal prediction to evaluate machine unlearning performance.

*Table 1.* Unlearning performance measured by existing metrics across RT, FT and RL methods. All values in percent (%). The sign ↑ (↓) represents the greater (smaller) is better.

| Methods | 10% Random Forgetting | | | | 50% Random Forgetting | | | |
|---|---|---|---|---|---|---|---|---|
| | UA ↑ | RA ↑ | TA ↑ | MIA ↓ | UA ↑ | RA ↑ | TA ↑ | MIA ↓ |
| RT | 8.62 | 99.69 | 91.83 | 86.92 | 10.98 | 99.80 | 89.16 | 82.79 |
| FT | 3.84 | 98.14 | 91.57 | 92.00 | 2.59 | 99.08 | 91.77 | 92.92 |
| RL | 7.55 | 97.41 | 90.60 | 74.21 | 10.48 | 93.91 | 85.78 | 61.15 |

*Table 2.* Mis-label (mis-classification) count and in-set ratio of UA metric for RT, FT and RL on **CIFAR-10** with **ResNet-18** under **10%** and **50%** **random data forgetting** scenarios. In all settings, over 30% of mis-label data remains within the conformal prediction set in UA. More results of other unlearning methods can be found in Appendix D.

| Methods | 10% Random Forgetting | | | 50% Random Forgetting | | |
|---|---|---|---|---|---|---|
| | Mis-label ↑ | In-set ↓ | Ratio ↓ | Mis-label ↑ | In-set ↓ | Ratio ↓ |
| RT | 431 | 132 | 30.6% | 2,745 | 1,573 | 57.3% |
| FT | 192 | 112 | 58.3% | 647 | 431 | 66.6% |
| RL | 380 | 173 | 45.5% | 2,625 | 1,795 | 68.4% |

### 3.2. Identifying Fake Forgetting in UA

In this section, we show that a conformal prediction-based recovery technique can reconstruct the true label with high probability even when one forget data point is misclassified. This highlights a critical blind spot in the existing UA metric from the perspective of uncertainty quantification. The first key question we pose is as follows:

> **(Q1)** *Can we recover the data that is identified as forgotten by the metric UA?*

If the ground truth of forget data falls within the conformal prediction set, we consider the recovery successful. Thus, **fake forgetting is defined as the scenario where a data point identified as forgotten by UA can still be recovered by conformal prediction**.

To substantiate our claim, we first conduct a preliminary experiment. The following existing metrics are adopted: unlearning accuracy (**UA**, i.e., $1-$ the accuracy on forget data), retaining accuracy (**RA**, i.e., the accuracy on retain data), test accuracy (**TA**, i.e., the accuracy on test data), and membership inference attack (**MIA**). See Appendix C for MIA implementation details. We evaluate 3 classic unlearning methods, Retrain (**RT**), Finetune (**FT**) (Warnecke et al., 2021), and Random Label (**RL**) (Graves et al., 2021). See Appendix A for a detailed introduction to the baselines. The results are trained on CIFAR-10 with ResNet-18 in a random data forgetting scenario. In Table 1, the UA results suggest that the unlearning models can fail to correctly classify part of the forget data. However, can the higher UA fully guarantee that the true labels of these forgotten data do not appear in any form within the model's predictions?

We employ conformal prediction to investigate whether the ground truth of forget data can still be recovered, specifically,

whether it appears within the conformal prediction sets. The confidence level and calibration set size are set to 95% and 2000, respectively. In Table 2, we count the number of data points identified as truly forgotten by UA (marked as *mis-label*) and the number of these *mis-label* points that can still be recovered (marked as *in-set*).

The results of UA reveal that, even though the model mis-classifies part of the forget data, on average 54.6% of these misclassified instances are still recovered by conformal prediction. Even for the RT baseline, UA does not reliably assess whether a data point has truly been forgotten, as 30.6% of UA-misclassified points can still be recovered. **We emphasize that we are not claiming the gold standard baseline RT itself exhibits fake forgetting. Instead, we point out that UA does not reliably measure the forgetting quality of any unlearning method, including RT.** This finding demonstrates that a high UA does not guarantee that the model has truly forgotten the data, highlighting that relying solely on UA to evaluate forgetting quality is fragile. For additional results on other unlearning methods, see Table 5 in Appendix D.1.

Overall, the high *recovery ratio* observed in Tables 2 indicates that misclassified forget data cannot be considered truly forgotten, as their traces can be readily detected via conformal prediction from the perspective of uncertainty quantification. This encloses that **the fake forgetting issue arises when the true label of UA-misclassified data falls within the conformal prediction set**.

### 3.3. Designing Metric Motivated by Conformal Prediction

Based on the limitation of UA metric shown in Section 3.2, it raises a question as follows:

> **(Q2)** *Can we develop metrics to address the fake forgetting issue of UA?*

Thus, we propose an enhanced metric that draw intuition from conformal prediction.

To overcome the fake forgetting inherent in UA, we introduce a novel metric Conformal Ratio (CR), which incorporates both coverage and set size in conformal prediction to provide a more comprehensive evaluation. Before defining CR, we introduce Coverage and Set Size.

Given a dataset $\mathcal{D}$, the definition of **Coverage** is as follows:

$$\text{Coverage} := \frac{1}{|\mathcal{D}|} \sum_{(\boldsymbol{x}, y_t) \in \mathcal{D}} \mathbb{I}(y_t \in \mathbb{C}(\boldsymbol{x})), \qquad (4)$$

where $y_t$ is the true label of data point $\boldsymbol{x}$. Indicator function $\mathbb{I}(\cdot)$ returns 1 if the enclosed condition is true and 0 otherwise. Coverage reflects the probability that the true label falls within the prediction set $\mathbb{C}(\boldsymbol{x})$. For $\mathcal{D} = \mathcal{D}_f$,

high coverage indicates that the model retains significant information about forget data, suggesting fake forgetting.

Given a dataset $\mathcal{D}$, **Set Size** is defined as follows:

$$\text{Set Size} := \frac{1}{|\mathcal{D}|} \sum_{(\boldsymbol{x}, y_t) \in \mathcal{D}} |\mathbb{C}(\boldsymbol{x})|, \qquad (5)$$

where $|\mathbb{C}(\boldsymbol{x})|$ denotes the set size of data point $\boldsymbol{x}$. When $y_t \in \mathbb{C}(\boldsymbol{x})$, a small set size indicates that fewer non-ground truth classes are included in the prediction set, reflecting stronger fake forgetting.

Based on Coverage and Set Size, we introduce the definition of **CR** for a dataset $\mathcal{D}$ as follows:

$$\text{CR} := \frac{\text{Coverage}}{\text{Set Size}} = \frac{\sum_{(\boldsymbol{x}, y_t) \in \mathcal{D}} \mathbb{I}(y_t \in \mathbb{C}(\boldsymbol{x}))}{\sum_{(\boldsymbol{x}, y_t) \in \mathcal{D}} |\mathbb{C}(\boldsymbol{x})|}. \quad (6)$$

CR balances the information captured by Coverage and Set Size. A lower CR value implies stronger forgetting. CR is inspired by conformal prediction, which is proposed to assess the model's behavior on new and unseen data, not on the training data. Thus, we emphasize that CR only measures forget data $\mathcal{D}_f$ and test data $\mathcal{D}_{test}$.

**Superiority of Our Metrics.** Existing accuracy-based metric UA suffers from a fake forgetting issue, since true labels of misclassified data points may still remain within the prediction set. In contrast, our metric CR addresses this issue by examining the entire conformal prediction set, providing a more reliable evaluation of forgetting quality. Besides evidence in Tables 2, Figures 10-13 in the Appendix also support this superiority of our metrics.

> **Evaluation Criteria of Our Metrics**
>
> We consider two different criteria[2] to measure unlearning performance with our metrics,
> ❶ **Gap to RT Criterion**: A lower gap to the RT method is better. The gap relative to RT is represented in blue text (•) in our result tables.
> ❷ **Limit-Based Criterion**: For the CR, a lower CR value of forget data $\mathcal{D}_f$ indicates stronger forgetting performance, while a higher CR value of $\mathcal{D}_{test}$ represents higher preserved model utility. In this setting, RT is no longer a gold-standard baseline.

## 4. Enhancing Machine Unlearning via Conformal Prediction

Based on the findings in Section 3.2, we observe that existing training-based unlearning methods are typically optimized with respect to loss functions that do not directly

---

[2]The appropriate evaluation criteria vary across unlearning application scenarios (Kurmanji et al., 2023): criterion ❶ is particularly relevant for the user privacy scenario, while criterion ❷ focuses on the bias removal scenario, including biases, outdated or incorrect information.

support the improvement of forgetting quality from our fake forgetting perspective. Specifically, **the optimization objectives of existing methods fail to ensure that the ground truth labels are sufficiently pushed out of the conformal prediction set**, which is key to overcoming fake forgetting. This raises a critical question:

> **(Q4)** *Can we explore advanced unlearning techniques via conformal prediction to optimize the existing unlearning model's forgetting quality?*

Therefore, we propose a novel and general conformal prediction-based unlearning framework CPU tailored for training-based unlearning methods, aimed at enhancing their forgetting quality. A key insight driving our framework is to overcome the issue exposed by fake forgetting. This emphasizes that the non-conformity scores of ground truth labels for forget data should be pushed beyond the conformal prediction threshold $\hat{q}$. Interestingly, this goal aligns naturally with the design of the C&W attack loss (Carlini & Wagner, 2017), which motivates our creative adaptation to the unlearning scenario.

As a starting point, we adapt the original C&W loss to the unlearning scenario, which serves as an intermediate formulation before integrating conformal prediction in our final framework. For the forget data $\mathcal{D}_f$, the objective of the C&W-inspired unlearning loss is to decrease the model's confidence in the true labels of $\mathcal{D}_f$. Based on this, the C&W-inspired unlearning loss is defined as:

$$\mathcal{L}_{cw}(\boldsymbol{x}, y_t) = \max\{p_t(\boldsymbol{x}) - \max_{i \neq t}\{p_i(\boldsymbol{x})\}, -\Delta\}, \quad (7)$$

where $(\boldsymbol{x}, y_t) \in \mathcal{D}_f$ and $\max\{\cdot\}$ is an operator that selects the largest value. $p_i(\boldsymbol{x})$ is the probability of class $y_i$, and $p_t(\boldsymbol{x})$ refers specifically to the probability assigned to the true label $y_t$. We denote $\max_{i \neq t}\{p_i(\boldsymbol{x})\}$ as the highest probability value of the non-ground truth classes. This loss $\mathcal{L}_{cw}$ maximizes the difference between the highest probability value for class $y_i$ ($i \neq t$) and the probability value for the true class $y_t$. It tries to decrease the probability of the true class $y_t$ and further increase that of the class $y_i$ with the highest probability. The margin parameter $\Delta$ controls the enforced margin between the true class and the strongest competing class. When the $\max_{i \neq t}\{p_i(\boldsymbol{x})\} - p_t(\boldsymbol{x}) < \Delta$, this loss encourages the model to decrease the true label's probability $p_t(\boldsymbol{x})$. Increasing the value of $\Delta$ further increase the margin between $\max_{i \neq t}\{p_i(\boldsymbol{x})\}$ and $p_t(\boldsymbol{x})$.

With this C&W loss, we can indeed reduce the probability assigned to the true label $y_t$, thereby compelling the model to misclassify the data point into another class $y_i$. However, this loss still fails to guarantee that the true label $y_t$ can be excluded from the conformal prediction set. If we let the threshold in conformal prediction play the role of $\max_{i \neq t}\{p_i(\boldsymbol{x})\}$ in Eq. 7, and push the non-conformity

score of $y_t$ further away from this threshold, the above issue can be effectively resolved. We then extend this formulation by integrating conformal prediction to explicitly enforce the exclusion of the ground-truth label from the conformal prediction set.

In conformal prediction, calibration data helps in estimating non-conformity scores and determining a threshold to ensure valid statistical guarantees about the model's uncertainty estimates. A portion of calibration data $\mathcal{D}'_c$ can be reserved for the unlearning phase, which is kept separate from the calibration data $\mathcal{D}_c$ used in the evaluation phase. With calibration data $\mathcal{D}'_c$, the threshold $\bar{q}$ for the unlearning phase is easily calculated given an $\alpha$. Given $\bar{q}$, by revising C&W-inspired unlearning loss with a calibration step, a general unlearning loss function is defined as follows:

$$\mathcal{L}_{cpu}(\boldsymbol{x}, y_t) = \max\{\bar{q} - S(\boldsymbol{x}, y_t), -\Delta\}. \quad (8)$$

We replace probability $p_t(\boldsymbol{x})$ and $\max_{i \neq t}\{p_i(\boldsymbol{x})\}$ in Eq. 7 with the threshold $\bar{q}$ and non-conformity score $S(\boldsymbol{x}, y_t)$ respectively. $\bar{q}$ is updated in each training epoch to obtain an accurate value. Since $\bar{q}$ is computed merely as a quantile, this process incurs negligible computational overhead (experimental evidence is provided in Appendix E.4).

The loss $\mathcal{L}_{cpu}$ adheres to the same principle of $\mathcal{L}_{cw}$, which encourages $S(\boldsymbol{x}, y_t) - \bar{q} \geq \Delta$. It helps to increase the non-conformity score $S(\boldsymbol{x}, y_t)$ of the true label $y_t$ to surpass the threshold $\bar{q}$. As an improvement over the loss $\mathcal{L}_{cw}$, the loss $\mathcal{L}_{cpu}$ makes it more difficult for the model to include the true label in the conformal prediction set. In this loss, even a small value of $\Delta$ is sufficient to achieve the desired effect, because the true label $y_t$ is excluded from the conformal prediction set once its non-conformity score $S(\boldsymbol{x}, y_t)$ exceeds the threshold $\bar{q}$. Therefore, in our work, we set $\Delta = 0.01$.

As a general framework, to preserve the efficacy of existing unlearning methods themselves, we reserve their original loss $\mathcal{L}_{original}$ in our framework. Consequently, we combine these terms to form the final objective loss function as:

$$\mathcal{L}_{total} = \mathcal{L}_{original} + \lambda \cdot \mathcal{L}_{cpu}, \quad (9)$$

where $\lambda$ is a hyperparameter that controls the forgetting strength.

## 5. Experiment

### 5.1. Experimental setting

**Datasets and Models.** We mainly focus on the image classification task and report experiments on CIFAR-10 (Krizhevsky, 2009) and Tiny ImageNet (Le & Yang, 2015) datasets with ResNet-18 (He et al., 2016) and ViT (Dosovitskiy et al., 2021) architectures. See Appendix E.3 for the results on the Swin-T (Liu et al., 2021) architecture and the Oxford-Pets (Parkhi et al., 2012) dataset.

**Baselines and Metrics.** We employ **12 different unlearning methods**, including **RT**, **FT** (Warnecke et al., 2021), **RL** (Graves et al., 2021), **Gradient Ascent (GA)** (Thudi et al., 2022), **Bad Teacher (Teacher)** (Tarun et al., 2023), **SCRUB** (Kurmanji et al., 2023), **SSD** (Foster et al., 2024), **Neg-Grad+** (Kurmanji et al., 2023), **Salun** (Fan et al., 2024b), **SFRon** (Huang et al., 2025), **LoTUS** (Spartalis et al., 2025), **MUNBa** (Wu & Harandi, 2025). See Appendix A for a detailed overview of these unlearning methods. We evaluate the performance of various unlearning methods using the existing metrics, including **UA**, **RA**, **TA**, **MIA**, as well as our proposed metric **CR**. See Appendix C for the detailed introduction to MIA and our implementation.

**Implementation Details.** For hyperparameters, we set the miscoverage rate $\alpha = 0.05$. We also report the results for $\alpha \in \{0.10, 0.15, 0.20\}$ in Appendix D. The margin parameter is set to $\Delta = 0.01$, since our loss formulation ensures that a small margin is sufficient to effectively exclude the true label from the conformal prediction set. Additionally, the unlearning loss weight $\lambda$ is selected from $\{0, 0.2, 0.5, 1\}$. Further details regarding training configurations and baseline setups are included in Appendix B.

### 5.2. Measure Unlearning Methods via CR

In this section, we explore how existing unlearning methods perform with the consideration of the fake forgetting perspective. We evaluate the performance of 12 various unlearning methods using the proposed metric CR, together with Coverage and Set Size. The experimental results are presented in Table 3, which summarizes the unlearning performance under 10% random data forgetting scenario on CIFAR-10 and Tiny ImageNet, respectively. See Tables 11 - 18 in Appendix D for additional experimental results on other forgetting scenarios, including class-wise, subclass-wise and worst-case forgetting.

We take the results on CIFAR-10 as an example for analysis of CR on forget data $\mathcal{D}_f$ based on two evaluation criteria proposed in Section 3.3. According to evaluation criterion ❶, the top 4 methods under the UA metric are *NegGrad+*, *RL*, *SFRon*, and *Salun*, as their unlearning accuracy is closest to the RT method. However, this ranking shifts slightly under the CR metric, where the top 4 become *Salun*, *Neg-Grad+*, *SFRon*, and *RL*. CR metric identifies that Salun performs better in forgetting quality and can deal with the fake forgetting issue well, while RL faces a fake forgetting situation and performs poorly on our metric CR. This observation suggests that methods excelling in the traditional UA metric may not perform well under the CR metric. **The underlying rationale behind this is that the CR metric takes into account the possibility that the true labels of some misclassified forget data points may still remain within the prediction set**. This observation aligns with the insights we discussed in Section 3.2 regarding the fake

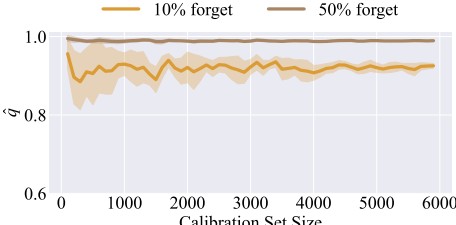

*Figure 2.* The stability of $\hat{q}$ in different calibration set sizes. When the calibration set size is greater than 1000, the fluctuations of $\hat{q}$ remain within a stable range.

forgetting issue of the UA metric.

Regarding evaluation criterion ❷, a similar pattern is observed as with criterion ❶. Under the UA metric, the top 4 methods in terms of forgetting quality are *NegGrad+*, *RT*, *RL* and *SFRon*. However, under the CR metric, the top 4 shift to *RL*, *RT*, *Salun* and *NegGrad+*. This indicates that some unlearning methods, such as NegGrad+, show poor forgetting quality when viewed from the fake forgetting perspective. This also highlights that the CR captures critical scenarios overlooked by UA, specifically the potential retention of true labels within prediction sets for the forget data points. CR ensures a more robust and reliable evaluation for unlearning quality.

### 5.3. Ablation Study for Confidence Level and Calibration Set Size in CR Metric

In conformal prediction, the confidence level $1 - \alpha$ (i.e., miscoverage rate $\alpha$) and calibration set size are two factors. We next discuss the suitable settings for these two factors and the rationale behind them.

**Confidence Level $1 - \alpha$.** A higher confidence level $1 - \alpha$, guarantees more reliable coverage. In the conformal prediction related works (Angelopoulos & Bates, 2021; Papadopoulos et al., 2002; Romano et al., 2020a; Tailor et al.), $\alpha = 0.05$ is widely adopted as a standard in most cases, reflecting its common use in statistical hypothesis testing to balance false positives and practical usability. Following prior work, we set $\alpha = 0.05$ by default, while also reporting results for higher values (0.10, 0.15, and 0.20) in Appendix D.5 to account for scenarios where a more relaxed confidence level is needed. Unless otherwise noted, all analyses use the default $\alpha = 0.05$.

**Calibration Set Size.** A portion of the validation data is set aside as calibration data, ensuring it remains independent from both the training and test data. The calibration set must be sufficient to avoid abnormal $\hat{q}$ values caused by outliers from small samples, which can destabilize coverage estimates. Figure 2 illustrates the stability of $\hat{q}$ across varying calibration set sizes on CIFAR-10 with ResNet-18. The results are smoothed using a B-spline. It shows that for different settings using ResNet-18 on CIFAR-10, after the calibration set size is larger than 1000, abnormal $\hat{q}$

*Table 3.* Unlearning performance on **CIFAR-10** with **ResNet-18** and **Tiny ImageNet** with **ViT** in **10%** **random data forgetting**. The results are average values from 3 independent trials and the standard deviation values are reported in Appendix D. For evaluation criterion ❶, performance differences compared to the RT method are highlighted with (•). For clarity in observing criterion ❷, the sign ↑ represents greater is better, while ↓ denotes ideally small. It shows the unlearning methods that excel under the existing metric UA do not necessarily perform well under our CR metric.

| Methods | Existing Metrics | | | | Coverage | | Set Size | | CR | |
| --- | --- | --- | --- | --- | --- | --- | --- | --- | --- | --- |
| | UA ↑ | RA ↑ | TA ↑ | MIA ↑ | $\mathcal{D}_f \downarrow$ | $\mathcal{D}_{test} \uparrow$ | $\mathcal{D}_f \uparrow$ | $\mathcal{D}_{test} \downarrow$ | $\mathcal{D}_f \downarrow$ | $\mathcal{D}_{test} \uparrow$ |
| **CIFAR-10 with ResNet-18** | | | | | | | | | | |
| RT | 8.6%(0.0) | 99.7%(0.0) | 91.8%(0.0) | 86.9%(0.0) | 0.941(0.000) | 0.944(0.000) | 1.089(0.000) | 1.074(0.000) | 0.864(0.000) | 0.879(0.000) |
| FT | 3.8%(4.8) | 98.1%(1.6) | 91.6%(0.2) | 92.0%(5.1) | 0.994(0.053) | 0.951(0.007) | 1.008(0.081) | 1.026(0.048) | 0.986(0.122) | 0.927(0.048) |
| RL | 7.6%(1.0) | 97.4%(2.3) | 90.6%(1.2) | 74.2%(12.7) | 0.970(0.029) | 0.949(0.005) | 1.242(0.153) | 1.197(0.123) | 0.788(0.076) | 0.796(0.083) |
| GA | 0.6%(8.0) | 99.5%(0.2) | 94.1%(2.3) | 98.8%(11.9) | 0.994(0.053) | 0.945(0.001) | 1.002(0.087) | 1.009(0.065) | 0.994(0.130) | 0.936(0.057) |
| Teacher | 0.8%(7.8) | 99.4%(0.3) | 93.5%(1.7) | 87.2%(0.3) | 0.991(0.050) | 0.941(0.003) | 1.003(0.086) | 1.021(0.053) | 0.993(0.129) | 0.922(0.043) |
| SSD | 0.5%(8.1) | 99.5%(0.2) | 94.2%(2.4) | 98.8%(11.9) | 0.996(0.055) | 0.945(0.001) | 0.999(0.090) | 1.008(0.066) | 0.994(0.130) | 0.936(0.057) |
| NegGrad+ | 8.7%(0.1) | 98.8%(0.9) | 92.2%(0.4) | 90.3%(3.4) | 0.934(0.007) | 0.948(0.004) | 1.068(0.021) | 1.086(0.012) | 0.875(0.011) | 0.873(0.006) |
| SCRUB | 3.4%(5.2) | 98.0%(1.7) | 92.0%(0.2) | 93.8%(6.9) | 0.984(0.043) | 0.946(0.002) | 1.060(0.029) | 1.085(0.011) | 0.952(0.088) | 0.873(0.006) |
| LoTUS | 0.7%(7.9) | 99.2%(0.5) | 93.5%(1.7) | 97.8%(10.9) | 0.996(0.055) | 0.955(0.011) | 1.014(0.075) | 1.063(0.011) | 0.980(0.116) | 0.899(0.020) |
| MUNBa | 0.6%(8.0) | 99.8%(0.1) | 94.4%(2.6) | 96.3%(9.4) | 0.050(0.891) | 0.998(0.054) | 0.994(0.095) | 0.944(0.130) | 0.998(0.134) | 0.996(0.117) |
| Salun | 3.7%(4.9) | 98.9%(0.8) | 91.8%(0.0)) | 57.6%(29.3) | 0.987(0.046) | 0.950(0.006) | 1.132(0.043) | 1.143(0.069) | 0.872(0.008) | 0.832(0.047) |
| SFRon | 4.8%(3.8) | 97.4%(2.3) | 91.4%(0.4) | 91.6%(4.7) | 0.977(0.036) | 0.953(0.009) | 1.100(0.011) | 1.143(0.069) | 0.889(0.025) | 0.834(0.045) |
| **Tiny ImageNet with ViT** | | | | | | | | | | |
| RT | 14.7%(0.0) | 98.8%(0.0) | 86.0%(0.0) | 85.4%(0.0) | 0.944(0.000) | 0.949(0.000) | 1.876(0.000) | 1.840(0.000) | 0.503(0.000) | 0.516(0.000) |
| FT | 6.9%(7.8) | 97.9%(0.9) | 84.1%(1.9) | 91.7%(6.3) | 0.994(0.050) | 0.950(0.001) | 2.133(0.257) | 2.440(0.600) | 0.466(0.037) | 0.389(0.127) |
| RL | 26.9%(12.2) | 96.0%(2.8) | 81.4%(4.6) | 72.4%(13.0) | 0.969(0.025) | 0.952(0.003) | 17.890(16.014) | 8.572(6.732) | 0.054(0.449) | 0.111(0.405) |
| GA | 3.2%(11.5) | 97.4%(1.4) | 84.9%(1.1) | 99.7%(14.3) | 0.996(0.052) | 0.947(0.002) | 1.539(0.337) | 2.018(0.178) | 0.647(0.144) | 0.469(0.047) |
| Teacher | 17.3%(2.6) | 86.7%(12.1) | 79.0%(7.0) | 82.7%(2.7) | 0.977(0.033) | 0.956(0.006) | 5.473(3.597) | 5.080(3.240) | 0.179(0.324) | 0.188(0.328) |
| SSD | 1.5%(13.2) | 98.5%(0.3) | 86.1%(0.1) | 96.8%(11.4) | 0.998(0.054) | 0.950(0.001) | 1.354(0.522) | 1.827(0.013) | 0.737(0.234) | 0.520(0.004) |
| NegGrad+ | 19.4%(4.7) | 98.3%(0.5) | 84.0%(2.0) | 91.1%(5.7) | 0.999(0.055) | 0.890(0.059) | 0.949(0.927) | 1.614(0.227) | 1.052(0.823) | 0.552(1.289) |
| SCRUB | 1.3%(13.4) | 98.7%(0.1) | 86.3%(0.3) | 94.1%(8.7) | 0.998(0.054) | 0.950(0.001) | 1.309(0.567) | 1.769(0.071) | 0.761(0.257) | 0.537(0.021) |
| LoTUS | 7.5%(7.2) | 85.6%(13.2) | 78.6%(7.4) | 89.7%(4.3) | 0.995(0.051) | 0.949(0.000) | 2.747(0.871) | 3.711(1.871) | 0.290(0.213) | 0.256(0.260) |
| MUNBa | 1.9%(12.8) | 99.0%(0.2) | 86.2%(0.2) | 91.6%(6.2) | 0.998(0.054) | 0.949(0.000) | 1.408(0.468) | 1.844(0.003) | 0.770(0.267) | 0.515(0.001) |
| Salun | 9.2%(5.5) | 97.7%(1.1) | 83.6%(2.4) | 70.0%(15.4) | 0.995(0.051) | 0.964(0.015) | 2.803(0.927) | 2.726(0.886) | 0.528(1.347) | 0.376(1.464) |
| SFRon | 9.3%(5.4) | 97.0%(1.8) | 83.9%(2.1) | 92.5%(7.1) | 0.989(0.045) | 0.948(0.001) | 2.000(0.124) | 2.208(0.368) | 0.495(0.008) | 0.429(0.086) |

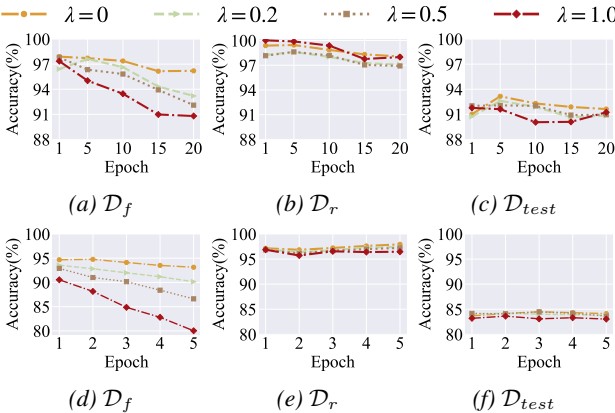

— • — $\lambda = 0$  — — — $\lambda = 0.2$  ·····■···· $\lambda = 0.5$  —♦— $\lambda = 1.0$

(a) $\mathcal{D}_f$   (b) $\mathcal{D}_r$   (c) $\mathcal{D}_{test}$

(d) $\mathcal{D}_f$   (e) $\mathcal{D}_r$   (f) $\mathcal{D}_{test}$

*Figure 3.* CPU accuracy of $\mathcal{D}_f$, $\mathcal{D}_r$ and $\mathcal{D}_{test}$ under different $\lambda$ values across each epoch on CIFAR-10 (a-c) and Tiny ImageNet (d-f). As $\lambda$ increases, accuracy on $\mathcal{D}_f$ drops significantly, while retain and test accuracy remain stable.

values do not occur anymore, and a stable threshold $\hat{q}$ can be obtained in both 10% and 50% random data forgetting scenarios. Similarly, we analyze the calibration set size of the class-wise forgetting scenario and find that fewer calibration data points are required compared to random data forgetting. This is because the targeted class forgetting reduces the complexity of the distribution, unlike the broader variability introduced by random data forgetting. See the results on the class-wise forgetting scenario and the results on Tiny-ImageNet in Appendix D.3.

### 5.4. Performance of Our Unlearning Framework

In this experiment, we apply RT, FT, and RL methods to our framework CPU, i.e., CPU-RT, CPU-FT and CPU-RL. Table 4 presents the results for CIFAR-10 with ResNet-18 and Tiny ImageNet with ViT in 10% random data forgetting. We vary $\lambda$ in the range $[0, 0.2, 0.5, 1]$, where $\lambda = 0$ represents the method without our framework applied. See Tables 7–8 in Apendix E for additional baselines incorporating CPU, as well as a broader range of architectures and datasets.

From the perspective of evaluation criterion ❶, we take CPU-FT as an example for analysis. The gap (blue text (•)) between CPU-FT and RT on the existing metric UA decreases effectively as $\lambda$ increases. Specifically, the UA gap decreases from 4.8% to 0.7% on CIFAR-10 and from 7.8% to 0.9% on Tiny ImageNet. It is worth noting that the model utility remains relatively stable on the RA and TA results. Similarly, $\text{CR}_{\mathcal{D}_f}$ metric is also decreased when $\lambda > 0$. For the average gap across UA, RA, and TA metrics, the CPU-FT method achieves a promising average gap of 1.47 on ResNet-18 when $\lambda = 0.5$, compared to an average gap of 2.2 when $\lambda = 0$. Similarly, on the ViT model, CPU-FT reduces the average gap from 3.53 to 1.63 when $\lambda = 0.5$. It is obvious that our framework can strongly improve forgetting strength. That means the methods that are prone to over-forgetting, such as RL, perform adequately without requiring CPU for additional enhancements under

*Table 4.* Performance of our unlearning framework CPU. We show the performance on **CIFAR-10** with **ResNet-18** and **Tiny ImageNet** with **ViT** in **10% random data forgetting**. $\lambda = 0$ represents the baseline without our framework applied. It shows our CPU significantly improves the forgetting quality, not only across our metric CR but also across UA, while preserving stable predictive performance.

| Methods | $\lambda = 0$ | | | | | | $\lambda = 0.2$ | | | | | |
| | UA ↑ | RA ↑ | TA ↑ | MIA ↓ | $CR_{\mathcal{D}_f}$ ↓ | $CR_{\mathcal{D}_{test}}$ ↑ | UA ↑ | RA ↑ | TA ↑ | MIA ↓ | $CR_{\mathcal{D}_f}$ ↓ | $CR_{\mathcal{D}_{test}}$ ↑ |
|---|---|---|---|---|---|---|---|---|---|---|---|---|
| **CIFAR-10 with ResNet-18** | | | | | | | | | | | | |
| CPU-RT | 8.6%(0.0) | 99.7%(0.0) | 91.8%(0.0) | 86.9%(0.0) | 0.864(0.000) | 0.879(0.000) | 10.8%(2.2) | 98.3%(1.4) | 91.0%(0.8) | 86.2%(0.7) | 0.788(0.076) | 0.824(0.055) |
| CPU-FT | 3.8%(4.8) | 98.1%(1.6) | 91.6%(0.2) | 92.0%(5.1) | 0.986(0.122) | 0.927(0.048) | 6.8%(1.8) | 97.0%(2.7) | 90.8%(1.0) | 90.7%(3.8) | 0.844(0.020) | 0.829(0.050) |
| CPU-RL | 7.6%(1.0) | 97.4%(2.3) | 90.6%(1.2) | 74.2%(12.7) | 0.788(0.076) | 0.796(0.083) | 9.7%(1.1) | 96.6%(3.1) | 89.4%(2.4) | 73.8%(13.1) | 0.709(0.155) | 0.736(0.143) |
| **Tiny ImageNet with ViT** | | | | | | | | | | | | |
| CPU-RT | 14.7%(0.0) | 98.8%(0.0) | 86.0%(0.0) | 85.4%(0.0) | 0.503(0.000) | 0.516(0.000) | 19.3%(4.6) | 98.8%(0.0) | 86.0%(0.0) | 84.8%(0.6) | 0.458(0.045) | 0.516(0.000) |
| CPU-FT | 6.9%(7.8) | 97.9%(0.9) | 84.1%(1.9) | 91.7%(6.3) | 0.466(0.037) | 0.389(0.127) | 9.8%(4.9) | 97.4%(1.4) | 83.6%(2.4) | 90.2%(4.8) | 0.441(0.062) | 0.399(0.117) |
| CPU-RL | 26.9%(12.2) | 96.0%(2.8) | 81.4%(4.6) | 72.4%(13.0) | 0.054(0.449) | 0.111(0.405) | 31.8%(17.1) | 95.3%(3.5) | 80.9%(5.1) | 71.4%(14.0) | 0.051(0.452) | 0.111(0.405) |

| Methods | $\lambda = 0.5$ | | | | | | $\lambda = 1.0$ | | | | | |
| | UA ↑ | RA ↑ | TA ↑ | MIA ↓ | $CR_{\mathcal{D}_f}$ ↓ | $CR_{\mathcal{D}_{test}}$ ↑ | UA ↑ | RA ↑ | TA ↑ | MIA ↓ | $CR_{\mathcal{D}_f}$ ↓ | $CR_{\mathcal{D}_{test}}$ ↑ |
|---|---|---|---|---|---|---|---|---|---|---|---|---|
| **CIFAR-10 with ResNet-18** | | | | | | | | | | | | |
| CPU-RT | 14.0%(5.4) | 97.8%(1.9) | 90.4%(0.4) | 85.8%(1.1) | 0.763(0.101) | 0.825(0.054) | 17.7%(9.1) | 96.8%(2.9) | 90.5%(1.3) | 84.1%(2.8) | 0.719(0.145) | 0.820(0.059) |
| CPU-FT | 7.9%(0.7) | 96.9%(2.8) | 90.9%(0.9) | 88.9%(2.0) | 0.853(0.011) | 0.843(0.036) | 9.2%(0.6) | 97.9%(1.8) | 91.2%(0.6) | 86.5%(0.4) | 0.835(0.029) | 0.854(0.025) |
| CPU-RL | 9.9%(1.3) | 96.9%(2.8) | 89.7%(2.1) | 72.9%(14.0) | 0.708(0.156) | 0.731(0.148) | 12.6%(4.0) | 95.3%(4.4) | 88.1%(3.7) | 71.1%(15.8) | 0.629(0.235) | 0.669(0.210) |
| **Tiny ImageNet with ViT** | | | | | | | | | | | | |
| CPU-RT | 26.4%(11.7) | 98.7%(0.1) | 85.8%(0.2) | 84.3%(1.1) | 0.396(0.107) | 0.489(0.027) | 35.7%(21.0) | 98.6%(0.2) | 85.2%(0.8) | 82.1%(3.3) | 0.346(0.157) | 0.481(0.035) |
| CPU-FT | 13.6%(0.9) | 97.2%(1.6) | 83.6%(2.4) | 89.3%(3.9) | 0.413(0.090) | 0.401(0.115) | 20.0%(5.3) | 96.4%(2.4) | 82.9%(3.1) | 86.0%(0.6) | 0.342(0.161) | 0.363(0.153) |
| CPU-RL | 36.2%(21.5) | 95.3%(3.5) | 80.4%(5.6) | 70.1%(15.3) | 0.051(0.452) | 0.121(0.395) | 40.2%(25.5) | 94.5%(4.3) | 79.5%(6.5) | 68.5%(16.9) | 0.048(0.455) | 0.119(0.397) |

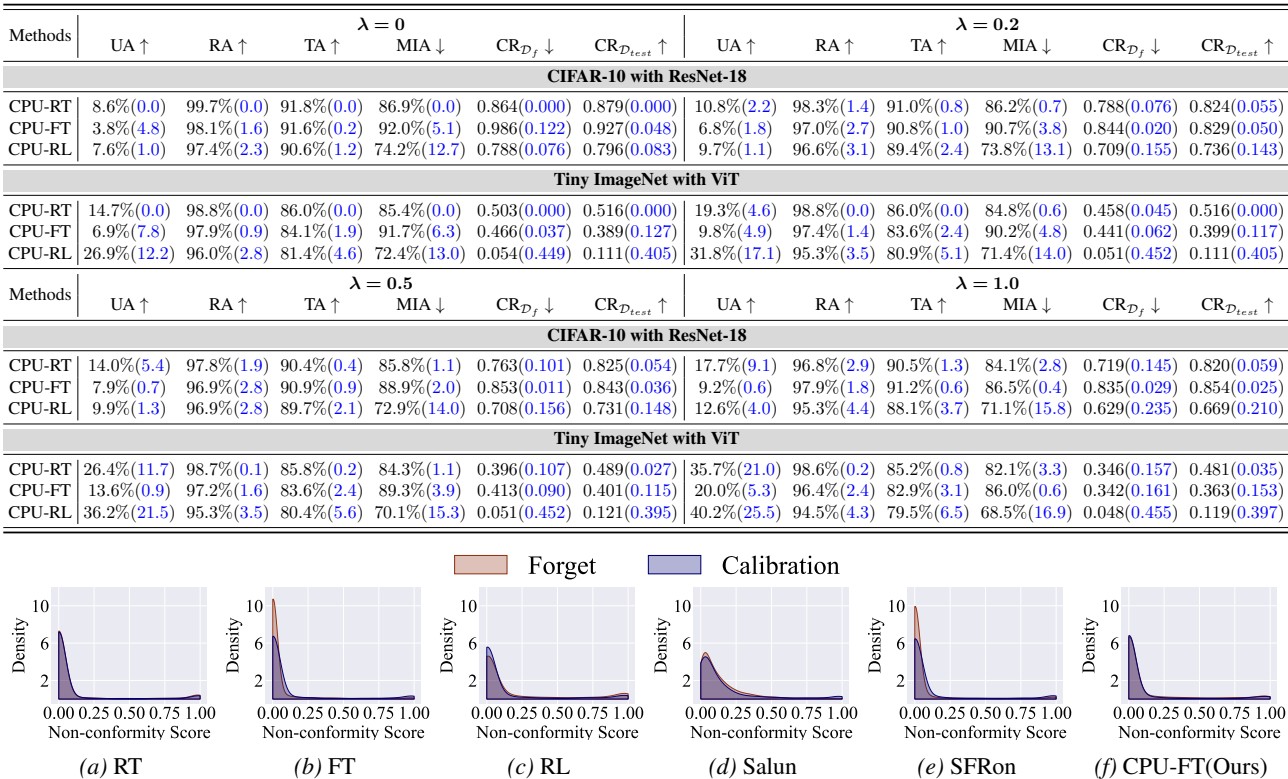

*Figure 4.* Distribution of non-conformity scores. Our CPU-FT closely matches the RT baseline, achieving alignment between the distributions of forgetting data and calibration data.

our evaluation criterion ❶.

For evaluation criterion ❷, when $\lambda = 0.5$, the UA improves by an average of $3.93\%$ on ResNet-18 and $9.23\%$ on ViT over all methods, while TA decreases only slightly by $1.0\%$ and $0.57\%$ on ResNet-18 and ViT respectively. As similarly shown in the CR metric, the value of $CR_{\mathcal{D}_{test}}$ remains nearly unchanged compared to the baseline ($\lambda = 0$) with only 0.03 drop on average, while $CR_{\mathcal{D}_f}$ shows a greater reduction with an average of 0.08 across all methods.

Moreover, in Figure 3, we further present the CPU-FT accuracy on forget data $\mathcal{D}_f$, retain data $\mathcal{D}_r$ and test data $\mathcal{D}_{test}$ under different $\lambda$ values across each epoch on Tiny ImageNet with ViT for $10\%$ random data forgetting. As $\lambda$ increases, the accuracy on $\mathcal{D}_f$ drops quickly, showing stronger unlearning effectiveness, while the accuracy on $\mathcal{D}_r$ and $\mathcal{D}_{test}$ remains stable. In summary, the experimental results demonstrate that our framework significantly improves forgetting quality while maintaining stable model utility. In addition, our CPU does not introduce noticeable additional training overhead, as shown in Table 9 in Appendix E.4.

Additionally, we visualize the distributions of non-conformity scores on ResNet-18 with CIFAR-10 for both the forgetting and calibration data in Figure 4. Since the calibration data consist of unseen samples, aligning the non-conformity score distribution of the forget data with that of the calibration set provides a natural and principled objective for evaluating unlearning quality. Such alignment signifies that the model treats the forgetting samples as if they had never been seen, consistent with the behavior of the RT baseline. The results show that under our CPU-FT ($\lambda = 1.0$), the distribution of the forget data is the closest to the calibration data and most closely matches the gold-standard RT baseline. For a comprehensive comparison, the non-conformity score distributions for other unlearning methods are provided in Figure 9 in Appendix F.

## 6. Conclusion

Motivated by conformal prediction, we introduce new metric, CR, to enhance the evaluation reliability of machine unlearning. In addition, our unlearning framework CPU, which incorporates the adapted C&W loss with conformal prediction, improves unlearning quality. Together, we provide a more rigorous foundation for machine unlearning.

## Impact Statement

This paper presents work whose goal is to advance the field of Machine Learning. There are many potential societal consequences of our work, none of which we feel must be specifically highlighted here.

## Acknowledgments

This work was supported in part by the National Science Foundation under grants IIS-2246157, FMitF-2319243, and the Department of Energy under grant DE-CR0000042. The project was also supported by computational resources provided by the NSF ACCESS and Argonne National Lab.

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

# Appendix

## A. Baseline Details

We introduce the details of our unlearning baselines as follows : **RT** retrains the model from scratch using only the remaining dataset $\mathcal{D}_r$. **FT** (Warnecke et al., 2021) fine-tunes the pre-trained model $\theta_o$ on the remaining dataset $\mathcal{D}_r$. **RL** (Graves et al., 2021) fine-tunes the model on the forget dataset $\mathcal{D}_f$ using randomly assigned labels to enforce forgetting. **GA** (Thudi et al., 2022) performs gradient ascent on the forget data $\mathcal{D}_f$, which often harms the model's utility. **Teacher** (Tarun et al., 2023) distills knowledge from a corrupted teacher model to the student, aiming to uniformly increase the loss on forgetting samples but often causing catastrophic forgetting. **SSD** (Foster et al., 2024) induces forgetting by identifying and dampening parameters highly associated with the forgetting set using the Fisher information matrix, without retraining. **NegGrad+** (Kurmanji et al., 2023) addresses GA's issue by combining fine-tuning on $\mathcal{D}_r$ and gradient ascent on $\mathcal{D}_f$. **SCRUB** (Kurmanji et al., 2023) uses the original model as a teacher and optimizes the student to minimize the KL divergence on the retain data while maximizing it on the forget data. **Salun** (Fan et al., 2024b) performs unlearning by optimizing only the salient parameters of the model identified from the randomly labeled forget data. **SFRon** (Huang et al., 2025) embeds the unlearning update into the parameter manifold shaped by the retained data using Hessian modulation, approximated via a fast-slow update strategy. **LoTUS** (Spartalis et al., 2025) (Spartalis et al., 2025) utilizes knowledge distillation with dynamic logits tempering and Gumbel noise to systematically increase the prediction entropy on the forget data. **MUNBa** (Wu & Harandi, 2025) poses machine unlearning as a cooperative bargaining game between forgetting and preservation players, using a Nash bargaining closed-form solution to resolve gradient conflicts and achieve an optimal Pareto stationary point.

## B. Setting Details

For CIFAR-10 with ResNet-18 architecture, we train the original model from scratch for 200 epochs using SGD with a Cosine Annealing learning rate schedule, starting from an initial learning rate of 0.1. We set the momentum to 0.9 and a batch size of 64. The RT model adopts the same training configuration. Other models are trained for the following durations: FT for 20 epochs, GA for 1 epoch (to avoid over-forgetting and significant RA degradation), NegGrad+ for 10 epochs (reduced to 2 epochs in class-wise scenarios), RL, SCRUB, SFRon, Salun, LoTUS and MUNBa for 10 epochs. All other hyperparameters match those of the original model.

For the ViT architecture, we initialize the original model by training a pretrained ViT model for 15 epochs on Tiny ImageNet. We start with a learning rate of 0.001, while other training parameters match those used for ResNet-18. We use SGD and set the momentum to 0.9 and a batch size of 64. The RT model follows the same training procedure as the original model. Other models are trained for the following durations: GA for 1 epoch, and all other methods for 5 epochs. All other hyperparameters are consistent with the original model's training.

For CIFAR-10/Tiny ImageNet, we randomly select 200/50 data points per class (2000/10000 data points in total) as calibration data $\mathcal{D}_c$ and $\mathcal{D}_c'$, respectively. The calibration data $\mathcal{D}_c$ does not participate in the model training or unlearning processes and is only used for calibrating the threshold $\hat{q}$, while $\mathcal{D}_c'$ is used in the process of our unlearning framework to generate $\bar{q}$. All experiments are conducted on 1 Tesla V100-SXM2 GPU card with 32GB memory in a single node.

## C. MIA Implementation Details

Following prior works (Jia et al., 2023; Kurmanji et al., 2023; Zhao et al., 2024; Song et al., 2019; Yeom et al., 2018), we adopt a confidence-based membership inference attack to evaluate the privacy preservation of the unlearning model. Specifically, we construct an MIA predictor by training it on a balanced dataset sampled from the retain set $\mathcal{D}_r$ (labeled as members) and the test set $\mathcal{D}_{test}$ (labeled as non-members). The trained support vector classifier (SVC) is then applied to the unlearning model $\theta_u$ during evaluation.

To measure unlearning effectiveness, we compute the MIA success rate, which quantifies how many samples in the forget set $\mathcal{D}_f$ are still predicted as training members by the MIA predictor. Formally,

$$\text{MIA} = \frac{\text{TP}}{|\mathcal{D}_f|}, \tag{10}$$

where TP represents the count of forget samples still identified as training samples and $|\mathcal{D}_f|$ is the size of the forget data $\mathcal{D}_f$.

Intuitively, since the MIA score reflects the success rate of membership inference attacks on the forget data, a lower score indicates that less membership information about $\mathcal{D}_f$ is retained in $\theta_u$, implying stronger privacy preservation and more effective unlearning.

## D. Evaluating MU Methods under CR

### D.1. Mis-label Number and In-set Ratios under UA

*Table 5.* Mis-label number and in-set ratios of UA metric.

| Methods | 10% Forgetting | | | 50% Forgetting | | |
|---|---|---|---|---|---|---|
| | Mis-label ↑ | In-set ↓ | Ratio ↓ | Mis-label ↑ | In-set ↓ | Ratio ↓ |
| RT | 431 | 132 | 30.6% | 2,745 | 1,573 | 57.3% |
| FT | 192 | 112 | 58.3% | 647 | 431 | 66.6% |
| RL | 380 | 173 | 45.5% | 2,625 | 1,795 | 68.4% |
| GA | 30 | 2 | 6.7% | 150 | 9 | 6.0% |
| Teacher | 40 | 4 | 10% | 400 | 37 | 9.3% |
| SSD | 25 | 2 | 8.0% | 116 | 9 | 7.8% |
| SCRUB | 170 | 117 | 62.2 | 550 | 271 | 49.3% |
| LoTUS | 35 | 6 | 17.1% | 175 | 20 | 11.4% |
| MUNBa | 30 | 14 | 46.7% | 400 | 201 | 50.3% |
| NegGrad+ | 435 | 115 | 26.4% | 711 | 249 | 35.5% |
| Salun | 185 | 117 | 63.2% | 1,065 | 695 | 65.3% |
| SFRon | 240 | 125 | 52.1% | 1,000 | 610 | 61.0% |

Conformal prediction is applied to UA predictions to determine the number of misclassified data points (mis-label) and the number of these points that fall within the conformal prediction set (in-set). We evaluate both the UA metric by counting the misclassified data points and calculating how many of them are included in the conformal prediction set. The detailed results are presented in Table 5, which is the extended results of Table 2.

### D.2. Mis-label Number and In-set Ratios under MIA

As shown in Table 6, a similar fake forgetting phenomenon also occurs on **MIA**, supporting the broader value of our uncertainty-based perspective. The key insight in our work not only reveals limitations in UA, but can also be extended to other accuracy-based evaluation metrics. In MIA, '0' indicates a data point is forgotten, while '1' means it is still identified as a training member. The *mis-label* column of MIA refers to the number of data points that are predicted as '0'. The *in-set* here refers to the number of *mis-label* data points whose conformal prediction set still includes '1'. Thus, the *recover ratio* indicates that, although the MIA fails to identify an average of 18.33% of the forget data as training membership, conformal prediction can still recover 54.7% of these forget data within prediction sets.

### D.3. Calibration Set Size

Figure 5 illustrates the stability of $\hat{q}$ across varying calibration set sizes under random and class-wise forgetting on CIFAR-10 with ResNet-18 (Figure 5(a) and (b)), as well as under random forgetting on Tiny-ImageNet with ViT (Figure 5(c)). The results are smoothed using a B-spline. It shows that for different settings using ResNet-18 on CIFAR-10, after the calibration set size is larger than 1000, abnormal $\hat{q}$ values do not occur anymore, and a stable threshold $\hat{q}$ can be obtained in both 10% and 50% random data forgetting scenarios. Similarly, we analyze the calibration set size of the class-wise forgetting scenario and find that fewer calibration data points are required compared to random data forgetting. This is because the targeted class forgetting reduces the complexity of the distribution, unlike the broader variability introduced by random data forgetting. See the results on the class-wise forgetting scenario and the results on Tiny-ImageNet in Appendix D.3.

### D.4. Distribution Comparison of Forgotten Data on UA and CR

As shown in Figures 10-13, we further analyze the probability and loss distributions of ground truth labels for data identified as truly forgotten by CR (i.e., out-set) and UA (i.e., mis-label), respectively. The distribution curves are fitted using KDE for clearer visualization. The softmax outputs for 'out-set' are consistently near 0 compared to 'mis-label', which strongly suggests that 'out-set' more rigorously captures real forgotten data. In the cross-entropy loss distribution, forgotten data

| | 10% **Random Forgetting** | | | 50% **Random Forgetting** | | |
|---|---|---|---|---|---|---|
| Methods | Mis-label ↑ | In-set ↓ | Ratio ↓ | Mis-label ↑ | In-set ↓ | Ratio ↓ |
| **Mis-label and In-set Ratio of MIA** | | | | | | |
| RT | 654 | 209 | 32.0% | 4,303 | 1,391 | 32.3% |
| FT | 400 | 216 | 54.0% | 1,769 | 813 | 46.0% |
| RL | 1,289 | 1,011 | 78.4% | 9,713 | 8,295 | 85.4% |

*Table 6.* Mis-label (mis-classification) counts and in-set ratios of MIA metric for RT, FT, and RL on **CIFAR-10** with **ResNet-18** under **10%** and **50%** random data forgetting scenarios. In all settings, over 30% of mis-labeled samples remain within the conformal prediction set in MIA, confirming that the fake forgetting phenomenon also appears in MIA metric.

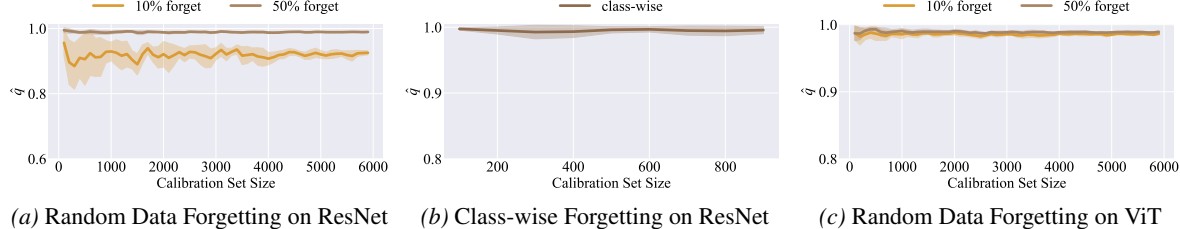

*(a)* Random Data Forgetting on ResNet    *(b)* Class-wise Forgetting on ResNet    *(c)* Random Data Forgetting on ViT

*Figure 5.* The stability of $\hat{q}$ in different calibration set sizes.

identified by CR consistently show higher cross-entropy loss than UA. Higher loss indicates better forgetting quality, which further validates that CR better removes fake forget data.

### D.5. CR Metric

Tables 13 and 14 show the unlearning performance on CIFAR-10 with ResNet-18 in 10% and 50% random data forgetting scenarios, while Table 15 is the results in class-wise forgetting scenario. Tables 16 and 17 present the unlearning performance on Tiny ImageNet with ResNet-18 in the random data forgetting scenario, while Table 18 details the unlearning performance in the class-wise forgetting scenario. For class-wise forgetting scenario, we note $\mathcal{D}_{test} = \mathcal{D}_{tf} \cup \mathcal{D}_{tr}$. $\mathcal{D}_{tf}$ corresponds to the test-forget data exclusively containing the forget class, while $\mathcal{D}_{tr}$ represents the test-retain data within the test data $\mathcal{D}_{test}$.

For all unlearning methods, as $\alpha$ level increases, it results in reduced Coverage and smaller Set Size. This happens because a higher $\alpha$ loosens the conformal threshold $\hat{q}$, allowing fewer predictions to be included within the prediction set for each data point. On the contrary, the CR tends to increase with increasing $\alpha$. Although both Coverage and Set Size may decrease, Set Size often decreases more significantly. Consequently, the CR value of $\mathcal{D}_f$ generally becomes larger as $\alpha$ increases. It is natural that the adjustment of $\alpha$ affects both Coverage and Set Size. However, the final CR value really depends on the model's performance itself. For a strict evaluation, we encourage setting $\alpha$ to 0.5.

When $\alpha$ is set to 0.2, most methods show a value of Set Size less than 1 in both Tables 13, 14, 16, 17. The intuition behind it is that conformal prediction, as a static predictor, is intrinsically tied to the model's base prediction performance and accuracy. When the model's accuracy is significantly higher than the confidence level, conformal prediction can achieve the required coverage with ease. In fact, it can generate partial empty prediction sets for some data points while still meeting the target coverage. Thus, the choice of $\alpha$ is crucial. Overly high $\alpha$ values may skew evaluation results by failing to let CR accurately reflect model performance. Therefore, we emphasize that a small $\alpha$ is generally appropriate for most unlearning scenarios.

Notably, the insights gained from the random data forgetting scenario can also be extended to the class-wise forgetting scenario. Additionally, in the class-wise scenario, some unlearning methods like RT and RL with UA = 100% and CR approaching 0% indicate they are truly effective at forgetting the specified class.

### D.6. Measuring Forgetting under Distribution Shifts

RL and Salun are unlearning methods that employ label corruption in their unlearning strategy, which can cause distribution shifts. Here, we introduce how to better measure forgetting under these circumstances. Figure 6(a) shows the non-conformity score distribution of calibration data $\mathcal{D}_c$ and forget data $\mathcal{D}_f$ in the unlearning model $\theta_u$ obtained by the RL method in Tiny ImageNet with ViT. It looks like there is a significant discrepancy between the distribution of the forget data and the

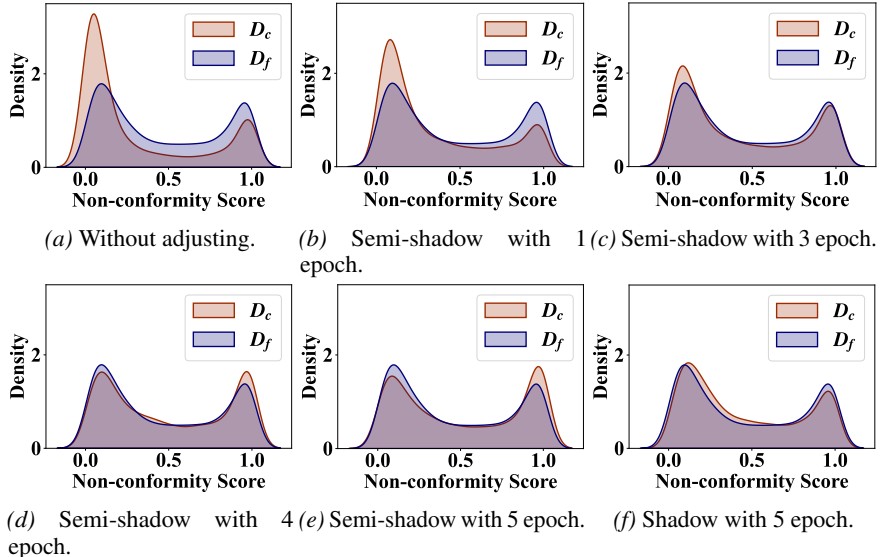

*(a)* Without adjusting.  *(b)* Semi-shadow with 1 *(c)* Semi-shadow with 3 epoch.
epoch.

*(d)* Semi-shadow with 4 *(e)* Semi-shadow with 5 epoch.  *(f)* Shadow with 5 epoch.
epoch.

*Figure 6.* Distribution shifting processing with different strategies. The distribution of calibration data gradually converges with that of forget data.

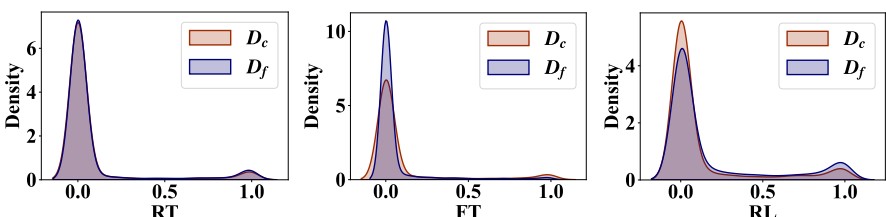

*Figure 7.* Non-conformity density of calibration data $\mathcal{D}_c$ and forget data $\mathcal{D}_f$ **without our unlearning framework** in CIFAR-10 with ResNet-18 under $10\%$ random data forgetting scenario.

calibration data.

To align the distribution of $\mathcal{D}_c$ with that of $\mathcal{D}_f$ and minimize the differences between them, we design a shadow model. To make the explanation clearer and more intuitive, we take RL as an example. In the RL unlearning method, the forget data is assigned random labels. Therefore, we apply the same random labeling process to the calibration data and train a shadow model accordingly. We designed two methods:

1. **Shadow model**. A shadow model replicates the behavior of forget data $\mathcal{D}_f$ throughout the unlearning process. A shadow model is a two-step approach: (1) it firstly trains a shadow original model $\theta'_o$ using train data $\mathcal{D}_{train}$ and clean calibration data $\mathcal{D}_c$ with the same epoch number as the original model $\theta_o$; (2) subsequently, we finetune the $\theta'_o$ using the random labeled calibration data.

2. **Semi-shadow model**. The semi-shadow model only adopts the second step in the shadow model. It finetunes the original model $\theta_o$ with random-labeled calibration data.

The results are presented in Figure 6, where (b)-(e) present the results of the semi-shadow model with different epochs and (f) illustrates the shadow model's result. Under the semi-shadow model, as the number of epochs increases, the distribution of calibration data gradually moves to the right until it becomes consistent with the distribution of forget data. It also shows that the shadow model demonstrates the best ability to handle distribution shifts compared to the semi-shadow model. However, this comes at the cost of higher computational overhead. Overall, the semi-shadow model offers a balanced trade-off between handling distribution shifts effectively and maintaining lower computational costs.

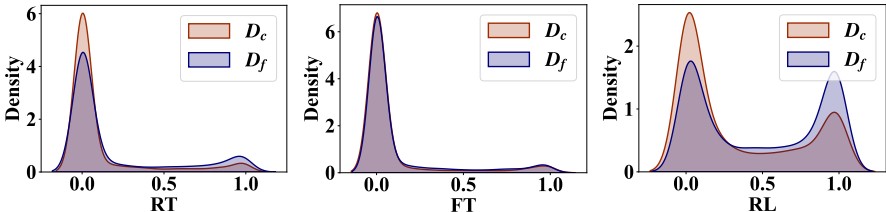

*Figure 8.* Non-conformity score density of calibration data $\mathcal{D}_c$ and forget data $\mathcal{D}_f$ **with our unlearning framework** in CIFAR-10 with ResNet-18 under 10% random data forgetting scenario. Our unlearning framework shifts the distribution of the forget data to the right, demonstrating improved forgetting quality.

*Table 7.* Performance of our unlearning framework CPU with more baselines integrated. We show the performance on **CIFAR-10** with **ResNet-18** in 10% **random data forgetting**. For evaluation criterion ❶, performance differences compared to the RT method are highlighted with (•). For clarity in observing criterion ❷, the sign ↑ represents greater is better, while ↓ denotes ideally small. $\lambda = 0$ represents the baseline without our framework applied. It shows our CPU significantly improves the forgetting quality, not only across our metric CR but also across UA, while preserving stable predictive performance.

| Methods | $\lambda = 0$ | | | | | | $\lambda = 0.2$ | | | | | |
| | UA ↑ | RA ↑ | TA ↑ | MIA ↓ | $CR_{\mathcal{D}_f}$ ↓ | $CR_{\mathcal{D}_{test}}$ ↑ | UA ↑ | RA ↑ | TA ↑ | MIA ↓ | $CR_{\mathcal{D}_f}$ ↓ | $CR_{\mathcal{D}_{test}}$ ↑ |
|---|---|---|---|---|---|---|---|---|---|---|---|---|
| CPU-RT | 8.6%(0.0) | 99.7%(0.0) | 91.8%(0.0) | 86.9%(0.0) | 0.864(0.000) | 0.879(0.000) | 10.8%(2.2) | 98.3%(1.4) | 91.0%(0.8) | 86.2%(0.7) | 0.788(0.076) | 0.824(0.055) |
| CPU-Salun | 3.7%(4.9) | 98.9%(0.8) | 91.8%(0.0)) | 57.6%(29.3) | 0.872(0.008) | 0.832(0.047) | 4.7%(3.9) | 98.8%(0.9) | 91.4%(0.4) | 57.4%(29.5) | 0.872(0.008) | 0.833(0.046) |
| CPU-SFRon | 4.8%(3.8) | 97.4%(2.3) | 91.4%(0.4) | 91.6%(4.7) | 0.889(0.025) | 0.834(0.045) | 5.5%(3.1) | 98.7%(1.0) | 91.3%(0.5) | 91.0%(4.1) | 0.857(0.007) | 0.831(0.048) |

| Methods | $\lambda = 0.5$ | | | | | | $\lambda = 1.0$ | | | | | |
| | UA ↑ | RA ↑ | TA ↑ | MIA ↓ | $CR_{\mathcal{D}_f}$ ↓ | $CR_{\mathcal{D}_{test}}$ ↑ | UA ↑ | RA ↑ | TA ↑ | MIA ↓ | $CR_{\mathcal{D}_f}$ ↓ | $CR_{\mathcal{D}_{test}}$ ↑ |
|---|---|---|---|---|---|---|---|---|---|---|---|---|
| CPU-RT | 14.0%(5.4) | 97.8%(1.9) | 90.4%(0.4) | 85.8%(1.1) | 0.763(0.101) | 0.825(0.054) | 17.7%(9.1) | 96.8%(2.9) | 90.5%(1.3) | 84.1%(2.8) | 0.719(0.145) | 0.820(0.059) |
| CPU-Salun | 6.8%(1.8) | 98.6%(1.1) | 90.9%(0.9) | 57.3%(29.6) | 0.819(0.045) | 0.822(0.057) | 8.6%(0.0) | 98.5%(1.2) | 91.4%(0.4) | 57.0%(29.9) | 0.724(0.140) | 0.821(0.058) |
| CPU-SFRon | 7.7%(0.9) | 98.5%(1.2) | 90.8%(1.0) | 89.1%(2.2) | 0.810(0.054) | 0.824(0.055) | 8.9%(0.3) | 98.5%(1.2) | 90.8%(1.0) | 86.4%(0.5) | 0.810(0.054) | 0.835(0.044) |

## E. Performance of Our Unlearning Framework CPU

### E.1. Unlearning Performance

Table 19 presents the performance of our unlearning framework, including $\alpha \in [0.05, 0.1, 0.15, 0.2]$. We explored the impact of varying $\lambda$ within the range $[0, 0.2, 0.5, 0.1]$, where $\lambda = 0$ serves as the baseline without applying our framework, which can be found in Tables 13 and 16. The results reveal a clear trend: as $\lambda$ increases, the UA improves significantly across all methods, accompanied by a substantial reduction in $CR_{\mathcal{D}_f}$. Interestingly, the RA, TA, and $CR\mathcal{D}test$ metrics remain relatively stable. These results underscore the effectiveness of our unlearning framework in achieving substantial improvements in forgetting quality while preserving the stability of the model's predictive performance.

Furthermore, we conduct an ablation study and analyze the impact of using our unlearning framework. As illustrated in Figures 7 and 8, we compare the density distributions of non-conformity scores for calibration data $\mathcal{D}_c$ and forget data $\mathcal{D}_f$ under the RT, FT, and RL unlearning methods. We set $\lambda$ to 1. Clearly, a higher non-conformity score for $\mathcal{D}_f$ indicates that it is less likely to be included in the conformal prediction set, reflecting more effective forgetting.

Comparing Figures 7 and 8, after applying our unlearning framework, we observe a significant rightward shift in the non-conformity score distribution of forget data, which is a promising signal according to evaluation criterion ❷. Furthermore, the FT distribution in Figure 8 exhibits substantial overlap with the calibration data, nearly matching the distribution observed in RT. Based on evaluation criterion ❶, since calibration data represents unseen examples, the similarity between forget data and calibration data distributions provides strong evidence of effective forgetting. Overall, the results evaluated on both evaluation criteria ❶ and ❷ consistently confirm the efficacy of our framework in enhancing forgetting quality.

### E.2. CPU Performance Integrated by More Unlearning Methods

In Table 7, we further report the CPU performance on existing unlearning methods, Salun and SFRon, denoted as CPU-Salun and CPU-SFRon. The results show that with a larger $\lambda$, CPU can more aggressively enhance the forgetting effect while maintaining model utility.

*Table 8.* Performance of baselines with and without our unlearning framework CPU. We show the performance on **Oxford-Pets** with **Swin-T** in **10% random data forgetting**. For evaluation criterion ❶, performance differences compared to the RT method are highlighted with (•). For clarity in observing criterion ❷, the sign ↑ represents greater is better, while ↓ denotes ideally small. $\lambda = 0$ represents the baseline without our framework applied. It shows our CPU significantly improves the forgetting quality, not only across our metric CR but also across UA, while preserving stable predictive performance.

| Methods | UA ↑ | RA ↑ | TA ↑ | MIA ↓ | $CR_{\mathcal{D}_f}$ ↓ | $CR_{\mathcal{D}_{test}}$ ↑ |
|---|---|---|---|---|---|---|
| RT | 6.5%(0.0) | 98.1%(0.0) | 93.1%(0.0) | 86.2%(0.0) | 0.822(0.000) | 0.857(0.000) |
| FT | 3.0%(3.5) | 99.5%(1.4) | 93.2%(0.1) | 94.0%(7.8) | 0.950(0.128) | 0.903(0.046) |
| CPU-FT | 7.3%(0.8) | 99.6%(1.5) | 92.2%(0.9) | 89.7%(3.5) | 0.902(0.079) | 0.897(0.041) |
| NegGrad+ | 3.3%(3.3) | 99.8%(1.7) | 93.2%(0.1) | 92.2%(6.0) | 0.919(0.097) | 0.877(0.020) |
| CPU-NegGrad+ | 7.3%(0.8) | 99.5%(1.4) | 92.8%(0.2) | 88.8%(2.6) | 0.897(0.075) | 0.878(0.021) |
| Salun | 4.9%(1.6) | 99.5%(1.4) | 92.3%(0.8) | 87.4%(1.2) | 0.876(0.054) | 0.864(0.008) |
| CPU-Salun | 6.8%(0.3) | 99.6%(1.5) | 92.5%(0.5) | 85.7%(0.5) | 0.846(0.024) | 0.850(0.007) |

### E.3. CPU Performance under the Oxford-Pet Dataset with the Swin-T Architecture

We evaluate the effectiveness of CPU on the Oxford-Pets dataset using the Swin-T architecture in 10% random data forgetting scenario. CPU consistently outperforms FT, NegGrad+, and Salun, demonstrating its strong generalizability across different datasets and model architectures.

*Table 9.* Training time comparison with and without our CPU loss under 10% random data forgetting scenario. The training time is reported in minutes.

| Methods | w/o CPU | w/ CPU |
|---|---|---|
| **CIFAR-10 with ResNet18** | | |
| RT | 70.1 | 72.1 |
| FT | 6.3 | 6.8 |
| RL | 6.3 | 6.8 |
| **Tiny ImageNet with ViT** | | |
| RT | 60.75 | 62.85 |
| FT | 20.2 | 22.1 |
| RL | 21.3 | 23.4 |

### E.4. Time Comparison

We compare the training time with and without our unlearning calibration process on CIFAR-10 and Tiny ImageNet under 10% random data forgetting scenario. As shown in Table 9, the training times with and without CPU support differ only marginally, confirming that our CPU loss computation introduces negligible overhead.

## F. Additional Experimental Results

### F.1. Other Conformal Prediction Methods

While we adopt vanilla split-conformal as the default due to its simplicity and reproducibility, our framework is not limited to this variant. Here, we report the results using other conformal prediction methods, LAC (Sadinle et al., 2019), EntmaxScore (Campos et al., 2025), and ASP (Romano et al., 2020b) on CIFAR-10 with ResNet18 under 10% random data forgetting.

As shown in the Table 10, the CR results of LAC and EntmaxScore are similar to those obtained using SCP in Table 3. This suggests that the results are stable under conformal prediction methods that offer formal coverage guarantees. However, APS produces different CR values compared to LAC, SCP, and EntmaxScore. This discrepancy is expected and is due to the inherent characteristics of APS, which make it unsuitable for evaluating unlearning metrics. APS generally produces loose prediction sets and is highly sensitive to noisy probability estimates in the lower-ranked classes (Angelopoulos et al., 2020), which introduces randomness in the ordering of unlikely classes and leads to unreliable set construction. Our findings indicate that not all conformal prediction methods are inherently suitable for evaluating forgetting quality. And the reliability of such evaluation depends critically on whether the resulting prediction sets faithfully capture the model's uncertainty.

*Table 10.* CR performance with different conformal prediction methods. The performance gap relative to the RT method is represented in (•).

| Methods | LAC | | EntmaxScore | | APS | |
|---|---|---|---|---|---|---|
| | CR($\mathcal{D}_f$)↓ | CR($\mathcal{D}_t$)↑ | CR($\mathcal{D}_f$)↓ | CR($\mathcal{D}_t$)↑ | CR($\mathcal{D}_f$)↓ | CR($\mathcal{D}_t$)↑ |
| RT | 0.862(0.000) | 0.876(0.000) | 0.863(0.000) | 0.877(0.000) | 0.805(0.000) | 0.836(0.000) |
| FT | 0.901(0.039) | 0.846(0.030) | 0.901(0.038) | 0.848(0.029) | 0.808(0.004) | 0.784(0.052) |
| RL | 0.676(0.186) | 0.752(0.124) | 0.883(0.020) | 0.838(0.039) | 0.573(0.232) | 0.670(0.166) |
| GA | 0.995(0.133) | 0.931(0.055) | 0.995(0.132) | 0.930(0.054) | 0.985(0.180) | 0.875(0.038) |
| Teacher | 0.988(0.127) | 0.915(0.039) | 0.987(0.125) | 0.917(0.040) | 0.511(0.293) | 0.536(0.300) |
| SSD | 0.995(0.133) | 0.933(0.057) | 0.994(0.131) | 0.930(0.054) | 0.985(0.181) | 0.876(0.039) |
| NegGrad+ | 0.865(0.003) | 0.863(0.013) | 0.869(0.006) | 0.870(0.006) | 0.860(0.056) | 0.856(0.020) |
| SCRUB | 0.940(0.078) | 0.867(0.009) | 0.908(0.045) | 0.857(0.020) | 0.925(0.120) | 0.872(0.036) |
| LoTUS | 0.972(0.110) | 0.908(0.032) | 0.986(0.123) | 0.908(0.031) | 0.984(0.179) | 0.918(0.082) |
| MUNBa | 0.992(0.130) | 0.952(0.076) | 0.994(0.131) | 0.962(0.085) | 0.989(0.184) | 0.952(0.116) |
| Salun | 0.881(0.019) | 0.839(0.037) | 0.878(0.015) | 0.839(0.038) | 0.407(0.398) | 0.430(0.407) |
| SFRon | 0.893(0.031) | 0.838(0.038) | 0.893(0.030) | 0.838(0.039) | 0.815(0.010) | 0.769(0.067) |

Overall, conformal prediction serves as a component within our uncertainty quantification-based evaluation framework. The simplest and most straightforward conformal prediction methods, especially SCP, are often the most suitable tools. While many recent conformal prediction variants improve upon different issues, e.g., by modifying the nonconformity scores or explicitly penalizing low-probability classes (Angelopoulos et al., 2020; Huang et al., 2023), these techniques often distort the nonconformity values across some classes. Since our goal is to use conformal prediction as a tool for designing fair metrics and evaluating forgetting quality, we intentionally avoid such modifications. Introducing these more complex methods could result in additional noise, thereby compromising the fairness and interpretability of our evaluation.

### F.2. Distribution of Non-conformity Score

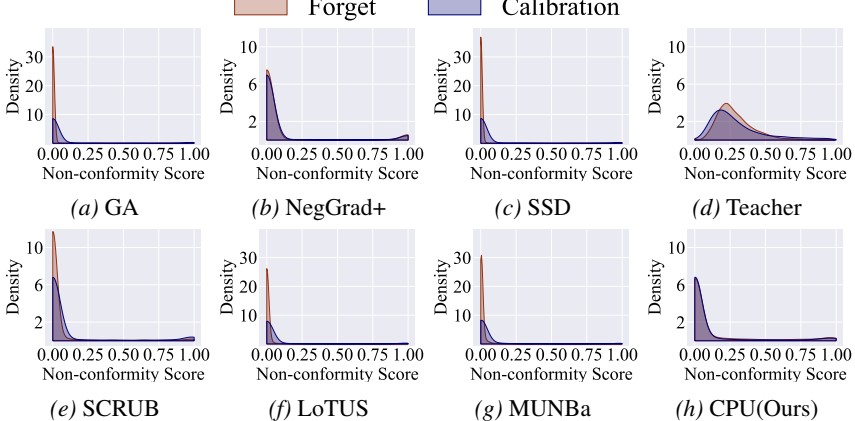

*Figure 9.* Distribution of non-conformity scores. Our CPU-FT closely matches the RT baseline, achieving alignment between the distributions of forgetting data and calibration data.

In this section, we provide the non-conformity score distributions for additional unlearning baselines, including GA, NegGrad+, SSD, Teacher, SCRUB, LoTUS, and MUNBa, which were not shown in the main paper due to space constraints. These results complement Figure 4 by enabling a more comprehensive comparison of how different unlearning methods align the forget data with the calibration (unseen) data. The corresponding distributions are presented in Figure 9.

## G. Other Forgetting Scenario

**Worst-case Forgetting scenario.** Random data forgetting may affect unlearning models differently, introducing variance and bias that make it a relatively weak evaluation setting. To more rigorously assess the effectiveness of our proposed metrics, we further evaluate them using worst-case forget sets (Fan et al., 2024a). As shown in Table 11, the results are consistent with our previous analysis.

*Table 11.* Unlearning performance on **CIFAR-10** with **ResNet-18** in **10% worst-case data forgetting** scenario. The results are reported in the format a±b, where a is the mean and b is the standard deviation from 3 independent trials. The performance gap relative to the RT method is represented in (•).

| Methods | Existing Metrics | | | Coverage | | Set Size | | CR | |
|---|---|---|---|---|---|---|---|---|---|
| | UA↑ | RA↑ | TA↑ | $\mathcal{D}_f$↓ | $\mathcal{D}_{test}$↑ | $\mathcal{D}_f$↑ | $\mathcal{D}_{test}$↓ | $\mathcal{D}_f$↓ | $\mathcal{D}_{test}$↑ |
| RT | 0.0%(0.0) | 99.2%(0.0) | 91.5%(0.0) | 1.000(0.000) | 0.948(0.000) | 1.000(0.000) | 1.116(0.000) | 1.000(0.000) | 0.850(0.000) |
| FT | 0.0%(0.0) | 99.8%(0.6) | 94.1%(2.6) | 1.000(0.000) | 0.938(0.010) | 1.000(0.000) | 0.992(0.124) | 1.000(0.000) | 0.945(0.095) |
| RL | 21.3%(21.3) | 97.4%(1.7) | 88.5%(3.0) | 0.976(0.024) | 0.955(0.007) | 6.753(5.753) | 2.192(1.076) | 0.146(0.854) | 0.441(0.409) |
| GA | 0.3%(0.3) | 96.9%(2.2) | 91.3%(0.2) | 0.999(0.001) | 0.954(0.006) | 1.029(0.029) | 1.179(0.063) | 0.971(0.029) | 0.810(0.040) |
| Teacher | 15.8%(15.8) | 97.9%(1.2) | 90.6%(0.9) | 0.850(0.150) | 0.946(0.002) | 1.177(0.177) | 1.249(0.133) | 0.745(0.255) | 0.760(0.090) |
| SSD | 0.0%(0.0) | 99.7%(0.5) | 94.0%(2.6) | 1.000(0.000) | 0.954(0.006) | 1.000(0.000) | 1.037(0.079) | 1.000(0.000) | 0.920(0.070) |
| NegGrad+ | 0.0%(0.0) | 99.8%(0.6) | 94.2%(2.7) | 1.000(0.000) | 0.947(0.001) | 1.000(0.000) | 1.012(0.104) | 1.000(0.000) | 0.936(0.086) |
| SCRUB | 86.9%(86.9) | 15.5%(83.7) | 15.7%(75.8) | 0.904(0.096) | 0.947(0.002) | 8.921(7.921) | 8.777(7.661) | 0.102(0.898) | 0.108(0.742) |
| LoTUS | 34.8%(34.8) | 36.5%(62.7) | 40.6%(50.9) | 0.991(0.009) | 0.950(0.002) | 4.650(3.650) | 5.519(4.403) | 0.214(0.786) | 0.173(0.677) |
| MUNBa | 0.0%(0.0) | 99.8%(0.6) | 94.2%(2.7) | 1.000(0.000) | 0.951(0.003) | 1.000(0.000) | 1.022(0.094) | 1.000(0.000) | 0.931(0.081) |
| Salun | 13.0%(13.0) | 97.6%(1.6) | 90.0%(1.5) | 0.962(0.038) | 0.947(0.001) | 3.991(2.991) | 1.567(0.451) | 0.246(0.754) | 0.606(0.244) |
| SFRon | 0.0%(0.0) | 99.5%(0.3) | 93.8%(2.4) | 1.000(0.000) | 0.956(0.008) | 1.000(0.000) | 1.053(0.063) | 1.000(0.000) | 0.908(0.058) |

*Table 12.* Unlearning performance on **CIFAR-20** with **ResNet18** in **subclass-wise forgetting** scenario.

| Methods | Existing Metrics | | | | Coverage | | | Set Size | | | CR | | |
|---|---|---|---|---|---|---|---|---|---|---|---|---|---|
| | UA↑ | $UA_{sf}$↑ | RA↑ | TA↑ | $\mathcal{D}_f$↓ | $\mathcal{D}_{tf}$↓ | $\mathcal{D}_{tr}$↑ | $\mathcal{D}_f$↑ | $\mathcal{D}_{tf}$↑ | $\mathcal{D}_{tr}$↓ | $\mathcal{D}_f$↓ | $\mathcal{D}_{tf}$↓ | $\mathcal{D}_{tr}$↑ |
| RT | 97.6%(0.0) | 94.0%(0.0) | 99.9%(0.0) | 84.5%(0.0) | 1.000(0.000) | 1.000(0.000) | 0.953(0.000) | 20.000(0.000) | 20.000(0.000) | 1.713(0.000) | 0.050(0.000) | 0.050(0.000) | 0.556(0.000) |
| FT | 70.9%(26.7) | 74.7%(19.3) | 95.7%(4.1) | 76.0%(8.6) | 0.994(0.006) | 0.987(0.013) | 0.952(0.001) | 17.637(2.363) | 16.893(3.107) | 3.091(1.377) | 0.057(0.007) | 0.059(0.009) | 0.312(0.245) |
| RL | 99.5%(1.9) | 94.7%(0.7) | 98.2%(1.7) | 76.7%(7.9) | 0.931(0.069) | 1.000(0.000) | 0.955(0.001) | 18.807(1.193) | 19.527(0.473) | 3.300(1.586) | 0.050(0.000) | 0.051(0.001) | 0.289(0.267) |
| GA | 40.7%(56.9) | 60.7%(33.3) | 99.0%(0.8) | 82.2%(2.3) | 0.999(0.001) | 0.993(0.007) | 0.954(0.001) | 18.305(1.695) | 17.553(2.447) | 2.409(0.695) | 0.055(0.005) | 0.057(0.007) | 0.397(0.159) |
| Teacher | 90.6%(7.0) | 97.3%(3.3) | 98.6%(1.3) | 81.3%(3.2) | 0.989(0.011) | 0.933(0.067) | 0.948(0.005) | 19.871(0.129) | 18.840(1.160) | 2.747(1.034) | 0.050(0.000) | 0.050(0.000) | 0.350(0.206) |
| SSD | 73.6%(24.0) | 80.0%(14.0) | 99.8%(0.0) | 84.5%(0.1) | 0.997(0.003) | 0.980(0.020) | 0.955(0.001) | 19.206(0.794) | 17.740(2.260) | 2.407(0.694) | 0.052(0.002) | 0.055(0.005) | 0.423(0.133) |
| NegGrad+ | 98.9%(1.3) | 100.0%(6.0) | 97.0%(2.8) | 80.9%(3.7) | 1.000(0.000) | 1.000(0.000) | 0.950(0.003) | 20.000(0.000) | 20.000(0.000) | 2.761(1.048) | 0.050(0.000) | 0.050(0.000) | 0.372(0.184) |
| SCRUB | 97.9%(0.3) | 95.3%(1.3) | 33.4%(66.5) | 33.6%(50.9) | 0.999(0.001) | 1.000(0.000) | 0.952(0.001) | 19.999(0.001) | 20.000(0.000) | 12.970(11.256) | 0.050(0.000) | 0.050(0.000) | 0.079(0.477) |
| LoTUS | 7.4%(90.2) | 29.3%(64.7) | 79.6%(20.2) | 70.8%(13.7) | 0.999(0.001) | 0.947(0.053) | 0.953(0.000) | 19.968(0.032) | 18.647(1.353) | 4.598(2.885) | 0.050(0.000) | 0.051(0.001) | 0.209(0.347) |
| MUNBa | 1.8%(95.8) | 24.7%(69.3) | 100.0%(0.1) | 85.5%(1.0) | 1.000(0.000) | 0.960(0.040) | 0.948(0.005) | 4.491(15.509) | 6.113(13.887) | 1.683(0.031) | 0.315(0.265) | 0.214(0.164) | 0.565(0.009) |
| Salun | 99.9%(2.3) | 96.0%(2.0) | 98.8%(1.0) | 78.9%(5.6) | 0.955(0.045) | 0.993(0.007) | 0.951(0.002) | 19.235(0.765) | 19.707(0.293) | 2.737(1.023) | 0.050(0.000) | 0.050(0.000) | 0.348(0.208) |
| SFRon | 99.9%(2.3) | 100.0%(6.0) | 91.9%(7.9) | 79.7%(4.9) | 1.000(0.000) | 1.000(0.000) | 0.951(0.003) | 20.000(0.000) | 20.000(0.000) | 2.587(0.874) | 0.050(0.000) | 0.050(0.000) | 0.370(0.186) |

**Subclass-wise Forgetting Scenario.** To further verify our metrics in other forgetting scenarios, we report subclass-wise forgetting results on CIFAR-20 (derived from CIFAR-100) using ResNet-18, following the setting proposed in (Foster et al., 2024). As shown in the Table 12, the findings align well with our prior analysis.

# H. Large Language Models Usage Statement

We used a large language model (LLM) to polish the language and improve the clarity of the paper. All content, including the core ideas, methodology, and experimental results, was originally created by the authors. The LLM was used exclusively as an editing tool to enhance readability and grammatical correctness, without generating any substantive or technical content.

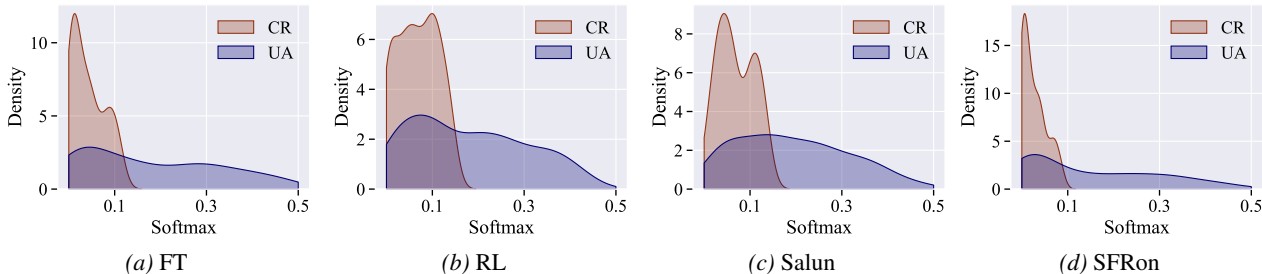

*Figure 10.* **Softmax distribution** in **10% random data forgetting** scenario. We analyze the softmax distributions of true labels for data identified as truly forgotten by CR and UA, respectively. The distribution curves are fitted using KDE for clearer visualization. The results illustrate the softmax distributions of CR consistently closer to 0 when compared to UA, providing strong evidence that CR is better than UA in accurately capturing and measuring 'real forgetting'.

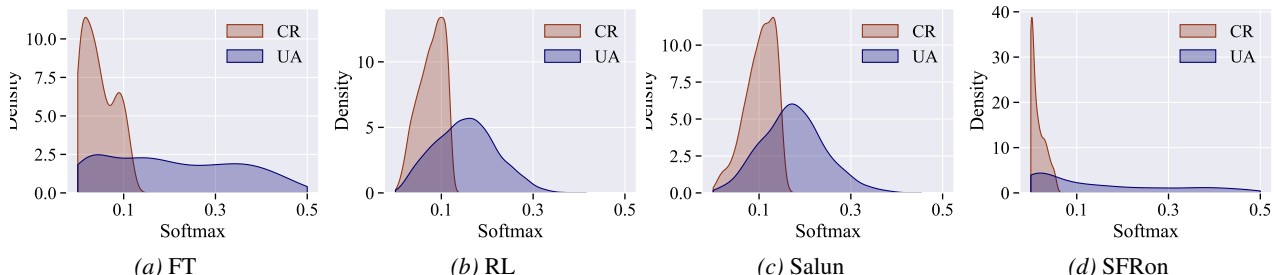

*Figure 11.* **Softmax distribution** in **50% random data forgetting** scenario.

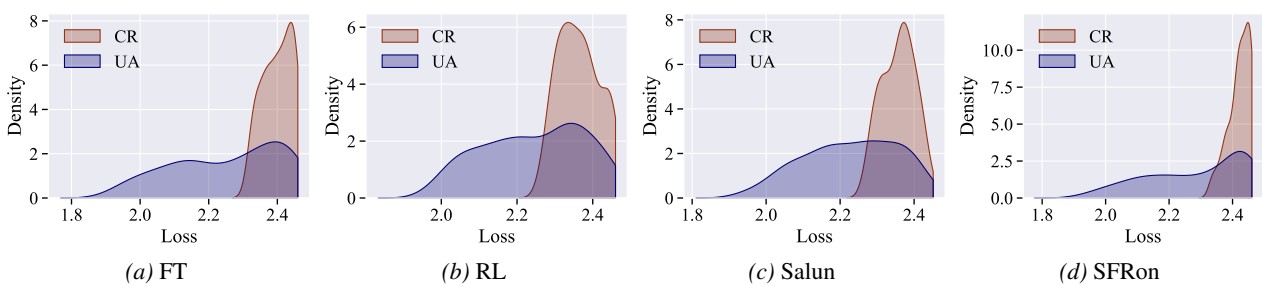

*Figure 12.* **Loss distribution** in **10% random data forgetting** scenario. We analyze the cross-entropy loss distributions of true labels for data identified as truly forgotten by CR and UA, respectively. Forgotten data identified by CR consistently show higher cross-entropy loss than UA. Higher loss indicates better forgetting quality, which further validates that CR better captures 'real forgetting'.

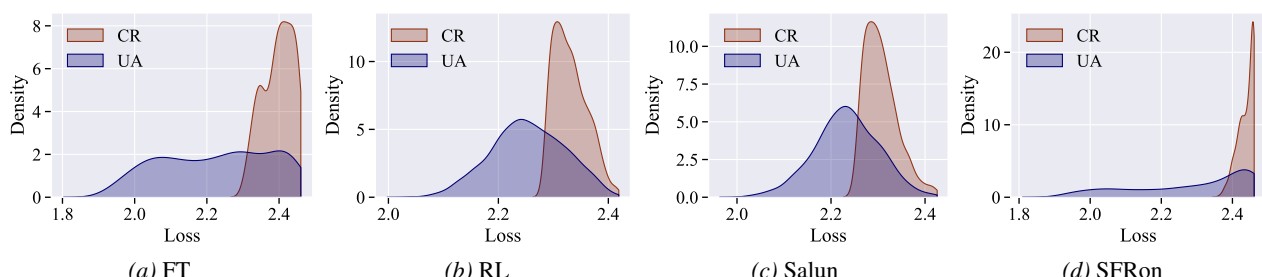

*Figure 13.* **Loss distribution** in **50% random data forgetting** scenario.

*Table 13.* Unlearning performance of 9 unlearning methods on **CIFAR-10** with **ResNet-18** in $10\%$ **random data forgetting** scenario. The results are reported in the format a±b, where a is the mean and b is the standard deviation from 3 independent trials. The performance gap relative to the RT method is represented in (•).

| Methods | $\alpha$ | Coverage | | Set Size | | CR | | $\hat{q}$ |
|---|---|---|---|---|---|---|---|---|
| | | $\mathcal{D}_f \downarrow$ | $\mathcal{D}_{test} \uparrow$ | $\mathcal{D}_f \uparrow$ | $\mathcal{D}_{test} \downarrow$ | $\mathcal{D}_f \downarrow$ | $\mathcal{D}_{test} \uparrow$ | |
| RT
**UA**8.6%, **RA**99.7%, **TA**91.8% | 0.05 | $0.941_{\pm 0.002}(0.000)$ | $0.944_{\pm 0.005}(0.000)$ | $1.089_{\pm 0.002}(0.000)$ | $1.074_{\pm 0.011}(0.000)$ | $0.864_{\pm 0.004}(0.000)$ | $0.879_{\pm 0.004}(0.000)$ | $0.883_{\pm 0.007}$ |
| | 0.1 | $0.881_{\pm 0.000}(0.000)$ | $0.895_{\pm 0.010}(0.000)$ | $0.934_{\pm 0.004}(0.000)$ | $0.947_{\pm 0.008}(0.000)$ | $0.943_{\pm 0.011}(0.000)$ | $0.945_{\pm 0.001}(0.000)$ | $0.192_{\pm 0.001}$ |
| | 0.15 | $0.820_{\pm 0.002}(0.000)$ | $0.839_{\pm 0.008}(0.000)$ | $0.841_{\pm 0.009}(0.000)$ | $0.867_{\pm 0.009}(0.000)$ | $0.975_{\pm 0.001}(0.000)$ | $0.968_{\pm 0.003}(0.000)$ | $0.015_{\pm 0.011}$ |
| | 0.2 | $0.780_{\pm 0.007}(0.000)$ | $0.808_{\pm 0.004}(0.000)$ | $0.789_{\pm 0.002}(0.000)$ | $0.824_{\pm 0.009}(0.000)$ | $0.988_{\pm 0.006}(0.000)$ | $0.981_{\pm 0.007}(0.000)$ | $0.003_{\pm 0.002}$ |
| FT
**UA**3.8%, **RA**98.1%, **TA**91.6% | 0.05 | $0.994_{\pm 0.001}(0.053)$ | $0.951_{\pm 0.004}(0.007)$ | $1.008_{\pm 0.003}(0.081)$ | $1.026_{\pm 0.008}(0.048)$ | $0.986_{\pm 0.003}(0.122)$ | $0.927_{\pm 0.004}(0.048)$ | $0.721_{\pm 0.045}$ |
| | 0.1 | $0.968_{\pm 0.001}(0.087)$ | $0.899_{\pm 0.005}(0.004)$ | $0.969_{\pm 0.001}(0.035)$ | $0.924_{\pm 0.008}(0.023)$ | $0.998_{\pm 0.001}(0.055)$ | $0.972_{\pm 0.003}(0.027)$ | $0.079_{\pm 0.013}$ |
| | 0.15 | $0.915_{\pm 0.003}(0.095)$ | $0.848_{\pm 0.002}(0.009)$ | $0.916_{\pm 0.003}(0.075)$ | $0.860_{\pm 0.001}(0.007)$ | $1.000_{\pm 0.000}(0.025)$ | $0.986_{\pm 0.002}(0.018)$ | $0.008_{\pm 0.000}$ |
| | 0.2 | $0.861_{\pm 0.010}(0.081)$ | $0.806_{\pm 0.008}(0.002)$ | $0.861_{\pm 0.010}(0.072)$ | $0.811_{\pm 0.009}(0.013)$ | $1.000_{\pm 0.000}(0.012)$ | $0.993_{\pm 0.001}(0.012)$ | $0.002_{\pm 0.000}$ |
| RL
**UA**7.6%, **RA**97.4%, **TA**90.6% | 0.05 | $0.970_{\pm 0.006}(0.029)$ | $0.949_{\pm 0.005}(0.005)$ | $1.242_{\pm 0.151}(0.153)$ | $1.197_{\pm 0.098}(0.123)$ | $0.788_{\pm 0.089}(0.076)$ | $0.796_{\pm 0.061}(0.083)$ | $0.877_{\pm 0.057}$ |
| | 0.1 | $0.913_{\pm 0.010}(0.032)$ | $0.897_{\pm 0.007}(0.002)$ | $0.975_{\pm 0.028}(0.041)$ | $0.980_{\pm 0.025}(0.033)$ | $0.936_{\pm 0.022}(0.007)$ | $0.916_{\pm 0.019}(0.029)$ | $0.572_{\pm 0.059}$ |
| | 0.15 | $0.825_{\pm 0.006}(0.005)$ | $0.843_{\pm 0.009}(0.004)$ | $0.854_{\pm 0.010}(0.013)$ | $0.888_{\pm 0.017}(0.021)$ | $0.966_{\pm 0.006}(0.009)$ | $0.949_{\pm 0.009}(0.019)$ | $0.329_{\pm 0.021}$ |
| | 0.2 | $0.755_{\pm 0.021}(0.025)$ | $0.798_{\pm 0.005}(0.010)$ | $0.774_{\pm 0.020}(0.015)$ | $0.832_{\pm 0.009}(0.008)$ | $0.976_{\pm 0.002}(0.012)$ | $0.959_{\pm 0.005}(0.022)$ | $0.234_{\pm 0.028}$ |
| GA
**UA**0.6%, **RA**99.5%, **TA**94.1% | 0.05 | $0.994_{\pm 0.003}(0.053)$ | $0.945_{\pm 0.008}(0.001)$ | $1.002_{\pm 0.010}(0.087)$ | $1.009_{\pm 0.010}(0.065)$ | $0.994_{\pm 0.016}(0.130)$ | $0.936_{\pm 0.011}(0.057)$ | $0.621_{\pm 0.015}$ |
| | 0.1 | $0.990_{\pm 0.005}(0.109)$ | $0.905_{\pm 0.019}(0.010)$ | $0.990_{\pm 0.014}(0.056)$ | $0.928_{\pm 0.005}(0.019)$ | $0.998_{\pm 0.002}(0.055)$ | $0.973_{\pm 0.012}(0.028)$ | $0.062_{\pm 0.016}$ |
| | 0.15 | $0.969_{\pm 0.012}(0.149)$ | $0.848_{\pm 0.004}(0.009)$ | $0.969_{\pm 0.014}(0.128)$ | $0.858_{\pm 0.019}(0.009)$ | $1.000_{\pm 0.014}(0.025)$ | $0.986_{\pm 0.008}(0.018)$ | $0.006_{\pm 0.009}$ |
| | 0.2 | $0.925_{\pm 0.012}(0.145)$ | $0.805_{\pm 0.022}(0.003)$ | $0.924_{\pm 0.007}(0.135)$ | $0.811_{\pm 0.013}(0.013)$ | $0.998_{\pm 0.013}(0.010)$ | $0.992_{\pm 0.012}(0.011)$ | $0.003_{\pm 0.005}$ |
| Teacher
**UA**0.8%, **RA**99.4%, **TA**93.5% | 0.05 | $0.991_{\pm 0.022}(0.050)$ | $0.941_{\pm 0.001}(0.003)$ | $1.003_{\pm 0.012}(0.086)$ | $1.021_{\pm 0.009}(0.053)$ | $0.993_{\pm 0.021}(0.129)$ | $0.922_{\pm 0.015}(0.043)$ | $0.744_{\pm 0.015}$ |
| | 0.1 | $0.967_{\pm 0.000}(0.086)$ | $0.898_{\pm 0.007}(0.003)$ | $0.963_{\pm 0.007}(0.029)$ | $0.929_{\pm 0.018}(0.018)$ | $0.998_{\pm 0.000}(0.055)$ | $0.969_{\pm 0.013}(0.024)$ | $0.591_{\pm 0.005}$ |
| | 0.15 | $0.913_{\pm 0.006}(0.093)$ | $0.845_{\pm 0.007}(0.006)$ | $0.912_{\pm 0.014}(0.071)$ | $0.859_{\pm 0.005}(0.008)$ | $0.996_{\pm 0.018}(0.021)$ | $0.983_{\pm 0.015}(0.015)$ | $0.481_{\pm 0.009}$ |
| | 0.2 | $0.865_{\pm 0.009}(0.085)$ | $0.806_{\pm 0.021}(0.002)$ | $0.866_{\pm 0.009}(0.077)$ | $0.816_{\pm 0.012}(0.008)$ | $0.998_{\pm 0.008}(0.010)$ | $0.988_{\pm 0.016}(0.007)$ | $0.426_{\pm 0.007}$ |
| SSD
**UA**0.5%, **RA**99.5%, **TA**94.2% | 0.05 | $0.996_{\pm 0.004}(0.055)$ | $0.945_{\pm 0.002}(0.001)$ | $0.999_{\pm 0.019}(0.090)$ | $1.008_{\pm 0.011}(0.066)$ | $0.994_{\pm 0.006}(0.130)$ | $0.936_{\pm 0.014}(0.057)$ | $0.622_{\pm 0.019}$ |
| | 0.1 | $0.987_{\pm 0.003}(0.106)$ | $0.902_{\pm 0.010}(0.007)$ | $0.990_{\pm 0.003}(0.056)$ | $0.926_{\pm 0.017}(0.021)$ | $0.998_{\pm 0.020}(0.055)$ | $0.973_{\pm 0.002}(0.028)$ | $0.063_{\pm 0.022}$ |
| | 0.15 | $0.967_{\pm 0.016}(0.147)$ | $0.849_{\pm 0.009}(0.010)$ | $0.965_{\pm 0.000}(0.124)$ | $0.862_{\pm 0.012}(0.005)$ | $1.002_{\pm 0.019}(0.027)$ | $0.990_{\pm 0.002}(0.022)$ | $0.007_{\pm 0.007}$ |
| | 0.2 | $0.922_{\pm 0.006}(0.142)$ | $0.803_{\pm 0.000}(0.005)$ | $0.923_{\pm 0.009}(0.134)$ | $0.811_{\pm 0.005}(0.013)$ | $1.002_{\pm 0.020}(0.014)$ | $0.992_{\pm 0.009}(0.011)$ | $0.001_{\pm 0.005}$ |
| NegGrad+
**UA**8.7%, **RA**98.8%, **TA**92.2% | 0.05 | $0.934_{\pm 0.007}(0.007)$ | $0.948_{\pm 0.007}(0.004)$ | $1.068_{\pm 0.017}(0.021)$ | $1.086_{\pm 0.022}(0.012)$ | $0.875_{\pm 0.008}(0.011)$ | $0.873_{\pm 0.011}(0.006)$ | $0.989_{\pm 0.013}$ |
| | 0.1 | $0.895_{\pm 0.004}(0.014)$ | $0.898_{\pm 0.008}(0.003)$ | $0.964_{\pm 0.008}(0.030)$ | $0.950_{\pm 0.013}(0.003)$ | $0.928_{\pm 0.005}(0.015)$ | $0.946_{\pm 0.005}(0.001)$ | $0.044_{\pm 0.041}$ |
| | 0.15 | $0.851_{\pm 0.013}(0.031)$ | $0.851_{\pm 0.016}(0.012)$ | $0.896_{\pm 0.016}(0.055)$ | $0.876_{\pm 0.019}(0.009)$ | $0.950_{\pm 0.003}(0.025)$ | $0.971_{\pm 0.003}(0.003)$ | $0.000_{\pm 0.000}$ |
| | 0.2 | $0.800_{\pm 0.006}(0.020)$ | $0.799_{\pm 0.001}(0.009)$ | $0.832_{\pm 0.006}(0.043)$ | $0.813_{\pm 0.001}(0.011)$ | $0.961_{\pm 0.002}(0.027)$ | $0.983_{\pm 0.001}(0.002)$ | $0.000_{\pm 0.000}$ |
| SCRUB
**UA**3.4%, **RA**98.0%, **TA**92.0% | 0.05 | $0.984_{\pm 0.004}(0.043)$ | $0.946_{\pm 0.006}(0.002)$ | $1.060_{\pm 0.027}(0.029)$ | $1.085_{\pm 0.034}(0.011)$ | $0.952_{\pm 0.016}(0.088)$ | $0.873_{\pm 0.022}(0.006)$ | $0.893_{\pm 0.056}$ |
| | 0.1 | $0.952_{\pm 0.003}(0.071)$ | $0.904_{\pm 0.004}(0.009)$ | $0.974_{\pm 0.002}(0.040)$ | $0.963_{\pm 0.003}(0.016)$ | $0.988_{\pm 0.001}(0.045)$ | $0.939_{\pm 0.002}(0.006)$ | $0.279_{\pm 0.005}$ |
| | 0.15 | $0.903_{\pm 0.022}(0.083)$ | $0.855_{\pm 0.020}(0.016)$ | $0.911_{\pm 0.026}(0.070)$ | $0.885_{\pm 0.028}(0.018)$ | $0.996_{\pm 0.002}(0.021)$ | $0.966_{\pm 0.009}(0.002)$ | $0.041_{\pm 0.029}$ |
| | 0.2 | $0.848_{\pm 0.009}(0.068)$ | $0.807_{\pm 0.008}(0.001)$ | $0.851_{\pm 0.010}(0.062)$ | $0.824_{\pm 0.009}(0.000)$ | $0.998_{\pm 0.000}(0.010)$ | $0.979_{\pm 0.003}(0.002)$ | $0.006_{\pm 0.001}$ |
| LoTUS
**UA**0.7%, **RA**99.2%, **TA**93.5% | 0.05 | $0.996_{\pm 0.001}(0.055)$ | $0.955_{\pm 0.006}(0.011)$ | $1.014_{\pm 0.004}(0.075)$ | $1.063_{\pm 0.021}(0.011)$ | $0.980_{\pm 0.006}(0.116)$ | $0.899_{\pm 0.012}(0.020)$ | $0.841_{\pm 0.059}$ |
| | 0.1 | $0.986_{\pm 0.005}(0.105)$ | $0.908_{\pm 0.007}(0.013)$ | $0.989_{\pm 0.006}(0.055)$ | $0.946_{\pm 0.011}(0.001)$ | $0.997_{\pm 0.001}(0.054)$ | $0.960_{\pm 0.004}(0.015)$ | $0.213_{\pm 0.079}$ |
| | 0.15 | $0.943_{\pm 0.018}(0.123)$ | $0.849_{\pm 0.018}(0.010)$ | $0.944_{\pm 0.018}(0.103)$ | $0.864_{\pm 0.020}(0.003)$ | $0.999_{\pm 0.000}(0.024)$ | $0.982_{\pm 0.002}(0.014)$ | $0.023_{\pm 0.015}$ |
| | 0.2 | $0.886_{\pm 0.009}(0.106)$ | $0.797_{\pm 0.015}(0.011)$ | $0.886_{\pm 0.009}(0.097)$ | $0.806_{\pm 0.016}(0.018)$ | $1.000_{\pm 0.000}(0.012)$ | $0.990_{\pm 0.001}(0.009)$ | $0.005_{\pm 0.002}$ |
| MUNBa
**UA**0.6%, **RA**99.8%, **TA**94.4% | 0.05 | $0.050_{\pm 0.050}(0.891)$ | $0.998_{\pm 0.000}(0.054)$ | $0.994_{\pm 0.000}(0.095)$ | $0.944_{\pm 0.001}(0.130)$ | $0.998_{\pm 0.000}(0.134)$ | $0.996_{\pm 0.001}(0.117)$ | $0.952_{\pm 0.001}$ |
| | 0.1 | $0.100_{\pm 0.100}(0.781)$ | $0.998_{\pm 0.000}(0.103)$ | $0.997_{\pm 0.002}(0.063)$ | $0.943_{\pm 0.001}(0.004)$ | $0.989_{\pm 0.001}(0.046)$ | $0.978_{\pm 0.002}(0.033)$ | $0.893_{\pm 0.005}$ |
| | 0.15 | $0.150_{\pm 0.150}(0.670)$ | $0.998_{\pm 0.000}(0.159)$ | $0.995_{\pm 0.001}(0.154)$ | $0.944_{\pm 0.000}(0.077)$ | $0.973_{\pm 0.012}(0.002)$ | $0.944_{\pm 0.023}(0.024)$ | $0.859_{\pm 0.024}$ |
| | 0.2 | $0.200_{\pm 0.200}(0.580)$ | $0.998_{\pm 0.000}(0.190)$ | $0.996_{\pm 0.000}(0.207)$ | $0.943_{\pm 0.001}(0.119)$ | $0.920_{\pm 0.014}(0.068)$ | $0.878_{\pm 0.012}(0.103)$ | $0.793_{\pm 0.016}$ |
| Salun
**UA**3.7%, **RA**98.9%, **TA**91.8% | 0.05 | $0.987_{\pm 0.002}(0.046)$ | $0.950_{\pm 0.001}(0.006)$ | $1.132_{\pm 0.007}(0.043)$ | $1.143_{\pm 0.002}(0.069)$ | $0.872_{\pm 0.006}(0.008)$ | $0.832_{\pm 0.003}(0.047)$ | $0.867_{\pm 0.001}$ |
| | 0.1 | $0.936_{\pm 0.010}(0.055)$ | $0.896_{\pm 0.008}(0.001)$ | $0.956_{\pm 0.012}(0.022)$ | $0.954_{\pm 0.011}(0.007)$ | $0.979_{\pm 0.003}(0.036)$ | $0.939_{\pm 0.003}(0.006)$ | $0.489_{\pm 0.029}$ |
| | 0.15 | $0.871_{\pm 0.005}(0.051)$ | $0.849_{\pm 0.008}(0.010)$ | $0.881_{\pm 0.006}(0.040)$ | $0.886_{\pm 0.010}(0.019)$ | $0.989_{\pm 0.002}(0.014)$ | $0.958_{\pm 0.002}(0.010)$ | $0.314_{\pm 0.020}$ |
| | 0.2 | $0.788_{\pm 0.010}(0.008)$ | $0.794_{\pm 0.001}(0.014)$ | $0.794_{\pm 0.010}(0.005)$ | $0.821_{\pm 0.004}(0.003)$ | $0.992_{\pm 0.001}(0.004)$ | $0.966_{\pm 0.003}(0.015)$ | $0.221_{\pm 0.005}$ |
| SFRon
**UA**4.8%, **RA**97.4%, **TA**91.4% | 0.05 | $0.977_{\pm 0.003}(0.036)$ | $0.953_{\pm 0.004}(0.009)$ | $1.100_{\pm 0.023}(0.011)$ | $1.143_{\pm 0.021}(0.069)$ | $0.889_{\pm 0.015}(0.025)$ | $0.834_{\pm 0.012}(0.045)$ | $0.926_{\pm 0.018}$ |
| | 0.1 | $0.945_{\pm 0.004}(0.064)$ | $0.905_{\pm 0.005}(0.010)$ | $0.986_{\pm 0.005}(0.052)$ | $0.977_{\pm 0.008}(0.030)$ | $0.958_{\pm 0.001}(0.015)$ | $0.927_{\pm 0.003}(0.018)$ | $0.435_{\pm 0.043}$ |
| | 0.15 | $0.895_{\pm 0.002}(0.075)$ | $0.847_{\pm 0.002}(0.008)$ | $0.912_{\pm 0.004}(0.071)$ | $0.879_{\pm 0.001}(0.012)$ | $0.982_{\pm 0.002}(0.007)$ | $0.963_{\pm 0.003}(0.005)$ | $0.082_{\pm 0.007}$ |
| | 0.2 | $0.857_{\pm 0.008}(0.077)$ | $0.808_{\pm 0.002}(0.000)$ | $0.868_{\pm 0.007}(0.079)$ | $0.826_{\pm 0.005}(0.002)$ | $0.988_{\pm 0.002}(0.000)$ | $0.978_{\pm 0.004}(0.003)$ | $0.025_{\pm 0.005}$ |

*Table 14.* Unlearning performance of 9 unlearning methods on **CIFAR-10** with **ResNet18** in **50% random data forgetting** scenario.

| Methods | $\alpha$ | Coverage | | Set Size | | CR | | $\hat{q}$ |
|---|---|---|---|---|---|---|---|---|
| | | $\mathcal{D}_f \downarrow$ | $\mathcal{D}_{test} \uparrow$ | $\mathcal{D}_f \uparrow$ | $\mathcal{D}_{test} \downarrow$ | $\mathcal{D}_f \downarrow$ | $\mathcal{D}_{test} \uparrow$ | |
| RT **UA**11.0%, **RA**99.8%, **TA**89.2% | 0.05 | $0.955_{\pm 0.004}(0.000)$ | $0.947_{\pm 0.005}(0.000)$ | $1.287_{\pm 0.001}(0.000)$ | $1.214_{\pm 0.010}(0.000)$ | $0.742_{\pm 0.005}(0.000)$ | $0.780_{\pm 0.006}(0.000)$ | $0.984_{\pm 0.002}$ |
| | 0.1 | $0.898_{\pm 0.011}(0.000)$ | $0.904_{\pm 0.010}(0.000)$ | $1.023_{\pm 0.005}(0.000)$ | $1.021_{\pm 0.003}(0.000)$ | $0.878_{\pm 0.003}(0.000)$ | $0.886_{\pm 0.003}(0.000)$ | $0.650_{\pm 0.004}$ |
| | 0.15 | $0.833_{\pm 0.007}(0.000)$ | $0.847_{\pm 0.005}(0.000)$ | $0.883_{\pm 0.002}(0.000)$ | $0.906_{\pm 0.003}(0.000)$ | $0.943_{\pm 0.010}(0.000)$ | $0.934_{\pm 0.005}(0.000)$ | $0.090_{\pm 0.004}$ |
| | 0.2 | $0.782_{\pm 0.005}(0.000)$ | $0.814_{\pm 0.004}(0.000)$ | $0.812_{\pm 0.010}(0.000)$ | $0.850_{\pm 0.009}(0.000)$ | $0.964_{\pm 0.005}(0.000)$ | $0.958_{\pm 0.003}(0.000)$ | $0.018_{\pm 0.006}$ |
| FT **UA**2.6%, **RA**99.1%, **TA**91.8% | 0.05 | $0.996_{\pm 0.000}(0.041)$ | $0.952_{\pm 0.002}(0.005)$ | $1.007_{\pm 0.000}(0.280)$ | $1.029_{\pm 0.004}(0.185)$ | $0.989_{\pm 0.001}(0.247)$ | $0.925_{\pm 0.002}(0.145)$ | $0.738_{\pm 0.014}$ |
| | 0.1 | $0.975_{\pm 0.006}(0.077)$ | $0.896_{\pm 0.013}(0.008)$ | $0.976_{\pm 0.006}(0.047)$ | $0.921_{\pm 0.017}(0.100)$ | $0.999_{\pm 0.000}(0.121)$ | $0.972_{\pm 0.004}(0.086)$ | $0.081_{\pm 0.033}$ |
| | 0.15 | $0.936_{\pm 0.004}(0.103)$ | $0.854_{\pm 0.004}(0.007)$ | $0.936_{\pm 0.004}(0.053)$ | $0.867_{\pm 0.006}(0.039)$ | $1.000_{\pm 0.000}(0.057)$ | $0.985_{\pm 0.002}(0.051)$ | $0.011_{\pm 0.002}$ |
| | 0.2 | $0.859_{\pm 0.010}(0.077)$ | $0.790_{\pm 0.010}(0.024)$ | $0.859_{\pm 0.010}(0.047)$ | $0.795_{\pm 0.011}(0.055)$ | $1.000_{\pm 0.000}(0.036)$ | $0.993_{\pm 0.001}(0.035)$ | $0.001_{\pm 0.000}$ |
| RL **UA**10.5%, **RA**93.9%, **TA**85.8% | 0.05 | $0.976_{\pm 0.001}(0.022)$ | $0.949_{\pm 0.002}(0.002)$ | $1.973_{\pm 0.396}(0.686)$ | $1.971_{\pm 0.406}(0.757)$ | $0.508_{\pm 0.100}(0.234)$ | $0.495_{\pm 0.098}(0.285)$ | $0.899_{\pm 0.012}$ |
| | 0.1 | $0.942_{\pm 0.011}(0.043)$ | $0.907_{\pm 0.009}(0.003)$ | $1.227_{\pm 0.103}(0.204)$ | $1.235_{\pm 0.107}(0.214)$ | $0.771_{\pm 0.064}(0.107)$ | $0.738_{\pm 0.064}(0.147)$ | $0.837_{\pm 0.016}$ |
| | 0.15 | $0.891_{\pm 0.013}(0.058)$ | $0.856_{\pm 0.012}(0.009)$ | $1.009_{\pm 0.047}(0.125)$ | $1.011_{\pm 0.045}(0.105)$ | $0.884_{\pm 0.039}(0.059)$ | $0.847_{\pm 0.037}(0.087)$ | $0.770_{\pm 0.022}$ |
| | 0.2 | $0.834_{\pm 0.003}(0.051)$ | $0.799_{\pm 0.005}(0.016)$ | $0.897_{\pm 0.026}(0.086)$ | $0.893_{\pm 0.025}(0.043)$ | $0.929_{\pm 0.024}(0.034)$ | $0.895_{\pm 0.022}(0.063)$ | $0.713_{\pm 0.028}$ |
| GA **UA**0.6%, **RA**99.5%, **TA**94.3% | 0.05 | $0.996_{\pm 0.000}(0.041)$ | $0.945_{\pm 0.008}(0.002)$ | $1.003_{\pm 0.007}(0.284)$ | $1.005_{\pm 0.007}(0.209)$ | $1.050_{\pm 0.007}(0.308)$ | $0.945_{\pm 0.007}(0.165)$ | $0.616_{\pm 0.008}$ |
| | 0.1 | $0.985_{\pm 0.006}(0.087)$ | $0.902_{\pm 0.009}(0.002)$ | $0.989_{\pm 0.006}(0.034)$ | $0.926_{\pm 0.006}(0.095)$ | $1.095_{\pm 0.004}(0.217)$ | $0.916_{\pm 0.006}(0.030)$ | $0.057_{\pm 0.005}$ |
| | 0.15 | $0.966_{\pm 0.006}(0.133)$ | $0.848_{\pm 0.007}(0.001)$ | $0.966_{\pm 0.002}(0.083)$ | $0.857_{\pm 0.009}(0.049)$ | $1.141_{\pm 0.001}(0.198)$ | $0.879_{\pm 0.006}(0.055)$ | $0.005_{\pm 0.007}$ |
| | 0.2 | $0.929_{\pm 0.004}(0.147)$ | $0.809_{\pm 0.007}(0.005)$ | $0.932_{\pm 0.000}(0.120)$ | $0.817_{\pm 0.005}(0.033)$ | $1.150_{\pm 0.002}(0.186)$ | $0.871_{\pm 0.001}(0.087)$ | $0.001_{\pm 0.007}$ |
| Teacher **UA**1.6%, **RA**98.3%, **TA**91.7% | 0.05 | $0.985_{\pm 0.015}(0.030)$ | $0.944_{\pm 0.018}(0.003)$ | $1.066_{\pm 0.003}(0.221)$ | $1.143_{\pm 0.012}(0.071)$ | $0.923_{\pm 0.010}(0.181)$ | $0.823_{\pm 0.017}(0.043)$ | $0.857_{\pm 0.013}$ |
| | 0.1 | $0.949_{\pm 0.012}(0.051)$ | $0.909_{\pm 0.016}(0.005)$ | $0.970_{\pm 0.006}(0.053)$ | $0.986_{\pm 0.014}(0.035)$ | $0.980_{\pm 0.001}(0.102)$ | $0.918_{\pm 0.009}(0.032)$ | $0.834_{\pm 0.005}$ |
| | 0.15 | $0.885_{\pm 0.010}(0.052)$ | $0.849_{\pm 0.018}(0.002)$ | $0.894_{\pm 0.017}(0.011)$ | $0.893_{\pm 0.010}(0.013)$ | $0.992_{\pm 0.002}(0.049)$ | $0.950_{\pm 0.013}(0.016)$ | $0.813_{\pm 0.013}$ |
| | 0.2 | $0.818_{\pm 0.014}(0.036)$ | $0.798_{\pm 0.014}(0.016)$ | $0.823_{\pm 0.009}(0.011)$ | $0.826_{\pm 0.002}(0.024)$ | $0.997_{\pm 0.015}(0.033)$ | $0.971_{\pm 0.007}(0.013)$ | $0.793_{\pm 0.012}$ |
| SSD **UA**0.5%, **RA**99.5%, **TA**94.3% | 0.05 | $0.993_{\pm 0.005}(0.038)$ | $0.944_{\pm 0.011}(0.003)$ | $0.999_{\pm 0.007}(0.288)$ | $1.001_{\pm 0.009}(0.213)$ | $0.995_{\pm 0.009}(0.253)$ | $0.941_{\pm 0.013}(0.161)$ | $0.585_{\pm 0.014}$ |
| | 0.1 | $0.991_{\pm 0.015}(0.093)$ | $0.904_{\pm 0.014}(0.000)$ | $0.991_{\pm 0.001}(0.032)$ | $0.929_{\pm 0.011}(0.092)$ | $1.000_{\pm 0.011}(0.122)$ | $0.975_{\pm 0.010}(0.089)$ | $0.060_{\pm 0.011}$ |
| | 0.15 | $0.964_{\pm 0.016}(0.131)$ | $0.850_{\pm 0.011}(0.003)$ | $0.967_{\pm 0.009}(0.084)$ | $0.860_{\pm 0.014}(0.046)$ | $1.000_{\pm 0.001}(0.057)$ | $0.988_{\pm 0.003}(0.054)$ | $0.005_{\pm 0.010}$ |
| | 0.2 | $0.930_{\pm 0.018}(0.148)$ | $0.807_{\pm 0.002}(0.007)$ | $0.929_{\pm 0.002}(0.117)$ | $0.814_{\pm 0.017}(0.036)$ | $1.000_{\pm 0.003}(0.036)$ | $0.992_{\pm 0.001}(0.034)$ | $0.002_{\pm 0.005}$ |
| NegGrad+ **UA**2.8%, **RA**99.6%, **TA**92.9% | 0.05 | $0.986_{\pm 0.000}(0.031)$ | $0.949_{\pm 0.001}(0.001)$ | $1.039_{\pm 0.008}(0.248)$ | $1.062_{\pm 0.011}(0.152)$ | $0.949_{\pm 0.008}(0.207)$ | $0.893_{\pm 0.008}(0.113)$ | $0.855_{\pm 0.028}$ |
| | 0.1 | $0.951_{\pm 0.005}(0.053)$ | $0.903_{\pm 0.004}(0.001)$ | $0.964_{\pm 0.008}(0.059)$ | $0.944_{\pm 0.010}(0.076)$ | $0.987_{\pm 0.003}(0.109)$ | $0.956_{\pm 0.007}(0.070)$ | $0.177_{\pm 0.055}$ |
| | 0.15 | $0.889_{\pm 0.004}(0.056)$ | $0.845_{\pm 0.003}(0.002)$ | $0.892_{\pm 0.004}(0.009)$ | $0.861_{\pm 0.003}(0.045)$ | $0.996_{\pm 0.000}(0.053)$ | $0.981_{\pm 0.001}(0.047)$ | $0.012_{\pm 0.002}$ |
| | 0.2 | $0.825_{\pm 0.003}(0.043)$ | $0.796_{\pm 0.004}(0.018)$ | $0.827_{\pm 0.003}(0.015)$ | $0.805_{\pm 0.004}(0.045)$ | $0.999_{\pm 0.000}(0.035)$ | $0.989_{\pm 0.000}(0.032)$ | $0.002_{\pm 0.000}$ |
| SCRUB **UA**2.2%, **RA**98.8%, **TA**92.6% | 0.05 | $0.990_{\pm 0.004}(0.036)$ | $0.951_{\pm 0.015}(0.003)$ | $1.057_{\pm 0.057}(0.230)$ | $1.094_{\pm 0.082}(0.120)$ | $0.960_{\pm 0.034}(0.219)$ | $0.871_{\pm 0.051}(0.091)$ | $0.869_{\pm 0.127}$ |
| | 0.1 | $0.958_{\pm 0.004}(0.059)$ | $0.893_{\pm 0.007}(0.011)$ | $0.967_{\pm 0.008}(0.057)$ | $0.937_{\pm 0.016}(0.084)$ | $0.995_{\pm 0.003}(0.118)$ | $0.953_{\pm 0.011}(0.068)$ | $0.142_{\pm 0.074}$ |
| | 0.15 | $0.915_{\pm 0.008}(0.083)$ | $0.850_{\pm 0.006}(0.003)$ | $0.918_{\pm 0.007}(0.035)$ | $0.873_{\pm 0.002}(0.033)$ | $0.998_{\pm 0.001}(0.055)$ | $0.973_{\pm 0.005}(0.039)$ | $0.019_{\pm 0.002}$ |
| | 0.2 | $0.864_{\pm 0.025}(0.082)$ | $0.803_{\pm 0.020}(0.011)$ | $0.866_{\pm 0.025}(0.054)$ | $0.818_{\pm 0.019}(0.033)$ | $1.000_{\pm 0.000}(0.036)$ | $0.983_{\pm 0.002}(0.025)$ | $0.004_{\pm 0.002}$ |
| LoTUS **UA**0.7%, **RA**99.2%, **TA**93.8% | 0.05 | $0.996_{\pm 0.001}(0.041)$ | $0.956_{\pm 0.007}(0.009)$ | $1.012_{\pm 0.004}(0.276)$ | $1.055_{\pm 0.021}(0.159)$ | $0.982_{\pm 0.005}(0.240)$ | $0.907_{\pm 0.011}(0.127)$ | $0.813_{\pm 0.062}$ |
| | 0.1 | $0.982_{\pm 0.004}(0.084)$ | $0.902_{\pm 0.011}(0.002)$ | $0.984_{\pm 0.005}(0.039)$ | $0.933_{\pm 0.016}(0.088)$ | $0.998_{\pm 0.001}(0.120)$ | $0.966_{\pm 0.005}(0.081)$ | $0.163_{\pm 0.050}$ |
| | 0.15 | $0.950_{\pm 0.012}(0.117)$ | $0.853_{\pm 0.014}(0.007)$ | $0.951_{\pm 0.012}(0.068)$ | $0.869_{\pm 0.019}(0.037)$ | $0.999_{\pm 0.000}(0.056)$ | $0.982_{\pm 0.005}(0.048)$ | $0.034_{\pm 0.014}$ |
| | 0.2 | $0.889_{\pm 0.017}(0.107)$ | $0.800_{\pm 0.013}(0.014)$ | $0.890_{\pm 0.017}(0.078)$ | $0.808_{\pm 0.012}(0.042)$ | $1.000_{\pm 0.000}(0.036)$ | $0.990_{\pm 0.001}(0.032)$ | $0.009_{\pm 0.002}$ |
| MUNBa **UA**1.6%, **RA**98.9%, **TA**92.2% | 0.05 | $0.993_{\pm 0.001}(0.038)$ | $0.950_{\pm 0.006}(0.003)$ | $1.044_{\pm 0.004}(0.243)$ | $1.095_{\pm 0.014}(0.119)$ | $0.962_{\pm 0.001}(0.220)$ | $0.867_{\pm 0.006}(0.087)$ | $0.802_{\pm 0.006}$ |
| | 0.1 | $0.965_{\pm 0.005}(0.066)$ | $0.892_{\pm 0.002}(0.012)$ | $0.972_{\pm 0.005}(0.051)$ | $0.943_{\pm 0.000}(0.078)$ | $0.994_{\pm 0.001}(0.117)$ | $0.946_{\pm 0.001}(0.061)$ | $0.637_{\pm 0.044}$ |
| | 0.15 | $0.917_{\pm 0.010}(0.084)$ | $0.838_{\pm 0.013}(0.009)$ | $0.920_{\pm 0.010}(0.036)$ | $0.865_{\pm 0.019}(0.041)$ | $0.998_{\pm 0.001}(0.055)$ | $0.969_{\pm 0.007}(0.034)$ | $0.521_{\pm 0.100}$ |
| | 0.2 | $0.869_{\pm 0.023}(0.087)$ | $0.796_{\pm 0.017}(0.019)$ | $0.871_{\pm 0.024}(0.059)$ | $0.814_{\pm 0.017}(0.036)$ | $0.999_{\pm 0.000}(0.035)$ | $0.977_{\pm 0.001}(0.020)$ | $0.458_{\pm 0.068}$ |
| SFRon **UA**4.0%, **RA**97.3%, **TA**91.6% | 0.05 | $0.984_{\pm 0.004}(0.029)$ | $0.944_{\pm 0.010}(0.003)$ | $1.097_{\pm 0.010}(0.190)$ | $1.138_{\pm 0.021}(0.077)$ | $0.897_{\pm 0.004}(0.155)$ | $0.830_{\pm 0.009}(0.050)$ | $0.926_{\pm 0.019}$ |
| | 0.1 | $0.955_{\pm 0.007}(0.057)$ | $0.899_{\pm 0.008}(0.005)$ | $0.988_{\pm 0.011}(0.035)$ | $0.982_{\pm 0.016}(0.039)$ | $0.966_{\pm 0.006}(0.089)$ | $0.916_{\pm 0.011}(0.030)$ | $0.456_{\pm 0.099}$ |
| | 0.15 | $0.909_{\pm 0.010}(0.076)$ | $0.851_{\pm 0.010}(0.004)$ | $0.922_{\pm 0.009}(0.039)$ | $0.892_{\pm 0.009}(0.014)$ | $0.986_{\pm 0.002}(0.042)$ | $0.954_{\pm 0.005}(0.020)$ | $0.104_{\pm 0.032}$ |
| | 0.2 | $0.853_{\pm 0.009}(0.071)$ | $0.799_{\pm 0.014}(0.015)$ | $0.860_{\pm 0.009}(0.048)$ | $0.821_{\pm 0.015}(0.030)$ | $0.992_{\pm 0.003}(0.028)$ | $0.973_{\pm 0.003}(0.016)$ | $0.024_{\pm 0.001}$ |
| Salun **UA**4.3%, **RA**97.7%, **TA**89.4% | 0.05 | $0.988_{\pm 0.001}(0.034)$ | $0.951_{\pm 0.003}(0.004)$ | $1.314_{\pm 0.113}(0.027)$ | $1.381_{\pm 0.121}(0.167)$ | $0.756_{\pm 0.064}(0.014)$ | $0.692_{\pm 0.058}(0.088)$ | $0.871_{\pm 0.013}$ |
| | 0.1 | $0.956_{\pm 0.003}(0.058)$ | $0.897_{\pm 0.005}(0.007)$ | $1.015_{\pm 0.003}(0.008)$ | $1.021_{\pm 0.001}(0.001)$ | $0.941_{\pm 0.006}(0.064)$ | $0.878_{\pm 0.004}(0.007)$ | $0.776_{\pm 0.002}$ |
| | 0.15 | $0.910_{\pm 0.005}(0.078)$ | $0.847_{\pm 0.006}(0.000)$ | $0.937_{\pm 0.009}(0.054)$ | $0.916_{\pm 0.008}(0.010)$ | $0.972_{\pm 0.004}(0.029)$ | $0.924_{\pm 0.003}(0.010)$ | $0.714_{\pm 0.010}$ |
| | 0.2 | $0.856_{\pm 0.008}(0.074)$ | $0.796_{\pm 0.010}(0.019)$ | $0.872_{\pm 0.008}(0.060)$ | $0.844_{\pm 0.008}(0.006)$ | $0.982_{\pm 0.003}(0.019)$ | $0.943_{\pm 0.004}(0.015)$ | $0.669_{\pm 0.008}$ |

*Table 15.* Unlearning performance of 9 unlearning methods on **CIFAR-10** with **ResNet18** in **class-wise forgetting** scenario.

| Methods | $\alpha$ | Coverage $\mathcal{D}_f\downarrow$ | Coverage $\mathcal{D}_{tf}\downarrow$ | Coverage $\mathcal{D}_{tr}\uparrow$ | Set Size $\mathcal{D}_f\uparrow$ | Set Size $\mathcal{D}_{tf}\uparrow$ | Set Size $\mathcal{D}_{tr}\downarrow$ | CR $\mathcal{D}_f\downarrow$ | CR $\mathcal{D}_{tf}\downarrow$ | CR $\mathcal{D}_{tr}\uparrow$ | $\hat{q}_f$ | $\hat{q}_{test}$ |
|---|---|---|---|---|---|---|---|---|---|---|---|---|
| **RT** UA100%, UA$_{tf}$100%, RA99.9%, TA92.4% | 0.05 | $1.000_{\pm0.001}(0.000)$ | $1.000_{\pm0.001}(0.000)$ | $0.964_{\pm0.008}(0.000)$ | $10.000_{\pm0.000}(0.000)$ | $10.000_{\pm0.000}(0.000)$ | $1.148_{\pm0.013}(0.000)$ | $0.100_{\pm0.000}(0.000)$ | $0.100_{\pm0.000}(0.000)$ | $0.840_{\pm0.002}(0.000)$ | $1.000_{\pm0.000}$ | $0.982_{\pm0.003}$ |
| | 0.1 | $1.000_{\pm0.000}(0.000)$ | $1.000_{\pm0.001}(0.000)$ | $0.882_{\pm0.011}(0.000)$ | $10.000_{\pm0.000}(0.000)$ | $10.000_{\pm0.000}(0.000)$ | $0.922_{\pm0.009}(0.000)$ | $0.100_{\pm0.000}(0.000)$ | $0.100_{\pm0.001}(0.000)$ | $0.956_{\pm0.007}(0.000)$ | $1.000_{\pm0.001}$ | $0.080_{\pm0.003}$ |
| | 0.15 | $1.000_{\pm0.000}(0.000)$ | $1.000_{\pm0.000}(0.000)$ | $0.856_{\pm0.012}(0.000)$ | $10.000_{\pm0.000}(0.000)$ | $10.000_{\pm0.000}(0.000)$ | $0.882_{\pm0.007}(0.000)$ | $0.100_{\pm0.001}(0.000)$ | $0.100_{\pm0.001}(0.000)$ | $0.970_{\pm0.004}(0.000)$ | $1.000_{\pm0.000}$ | $0.018_{\pm0.010}$ |
| | 0.2 | $1.000_{\pm0.000}(0.000)$ | $1.000_{\pm0.000}(0.000)$ | $0.814_{\pm0.010}(0.000)$ | $10.000_{\pm0.000}(0.000)$ | $10.000_{\pm0.000}(0.000)$ | $0.830_{\pm0.001}(0.000)$ | $0.100_{\pm0.001}(0.000)$ | $0.100_{\pm0.001}(0.000)$ | $0.981_{\pm0.002}(0.000)$ | $1.000_{\pm0.000}$ | $0.003_{\pm0.001}$ |
| **FT** UA100%, UA$_{tf}$100%, RA96.7%, TA90.8% | 0.05 | $0.994_{\pm0.003}(0.006)$ | $0.962_{\pm0.022}(0.038)$ | $0.944_{\pm0.011}(0.020)$ | $9.854_{\pm0.127}(0.146)$ | $9.403_{\pm0.501}(0.597)$ | $1.045_{\pm0.040}(0.103)$ | $0.101_{\pm0.001}(0.001)$ | $0.102_{\pm0.003}(0.002)$ | $0.904_{\pm0.028}(0.065)$ | $1.000_{\pm0.000}$ | $0.731_{\pm0.166}$ |
| | 0.1 | $0.969_{\pm0.011}(0.031)$ | $0.882_{\pm0.020}(0.118)$ | $0.908_{\pm0.010}(0.026)$ | $9.495_{\pm0.255}(0.505)$ | $8.528_{\pm0.571}(1.472)$ | $0.956_{\pm0.006}(0.034)$ | $0.102_{\pm0.002}(0.002)$ | $0.104_{\pm0.005}(0.004)$ | $0.950_{\pm0.007}(0.006)$ | $1.000_{\pm0.000}$ | $0.314_{\pm0.010}$ |
| | 0.15 | $0.951_{\pm0.014}(0.049)$ | $0.840_{\pm0.011}(0.160)$ | $0.851_{\pm0.031}(0.005)$ | $9.265_{\pm0.279}(0.735)$ | $8.131_{\pm0.523}(1.869)$ | $0.872_{\pm0.039}(0.010)$ | $0.103_{\pm0.003}(0.003)$ | $0.103_{\pm0.007}(0.003)$ | $0.976_{\pm0.009}(0.006)$ | $1.000_{\pm0.000}$ | $0.073_{\pm0.054}$ |
| | 0.2 | $0.942_{\pm0.014}(0.058)$ | $0.818_{\pm0.072}(0.182)$ | $0.838_{\pm0.016}(0.023)$ | $9.163_{\pm0.245}(0.837)$ | $7.934_{\pm0.533}(2.066)$ | $0.854_{\pm0.019}(0.024)$ | $0.103_{\pm0.003}(0.003)$ | $0.103_{\pm0.010}(0.003)$ | $0.981_{\pm0.005}(0.000)$ | $1.000_{\pm0.000}$ | $0.039_{\pm0.017}$ |
| **RL** UA100%, UA$_{tf}$100%, RA98.0%, TA92.7% | 0.05 | $0.995_{\pm0.002}(0.005)$ | $0.954_{\pm0.009}(0.046)$ | $0.959_{\pm0.015}(0.005)$ | $9.993_{\pm0.003}(0.007)$ | $9.900_{\pm0.011}(0.100)$ | $1.170_{\pm0.155}(0.022)$ | $0.100_{\pm0.000}(0.000)$ | $0.096_{\pm0.001}(0.004)$ | $0.828_{\pm0.097}(0.012)$ | $1.000_{\pm0.000}$ | $0.870_{\pm0.145}$ |
| | 0.1 | $0.984_{\pm0.003}(0.016)$ | $0.907_{\pm0.015}(0.093)$ | $0.918_{\pm0.021}(0.036)$ | $9.978_{\pm0.004}(0.022)$ | $9.800_{\pm0.019}(0.200)$ | $0.982_{\pm0.036}(0.059)$ | $0.099_{\pm0.000}(0.001)$ | $0.093_{\pm0.002}(0.007)$ | $0.936_{\pm0.022}(0.021)$ | $1.000_{\pm0.000}$ | $0.469_{\pm0.250}$ |
| | 0.15 | $0.961_{\pm0.009}(0.039)$ | $0.859_{\pm0.014}(0.141)$ | $0.870_{\pm0.019}(0.014)$ | $9.950_{\pm0.017}(0.050)$ | $9.700_{\pm0.066}(0.300)$ | $0.904_{\pm0.045}(0.021)$ | $0.097_{\pm0.001}(0.003)$ | $0.089_{\pm0.001}(0.011)$ | $0.964_{\pm0.027}(0.006)$ | $1.000_{\pm0.000}$ | $0.144_{\pm0.163}$ |
| | 0.2 | $0.935_{\pm0.027}(0.065)$ | $0.815_{\pm0.012}(0.185)$ | $0.804_{\pm0.016}(0.010)$ | $9.919_{\pm0.035}(0.081)$ | $9.637_{\pm0.076}(0.363)$ | $0.820_{\pm0.026}(0.010)$ | $0.094_{\pm0.002}(0.006)$ | $0.085_{\pm0.001}(0.015)$ | $0.981_{\pm0.012}(0.000)$ | $0.999_{\pm0.001}$ | $0.014_{\pm0.013}$ |
| **GA** UA84.6%, UA$_{tf}$82.5%, RA96.4%, TA89.6% | 0.05 | $1.000_{\pm0.003}(0.000)$ | $1.000_{\pm0.005}(0.000)$ | $0.948_{\pm0.004}(0.016)$ | $10.000_{\pm0.009}(0.000)$ | $10.000_{\pm0.005}(0.000)$ | $1.204_{\pm0.002}(0.056)$ | $0.100_{\pm0.007}(0.000)$ | $0.100_{\pm0.011}(0.000)$ | $0.787_{\pm0.011}(0.053)$ | $1.000_{\pm0.010}$ | $0.988_{\pm0.000}$ |
| | 0.1 | $1.000_{\pm0.003}(0.000)$ | $1.000_{\pm0.010}(0.000)$ | $0.899_{\pm0.008}(0.017)$ | $10.000_{\pm0.005}(0.000)$ | $10.000_{\pm0.006}(0.000)$ | $1.005_{\pm0.003}(0.083)$ | $0.100_{\pm0.012}(0.000)$ | $0.100_{\pm0.006}(0.000)$ | $0.894_{\pm0.002}(0.062)$ | $1.000_{\pm0.000}$ | $0.562_{\pm0.003}$ |
| | 0.15 | $1.000_{\pm0.006}(0.000)$ | $1.000_{\pm0.001}(0.000)$ | $0.843_{\pm0.011}(0.013)$ | $10.000_{\pm0.005}(0.000)$ | $10.000_{\pm0.006}(0.000)$ | $0.893_{\pm0.010}(0.011)$ | $0.100_{\pm0.004}(0.000)$ | $0.100_{\pm0.008}(0.000)$ | $0.944_{\pm0.007}(0.026)$ | $1.000_{\pm0.001}$ | $0.051_{\pm0.002}$ |
| | 0.2 | $0.828_{\pm0.003}(0.172)$ | $0.782_{\pm0.011}(0.218)$ | $0.838_{\pm0.010}(0.024)$ | $9.550_{\pm0.007}(0.450)$ | $9.366_{\pm0.002}(0.634)$ | $0.884_{\pm0.000}(0.054)$ | $0.087_{\pm0.008}(0.013)$ | $0.084_{\pm0.005}(0.016)$ | $0.948_{\pm0.010}(0.033)$ | $1.000_{\pm0.002}$ | $0.038_{\pm0.003}$ |
| **Teacher** UA90.1%, UA$_{tf}$86.5%, RA99.5%, TA94.0% | 0.05 | $0.994_{\pm0.003}(0.006)$ | $0.959_{\pm0.002}(0.041)$ | $0.939_{\pm0.003}(0.025)$ | $9.877_{\pm0.000}(0.123)$ | $9.502_{\pm0.003}(0.498)$ | $1.000_{\pm0.004}(0.148)$ | $0.101_{\pm0.004}(0.001)$ | $0.101_{\pm0.004}(0.001)$ | $0.939_{\pm0.001}(0.099)$ | $0.955_{\pm0.005}$ | $0.588_{\pm0.004}$ |
| | 0.1 | $0.931_{\pm0.004}(0.069)$ | $0.904_{\pm0.001}(0.096)$ | $0.890_{\pm0.001}(0.008)$ | $9.199_{\pm0.002}(0.801)$ | $8.604_{\pm0.004}(1.396)$ | $0.914_{\pm0.004}(0.008)$ | $0.101_{\pm0.004}(0.001)$ | $0.105_{\pm0.004}(0.005)$ | $0.974_{\pm0.003}(0.018)$ | $0.926_{\pm0.004}$ | $0.116_{\pm0.005}$ |
| | 0.15 | $0.879_{\pm0.004}(0.121)$ | $0.881_{\pm0.001}(0.119)$ | $0.834_{\pm0.001}(0.022)$ | $8.730_{\pm0.002}(1.270)$ | $8.081_{\pm0.001}(1.919)$ | $0.845_{\pm0.005}(0.037)$ | $0.101_{\pm0.004}(0.001)$ | $0.109_{\pm0.002}(0.009)$ | $0.986_{\pm0.004}(0.016)$ | $0.921_{\pm0.001}$ | $0.017_{\pm0.002}$ |
| | 0.2 | $0.809_{\pm0.004}(0.191)$ | $0.841_{\pm0.004}(0.159)$ | $0.816_{\pm0.000}(0.002)$ | $8.141_{\pm0.003}(1.859)$ | $7.525_{\pm0.003}(2.475)$ | $0.824_{\pm0.003}(0.006)$ | $0.099_{\pm0.002}(0.001)$ | $0.112_{\pm0.003}(0.012)$ | $0.990_{\pm0.002}(0.009)$ | $0.916_{\pm0.005}$ | $0.010_{\pm0.003}$ |
| **SSD** UA1.16%, UA$_{tf}$7.75%, RA99.5%, TA94.3% | 0.05 | $0.995_{\pm0.014}(0.005)$ | $0.935_{\pm0.013}(0.065)$ | $0.940_{\pm0.007}(0.024)$ | $1.030_{\pm0.014}(8.970)$ | $1.067_{\pm0.013}(8.933)$ | $0.991_{\pm0.011}(0.157)$ | $0.966_{\pm0.010}(0.866)$ | $0.876_{\pm0.007}(0.776)$ | $0.949_{\pm0.010}(0.109)$ | $0.804_{\pm0.015}$ | $0.447_{\pm0.007}$ |
| | 0.1 | $0.984_{\pm0.021}(0.016)$ | $0.910_{\pm0.009}(0.090)$ | $0.880_{\pm0.001}(0.002)$ | $0.992_{\pm0.011}(9.008)$ | $0.982_{\pm0.005}(9.018)$ | $0.896_{\pm0.003}(0.026)$ | $0.992_{\pm0.003}(0.892)$ | $0.926_{\pm0.017}(0.826)$ | $0.981_{\pm0.012}(0.025)$ | $0.434_{\pm0.007}$ | $0.022_{\pm0.005}$ |
| | 0.15 | $0.960_{\pm0.012}(0.040)$ | $0.876_{\pm0.011}(0.124)$ | $0.847_{\pm0.007}(0.009)$ | $0.962_{\pm0.007}(9.038)$ | $0.931_{\pm0.006}(9.069)$ | $0.857_{\pm0.013}(0.025)$ | $0.998_{\pm0.016}(0.898)$ | $0.941_{\pm0.002}(0.841)$ | $0.989_{\pm0.002}(0.019)$ | $0.215_{\pm0.007}$ | $0.005_{\pm0.017}$ |
| | 0.2 | $0.895_{\pm0.020}(0.105)$ | $0.816_{\pm0.010}(0.184)$ | $0.823_{\pm0.015}(0.009)$ | $0.895_{\pm0.014}(9.105)$ | $0.850_{\pm0.004}(9.150)$ | $0.831_{\pm0.002}(0.001)$ | $0.999_{\pm0.001}(0.899)$ | $0.960_{\pm0.014}(0.860)$ | $0.991_{\pm0.003}(0.010)$ | $0.078_{\pm0.003}$ | $0.002_{\pm0.009}$ |
| **NegGrad+** UA96.2%, UA$_{tf}$95.2%, RA97.6%, TA92.8% | 0.05 | $0.989_{\pm0.016}(0.011)$ | $0.961_{\pm0.056}(0.039)$ | $0.945_{\pm0.027}(0.019)$ | $9.432_{\pm0.803}(0.568)$ | $9.038_{\pm1.360}(0.962)$ | $1.053_{\pm0.020}(0.096)$ | $0.105_{\pm0.007}(0.005)$ | $0.107_{\pm0.010}(0.007)$ | $0.897_{\pm0.008}(0.058)$ | $1.000_{\pm0.000}$ | $0.835_{\pm0.085}$ |
| | 0.1 | $0.980_{\pm0.029}(0.020)$ | $0.954_{\pm0.065}(0.046)$ | $0.881_{\pm0.028}(0.001)$ | $9.250_{\pm1.061}(0.750)$ | $8.836_{\pm1.647}(1.164)$ | $0.913_{\pm0.018}(0.009)$ | $0.106_{\pm0.009}(0.006)$ | $0.109_{\pm0.013}(0.009)$ | $0.965_{\pm0.012}(0.009)$ | $1.000_{\pm0.000}$ | $0.057_{\pm0.021}$ |
| | 0.15 | $0.952_{\pm0.068}(0.048)$ | $0.908_{\pm0.130}(0.092)$ | $0.849_{\pm0.026}(0.007)$ | $8.600_{\pm1.980}(1.400)$ | $8.077_{\pm2.719}(1.923)$ | $0.868_{\pm0.016}(0.014)$ | $0.113_{\pm0.018}(0.013)$ | $0.116_{\pm0.023}(0.016)$ | $0.977_{\pm0.012}(0.007)$ | $1.000_{\pm0.000}$ | $0.012_{\pm0.003}$ |
| | 0.2 | $0.958_{\pm0.060}(0.042)$ | $0.921_{\pm0.111}(0.079)$ | $0.814_{\pm0.007}(0.001)$ | $8.673_{\pm1.876}(1.327)$ | $8.219_{\pm2.519}(1.781)$ | $0.828_{\pm0.020}(0.002)$ | $0.112_{\pm0.017}(0.012)$ | $0.115_{\pm0.022}(0.015)$ | $0.983_{\pm0.015}(0.001)$ | $1.000_{\pm0.000}$ | $0.004_{\pm0.003}$ |
| **SCRUB** UA32.2%, UA$_{tf}$40.4%, RA98.2%, TA92.5% | 0.05 | $0.984_{\pm0.012}(0.016)$ | $0.936_{\pm0.020}(0.064)$ | $0.961_{\pm0.006}(0.003)$ | $3.867_{\pm1.826}(6.133)$ | $3.220_{\pm1.342}(6.780)$ | $1.144_{\pm0.088}(0.004)$ | $0.290_{\pm0.114}(0.190)$ | $0.321_{\pm0.110}(0.221)$ | $0.843_{\pm0.063}(0.003)$ | $0.998_{\pm0.002}$ | $0.928_{\pm0.095}$ |
| | 0.1 | $0.970_{\pm0.021}(0.030)$ | $0.912_{\pm0.019}(0.088)$ | $0.903_{\pm0.015}(0.021)$ | $3.183_{\pm1.416}(6.817)$ | $2.710_{\pm1.004}(7.290)$ | $0.958_{\pm0.044}(0.035)$ | $0.341_{\pm0.126}(0.241)$ | $0.364_{\pm0.115}(0.264)$ | $0.943_{\pm0.028}(0.013)$ | $0.994_{\pm0.005}$ | $0.282_{\pm0.265}$ |
| | 0.15 | $0.903_{\pm0.075}(0.097)$ | $0.824_{\pm0.067}(0.176)$ | $0.852_{\pm0.022}(0.004)$ | $2.093_{\pm0.963}(7.907)$ | $1.881_{\pm0.784}(8.119)$ | $0.877_{\pm0.018}(0.006)$ | $0.478_{\pm0.151}(0.378)$ | $0.477_{\pm0.141}(0.377)$ | $0.972_{\pm0.005}(0.002)$ | $0.947_{\pm0.048}$ | $0.023_{\pm0.003}$ |
| | 0.2 | $0.881_{\pm0.051}(0.119)$ | $0.783_{\pm0.020}(0.217)$ | $0.810_{\pm0.027}(0.004)$ | $1.764_{\pm0.536}(8.236)$ | $1.610_{\pm0.435}(8.390)$ | $0.825_{\pm0.031}(0.005)$ | $0.527_{\pm0.135}(0.427)$ | $0.509_{\pm0.129}(0.409)$ | $0.981_{\pm0.009}(0.000)$ | $0.906_{\pm0.104}$ | $0.007_{\pm0.006}$ |
| **LoTUS** UA4.8%, UA$_{tf}$12.1%, RA92.6%, TA89.3% | 0.05 | $0.985_{\pm0.009}(0.015)$ | $0.944_{\pm0.019}(0.056)$ | $0.954_{\pm0.021}(0.011)$ | $3.405_{\pm3.532}(6.595)$ | $2.922_{\pm2.780}(7.078)$ | $1.311_{\pm0.224}(0.163)$ | $0.560_{\pm0.427}(0.460)$ | $0.562_{\pm0.403}(0.462)$ | $0.739_{\pm0.103}(0.101)$ | $0.857_{\pm0.134}$ | $0.911_{\pm0.060}$ |
| | 0.1 | $0.959_{\pm0.024}(0.041)$ | $0.916_{\pm0.009}(0.084)$ | $0.899_{\pm0.028}(0.017)$ | $1.266_{\pm0.500}(8.734)$ | $1.313_{\pm0.550}(8.687)$ | $1.019_{\pm0.084}(0.097)$ | $0.824_{\pm0.254}(0.724)$ | $0.772_{\pm0.267}(0.672)$ | $0.884_{\pm0.048}(0.072)$ | $0.771_{\pm0.129}$ | $0.637_{\pm0.181}$ |
| | 0.15 | $0.866_{\pm0.081}(0.134)$ | $0.842_{\pm0.036}(0.158)$ | $0.852_{\pm0.032}(0.005)$ | $0.957_{\pm0.205}(9.043)$ | $1.023_{\pm0.232}(8.977)$ | $0.910_{\pm0.049}(0.028)$ | $0.920_{\pm0.101}(0.820)$ | $0.845_{\pm0.150}(0.745)$ | $0.936_{\pm0.020}(0.034)$ | $0.701_{\pm0.121}$ | $0.372_{\pm0.177}$ |
| | 0.2 | $0.827_{\pm0.074}(0.173)$ | $0.805_{\pm0.039}(0.195)$ | $0.826_{\pm0.025}(0.012)$ | $0.867_{\pm0.112}(9.133)$ | $0.914_{\pm0.088}(9.086)$ | $0.867_{\pm0.027}(0.037)$ | $0.957_{\pm0.041}(0.857)$ | $0.886_{\pm0.087}(0.786)$ | $0.953_{\pm0.008}(0.028)$ | $0.645_{\pm0.031}$ | $0.228_{\pm0.050}$ |
| **MUNBa** UA57.9%, UA$_{tf}$70.3%, RA99.8%, TA95.1% | 0.05 | $0.982_{\pm0.015}(0.018)$ | $0.952_{\pm0.015}(0.048)$ | $0.964_{\pm0.011}(0.000)$ | $8.973_{\pm0.997}(1.027)$ | $8.030_{\pm1.152}(1.970)$ | $1.038_{\pm0.021}(0.110)$ | $0.111_{\pm0.014}(0.011)$ | $0.120_{\pm0.020}(0.020)$ | $0.929_{\pm0.011}(0.089)$ | $0.998_{\pm0.001}$ | $0.786_{\pm0.108}$ |
| | 0.1 | $0.967_{\pm0.021}(0.033)$ | $0.912_{\pm0.023}(0.088)$ | $0.921_{\pm0.011}(0.039)$ | $7.989_{\pm1.898}(2.011)$ | $7.008_{\pm1.843}(2.992)$ | $0.944_{\pm0.009}(0.022)$ | $0.127_{\pm0.036}(0.027)$ | $0.138_{\pm0.043}(0.038)$ | $0.975_{\pm0.003}(0.019)$ | $0.996_{\pm0.003}$ | $0.148_{\pm0.018}$ |
| | 0.15 | $0.959_{\pm0.008}(0.041)$ | $0.875_{\pm0.024}(0.125)$ | $0.869_{\pm0.014}(0.013)$ | $7.141_{\pm2.562}(2.859)$ | $6.242_{\pm2.362}(3.758)$ | $0.880_{\pm0.018}(0.002)$ | $0.150_{\pm0.067}(0.050)$ | $0.157_{\pm0.069}(0.057)$ | $0.987_{\pm0.005}(0.017)$ | $0.992_{\pm0.009}$ | $0.026_{\pm0.016}$ |
| | 0.2 | $0.929_{\pm0.027}(0.071)$ | $0.823_{\pm0.026}(0.177)$ | $0.804_{\pm0.011}(0.010)$ | $6.308_{\pm2.442}(3.692)$ | $5.486_{\pm2.110}(4.514)$ | $0.809_{\pm0.012}(0.021)$ | $0.170_{\pm0.086}(0.070)$ | $0.173_{\pm0.090}(0.073)$ | $0.994_{\pm0.002}(0.013)$ | $0.985_{\pm0.013}$ | $0.005_{\pm0.003}$ |
| **Salun** UA100%, UA$_{tf}$100%, RA99.6%, TA94.3% | 0.05 | $0.996_{\pm0.001}(0.004)$ | $0.941_{\pm0.008}(0.059)$ | $0.952_{\pm0.001}(0.012)$ | $9.996_{\pm0.002}(0.004)$ | $9.892_{\pm0.003}(0.108)$ | $1.028_{\pm0.001}(0.121)$ | $0.100_{\pm0.000}(0.000)$ | $0.095_{\pm0.001}(0.005)$ | $0.926_{\pm0.008}(0.087)$ | $1.000_{\pm0.000}$ | $0.785_{\pm0.049}$ |
| | 0.1 | $0.988_{\pm0.004}(0.012)$ | $0.906_{\pm0.011}(0.094)$ | $0.901_{\pm0.002}(0.020)$ | $9.985_{\pm0.003}(0.015)$ | $9.817_{\pm0.045}(0.183)$ | $0.928_{\pm0.006}(0.006)$ | $0.099_{\pm0.000}(0.001)$ | $0.092_{\pm0.001}(0.008)$ | $0.971_{\pm0.004}(0.015)$ | $1.000_{\pm0.000}$ | $0.042_{\pm0.011}$ |
| | 0.15 | $0.960_{\pm0.003}(0.040)$ | $0.851_{\pm0.005}(0.149)$ | $0.878_{\pm0.006}(0.022)$ | $9.952_{\pm0.000}(0.048)$ | $9.677_{\pm0.088}(0.323)$ | $0.896_{\pm0.005}(0.013)$ | $0.096_{\pm0.000}(0.004)$ | $0.088_{\pm0.000}(0.012)$ | $0.980_{\pm0.001}(0.010)$ | $1.000_{\pm0.000}$ | $0.009_{\pm0.001}$ |
| | 0.2 | $0.915_{\pm0.019}(0.085)$ | $0.807_{\pm0.038}(0.193)$ | $0.820_{\pm0.035}(0.005)$ | $9.893_{\pm0.024}(0.107)$ | $9.511_{\pm0.192}(0.489)$ | $0.828_{\pm0.039}(0.002)$ | $0.092_{\pm0.002}(0.008)$ | $0.085_{\pm0.002}(0.015)$ | $0.990_{\pm0.004}(0.009)$ | $1.000_{\pm0.000}$ | $0.001_{\pm0.001}$ |
| **SFRon** UA100%, UA$_{tf}$100%, RA99.3%, TA94.4% | 0.05 | $1.000_{\pm0.000}(0.000)$ | $1.000_{\pm0.000}(0.000)$ | $0.952_{\pm0.005}(0.013)$ | $10.000_{\pm0.000}(0.000)$ | $10.000_{\pm0.000}(0.000)$ | $1.022_{\pm0.030}(0.127)$ | $0.100_{\pm0.000}(0.000)$ | $0.100_{\pm0.000}(0.000)$ | $0.932_{\pm0.024}(0.092)$ | $1.000_{\pm0.000}$ | $0.677_{\pm0.206}$ |
| | 0.1 | $1.000_{\pm0.000}(0.000)$ | $1.000_{\pm0.000}(0.000)$ | $0.908_{\pm0.013}(0.026)$ | $10.000_{\pm0.000}(0.000)$ | $10.000_{\pm0.000}(0.000)$ | $0.937_{\pm0.028}(0.014)$ | $0.100_{\pm0.000}(0.000)$ | $0.100_{\pm0.000}(0.000)$ | $0.970_{\pm0.015}(0.014)$ | $1.000_{\pm0.000}$ | $0.089_{\pm0.092}$ |
| | 0.15 | $1.000_{\pm0.000}(0.000)$ | $1.000_{\pm0.000}(0.000)$ | $0.840_{\pm0.026}(0.016)$ | $10.000_{\pm0.000}(0.000)$ | $10.000_{\pm0.000}(0.000)$ | $0.849_{\pm0.026}(0.033)$ | $0.100_{\pm0.000}(0.000)$ | $0.100_{\pm0.000}(0.000)$ | $0.989_{\pm0.003}(0.019)$ | $1.000_{\pm0.000}$ | $0.002_{\pm0.001}$ |
| | 0.2 | $1.000_{\pm0.000}(0.000)$ | $1.000_{\pm0.000}(0.000)$ | $0.807_{\pm0.024}(0.008)$ | $10.000_{\pm0.000}(0.000)$ | $10.000_{\pm0.000}(0.000)$ | $0.813_{\pm0.025}(0.017)$ | $0.100_{\pm0.000}(0.000)$ | $0.100_{\pm0.000}(0.000)$ | $0.992_{\pm0.003}(0.010)$ | $1.000_{\pm0.000}$ | $0.001_{\pm0.001}$ |

*Table 16.* Unlearning performance of 9 unlearning methods on **Tiny ImageNet** with **ViT** in $10\%$ **random data forgetting** scenario.

| Methods | $\alpha$ | Coverage | | Set Size | | CR | | $\hat{q}$ |
|---|---|---|---|---|---|---|---|---|
| | | $\mathcal{D}_f \downarrow$ | $\mathcal{D}_{test} \uparrow$ | $\mathcal{D}_f \uparrow$ | $\mathcal{D}_{test} \downarrow$ | $\mathcal{D}_f \downarrow$ | $\mathcal{D}_{test} \uparrow$ | |
| RT
**UA**14.7%, **RA**98.8%, **TA**86.0% | 0.05 | $0.944_{\pm0.006}(0.000)$ | $0.949_{\pm0.026}(0.000)$ | $1.876_{\pm0.009}(0.000)$ | $1.840_{\pm0.014}(0.000)$ | $0.503_{\pm0.018}(0.000)$ | $0.516_{\pm0.018}(0.000)$ | $0.984_{\pm0.002}$ |
| | 0.1 | $0.892_{\pm0.006}(0.000)$ | $0.900_{\pm0.025}(0.000)$ | $1.151_{\pm0.002}(0.000)$ | $1.144_{\pm0.018}(0.000)$ | $0.775_{\pm0.016}(0.000)$ | $0.786_{\pm0.026}(0.000)$ | $0.853_{\pm0.003}$ |
| | 0.15 | $0.841_{\pm0.024}(0.000)$ | $0.850_{\pm0.017}(0.000)$ | $0.956_{\pm0.014}(0.000)$ | $0.956_{\pm0.017}(0.000)$ | $0.880_{\pm0.014}(0.000)$ | $0.889_{\pm0.019}(0.000)$ | $0.539_{\pm0.001}$ |
| | 0.2 | $0.790_{\pm0.015}(0.000)$ | $0.799_{\pm0.023}(0.000)$ | $0.846_{\pm0.004}(0.000)$ | $0.854_{\pm0.014}(0.000)$ | $0.934_{\pm0.012}(0.000)$ | $0.935_{\pm0.015}(0.000)$ | $0.238_{\pm0.012}$ |
| FT
**UA**6.9%, **RA**97.9%, **TA**84.1% | 0.05 | $0.994_{\pm0.005}(0.050)$ | $0.950_{\pm0.019}(0.001)$ | $2.133_{\pm0.008}(0.257)$ | $2.440_{\pm0.011}(0.600)$ | $0.466_{\pm0.009}(0.037)$ | $0.389_{\pm0.016}(0.127)$ | $0.994_{\pm0.020}$ |
| | 0.1 | $0.978_{\pm0.007}(0.086)$ | $0.903_{\pm0.003}(0.003)$ | $1.234_{\pm0.010}(0.083)$ | $1.317_{\pm0.001}(0.173)$ | $0.792_{\pm0.018}(0.017)$ | $0.685_{\pm0.001}(0.101)$ | $0.935_{\pm0.012}$ |
| | 0.15 | $0.938_{\pm0.001}(0.097)$ | $0.851_{\pm0.010}(0.001)$ | $1.014_{\pm0.005}(0.058)$ | $1.017_{\pm0.016}(0.061)$ | $0.925_{\pm0.007}(0.045)$ | $0.836_{\pm0.016}(0.053)$ | $0.681_{\pm0.003}$ |
| | 0.2 | $0.888_{\pm0.009}(0.098)$ | $0.801_{\pm0.012}(0.002)$ | $0.915_{\pm0.006}(0.069)$ | $0.885_{\pm0.000}(0.031)$ | $0.970_{\pm0.020}(0.036)$ | $0.905_{\pm0.005}(0.030)$ | $0.326_{\pm0.011}$ |
| RL
**UA**26.9%, **RA**96.0%, **TA**81.4% | 0.05 | $0.969_{\pm0.021}(0.025)$ | $0.952_{\pm0.008}(0.003)$ | $17.890_{\pm0.003}(16.014)$ | $8.572_{\pm0.010}(6.732)$ | $0.054_{\pm0.013}(0.449)$ | $0.111_{\pm0.002}(0.405)$ | $0.996_{\pm0.019}$ |
| | 0.1 | $0.892_{\pm0.017}(0.000)$ | $0.902_{\pm0.013}(0.002)$ | $2.639_{\pm0.017}(1.488)$ | $1.843_{\pm0.019}(0.699)$ | $0.338_{\pm0.022}(0.437)$ | $0.489_{\pm0.013}(0.297)$ | $0.971_{\pm0.014}$ |
| | 0.15 | $0.793_{\pm0.021}(0.048)$ | $0.855_{\pm0.008}(0.005)$ | $1.225_{\pm0.013}(0.269)$ | $1.164_{\pm0.000}(0.208)$ | $0.648_{\pm0.002}(0.232)$ | $0.734_{\pm0.000}(0.155)$ | $0.894_{\pm0.022}$ |
| | 0.2 | $0.681_{\pm0.010}(0.109)$ | $0.803_{\pm0.003}(0.004)$ | $0.831_{\pm0.006}(0.015)$ | $0.946_{\pm0.011}(0.092)$ | $0.820_{\pm0.022}(0.114)$ | $0.849_{\pm0.006}(0.086)$ | $0.715_{\pm0.013}$ |
| GA
**UA**3.2%, **RA**97.4%, **TA**84.9% | 0.05 | $0.996_{\pm0.003}(0.052)$ | $0.947_{\pm0.002}(0.002)$ | $1.539_{\pm0.004}(0.337)$ | $2.018_{\pm0.007}(0.178)$ | $0.647_{\pm0.003}(0.144)$ | $0.469_{\pm0.002}(0.047)$ | $0.988_{\pm0.004}$ |
| | 0.1 | $0.986_{\pm0.006}(0.094)$ | $0.900_{\pm0.000}(0.000)$ | $1.104_{\pm0.006}(0.047)$ | $1.224_{\pm0.005}(0.080)$ | $0.894_{\pm0.003}(0.119)$ | $0.736_{\pm0.006}(0.050)$ | $0.899_{\pm0.001}$ |
| | 0.15 | $0.967_{\pm0.002}(0.126)$ | $0.852_{\pm0.005}(0.002)$ | $1.003_{\pm0.008}(0.047)$ | $0.993_{\pm0.004}(0.037)$ | $0.964_{\pm0.005}(0.084)$ | $0.859_{\pm0.006}(0.030)$ | $0.632_{\pm0.009}$ |
| | 0.2 | $0.934_{\pm0.001}(0.144)$ | $0.800_{\pm0.007}(0.001)$ | $0.946_{\pm0.008}(0.100)$ | $0.871_{\pm0.008}(0.017)$ | $0.987_{\pm0.008}(0.053)$ | $0.919_{\pm0.005}(0.016)$ | $0.296_{\pm0.009}$ |
| Teacher
**UA**17.3%, **RA**86.7%, **TA**79.0% | 0.05 | $0.977_{\pm0.004}(0.033)$ | $0.956_{\pm0.003}(0.007)$ | $5.473_{\pm0.006}(3.597)$ | $5.080_{\pm0.004}(3.240)$ | $0.179_{\pm0.008}(0.324)$ | $0.188_{\pm0.002}(0.328)$ | $0.987_{\pm0.008}$ |
| | 0.1 | $0.930_{\pm0.003}(0.038)$ | $0.902_{\pm0.008}(0.002)$ | $1.991_{\pm0.004}(0.840)$ | $1.959_{\pm0.002}(0.815)$ | $0.467_{\pm0.004}(0.308)$ | $0.460_{\pm0.002}(0.326)$ | $0.971_{\pm0.007}$ |
| | 0.15 | $0.873_{\pm0.003}(0.032)$ | $0.850_{\pm0.009}(0.000)$ | $1.295_{\pm0.006}(0.339)$ | $1.319_{\pm0.005}(0.363)$ | $0.674_{\pm0.007}(0.206)$ | $0.645_{\pm0.003}(0.244)$ | $0.944_{\pm0.006}$ |
| | 0.2 | $0.816_{\pm0.007}(0.026)$ | $0.803_{\pm0.009}(0.004)$ | $1.020_{\pm0.006}(0.174)$ | $1.058_{\pm0.004}(0.204)$ | $0.800_{\pm0.005}(0.134)$ | $0.758_{\pm0.005}(0.177)$ | $0.910_{\pm0.006}$ |
| SSD
**UA**1.5%, **RA**98.5%, **TA**86.1% | 0.05 | $0.998_{\pm0.004}(0.054)$ | $0.950_{\pm0.006}(0.001)$ | $1.354_{\pm0.008}(0.522)$ | $1.827_{\pm0.002}(0.013)$ | $0.737_{\pm0.008}(0.234)$ | $0.520_{\pm0.008}(0.004)$ | $0.985_{\pm0.005}$ |
| | 0.1 | $0.993_{\pm0.008}(0.101)$ | $0.897_{\pm0.008}(0.003)$ | $1.039_{\pm0.002}(0.112)$ | $1.134_{\pm0.008}(0.010)$ | $0.956_{\pm0.007}(0.181)$ | $0.791_{\pm0.002}(0.005)$ | $0.852_{\pm0.001}$ |
| | 0.15 | $0.981_{\pm0.005}(0.140)$ | $0.853_{\pm0.001}(0.003)$ | $0.993_{\pm0.001}(0.037)$ | $0.962_{\pm0.005}(0.006)$ | $0.988_{\pm0.004}(0.108)$ | $0.887_{\pm0.004}(0.002)$ | $0.542_{\pm0.007}$ |
| | 0.2 | $0.956_{\pm0.002}(0.166)$ | $0.805_{\pm0.003}(0.006)$ | $0.960_{\pm0.003}(0.114)$ | $0.864_{\pm0.009}(0.010)$ | $0.996_{\pm0.005}(0.062)$ | $0.932_{\pm0.002}(0.003)$ | $0.249_{\pm0.006}$ |
| NegGrad+
**UA**19.4%, **RA**98.3%, **TA**84.0% | 0.05 | $0.999_{\pm0.000}(0.146)$ | $0.890_{\pm0.002}(0.030)$ | $0.949_{\pm0.002}(0.005)$ | $1.614_{\pm0.023}(0.665)$ | $1.052_{\pm0.002}(0.823)$ | $0.552_{\pm0.007}(1.289)$ | $0.995_{\pm0.000}$ |
| | 0.1 | $0.995_{\pm0.001}(0.144)$ | $0.848_{\pm0.000}(0.013)$ | $0.898_{\pm0.000}(0.005)$ | $1.093_{\pm0.005}(0.193)$ | $1.109_{\pm0.001}(0.042)$ | $0.775_{\pm0.003}(0.369)$ | $0.933_{\pm0.002}$ |
| | 0.15 | $0.987_{\pm0.000}(0.137)$ | $0.814_{\pm0.001}(0.047)$ | $0.850_{\pm0.001}(0.009)$ | $1.009_{\pm0.000}(0.159)$ | $1.161_{\pm0.001}(0.206)$ | $0.807_{\pm0.002}(0.149)$ | $0.685_{\pm0.002}$ |
| | 0.2 | $0.966_{\pm0.001}(0.108)$ | $0.783_{\pm0.003}(0.078)$ | $0.802_{\pm0.002}(0.012)$ | $0.972_{\pm0.000}(0.173)$ | $1.205_{\pm0.004}(0.359)$ | $0.805_{\pm0.003}(0.049)$ | $0.320_{\pm0.001}$ |
| SCRUB
**UA**1.3%, **RA**98.7%, **TA**86.3% | 0.05 | $0.998_{\pm0.000}(0.054)$ | $0.950_{\pm0.000}(0.001)$ | $1.309_{\pm0.007}(0.567)$ | $1.769_{\pm0.001}(0.071)$ | $0.761_{\pm0.000}(0.257)$ | $0.537_{\pm0.001}(0.021)$ | $0.984_{\pm0.000}$ |
| | 0.1 | $0.992_{\pm0.000}(0.100)$ | $0.897_{\pm0.000}(0.003)$ | $1.033_{\pm0.001}(0.118)$ | $1.122_{\pm0.002}(0.022)$ | $0.961_{\pm0.000}(0.185)$ | $0.800_{\pm0.001}(0.013)$ | $0.841_{\pm0.002}$ |
| | 0.15 | $0.980_{\pm0.002}(0.139)$ | $0.848_{\pm0.000}(0.002)$ | $0.990_{\pm0.001}(0.034)$ | $0.948_{\pm0.000}(0.008)$ | $0.992_{\pm0.001}(0.112)$ | $0.894_{\pm0.000}(0.005)$ | $0.497_{\pm0.003}$ |
| | 0.2 | $0.957_{\pm0.004}(0.167)$ | $0.803_{\pm0.001}(0.004)$ | $0.960_{\pm0.003}(0.114)$ | $0.858_{\pm0.001}(0.004)$ | $0.997_{\pm0.000}(0.063)$ | $0.936_{\pm0.000}(0.001)$ | $0.229_{\pm0.002}$ |
| LoTUS
**UA**7.5%, **RA**85.6%, **TA**78.6% | 0.05 | $0.995_{\pm0.000}(0.051)$ | $0.949_{\pm0.002}(0.000)$ | $2.747_{\pm0.103}(0.871)$ | $3.711_{\pm0.065}(1.871)$ | $0.290_{\pm0.006}(0.213)$ | $0.256_{\pm0.004}(0.260)$ | $0.993_{\pm0.000}$ |
| | 0.1 | $0.983_{\pm0.002}(0.091)$ | $0.898_{\pm0.002}(0.002)$ | $1.465_{\pm0.006}(0.314)$ | $1.781_{\pm0.015}(0.637)$ | $0.570_{\pm0.003}(0.205)$ | $0.504_{\pm0.003}(0.282)$ | $0.959_{\pm0.002}$ |
| | 0.15 | $0.961_{\pm0.004}(0.121)$ | $0.849_{\pm0.004}(0.001)$ | $1.149_{\pm0.006}(0.193)$ | $1.258_{\pm0.011}(0.303)$ | $0.749_{\pm0.006}(0.131)$ | $0.675_{\pm0.003}(0.215)$ | $0.872_{\pm0.006}$ |
| | 0.2 | $0.935_{\pm0.006}(0.145)$ | $0.799_{\pm0.003}(0.000)$ | $1.013_{\pm0.004}(0.167)$ | $1.015_{\pm0.010}(0.160)$ | $0.855_{\pm0.002}(0.079)$ | $0.787_{\pm0.006}(0.148)$ | $0.699_{\pm0.009}$ |
| MUNBa
**UA**1.9%, **RA**99.0%, **TA**86.2% | 0.05 | $0.998_{\pm0.001}(0.054)$ | $0.949_{\pm0.001}(0.000)$ | $1.408_{\pm0.029}(0.468)$ | $1.844_{\pm0.028}(0.003)$ | $0.770_{\pm0.008}(0.267)$ | $0.515_{\pm0.008}(0.001)$ | $0.986_{\pm0.001}$ |
| | 0.1 | $0.991_{\pm0.001}(0.098)$ | $0.899_{\pm0.002}(0.000)$ | $1.052_{\pm0.004}(0.098)$ | $1.137_{\pm0.006}(0.007)$ | $0.966_{\pm0.002}(0.190)$ | $0.791_{\pm0.002}(0.005)$ | $0.859_{\pm0.003}$ |
| | 0.15 | $0.975_{\pm0.002}(0.134)$ | $0.851_{\pm0.001}(0.001)$ | $0.988_{\pm0.000}(0.033)$ | $0.956_{\pm0.002}(0.001)$ | $0.993_{\pm0.000}(0.114)$ | $0.890_{\pm0.001}(0.000)$ | $0.530_{\pm0.008}$ |
| | 0.2 | $0.944_{\pm0.003}(0.154)$ | $0.805_{\pm0.003}(0.007)$ | $0.948_{\pm0.003}(0.102)$ | $0.864_{\pm0.005}(0.009)$ | $0.998_{\pm0.000}(0.064)$ | $0.932_{\pm0.002}(0.002)$ | $0.241_{\pm0.010}$ |
| Salun
**UA**9.2%, **RA**97.7%, **TA**83.6% | 0.05 | $0.995_{\pm0.003}(0.142)$ | $0.964_{\pm0.026}(0.103)$ | $2.803_{\pm1.607}(1.859)$ | $2.726_{\pm0.727}(1.777)$ | $0.528_{\pm0.454}(1.347)$ | $0.376_{\pm0.129}(1.464)$ | $0.988_{\pm0.001}$ |
| | 0.1 | $0.977_{\pm0.014}(0.126)$ | $0.924_{\pm0.040}(0.064)$ | $1.229_{\pm0.286}(0.337)$ | $1.281_{\pm0.120}(0.381)$ | $0.831_{\pm0.237}(0.319)$ | $0.728_{\pm0.104}(0.417)$ | $0.939_{\pm0.005}$ |
| | 0.15 | $0.936_{\pm0.041}(0.086)$ | $0.874_{\pm0.041}(0.013)$ | $0.972_{\pm0.103}(0.131)$ | $1.032_{\pm0.005}(0.182)$ | $0.974_{\pm0.155}(0.018)$ | $0.847_{\pm0.044}(0.109)$ | $0.819_{\pm0.003}$ |
| | 0.2 | $0.870_{\pm0.081}(0.012)$ | $0.810_{\pm0.017}(0.051)$ | $0.845_{\pm0.036}(0.055)$ | $0.925_{\pm0.046}(0.126)$ | $1.034_{\pm0.143}(0.188)$ | $0.876_{\pm0.025}(0.022)$ | $0.630_{\pm0.003}$ |
| SFRon
**UA**9.3%, **RA**97.0%, **TA**83.9% | 0.05 | $0.989_{\pm0.001}(0.045)$ | $0.948_{\pm0.001}(0.001)$ | $2.000_{\pm0.059}(0.124)$ | $2.208_{\pm0.037}(0.368)$ | $0.495_{\pm0.014}(0.008)$ | $0.429_{\pm0.007}(0.086)$ | $0.986_{\pm0.000}$ |
| | 0.1 | $0.960_{\pm0.003}(0.068)$ | $0.899_{\pm0.002}(0.001)$ | $1.227_{\pm0.017}(0.076)$ | $1.268_{\pm0.007}(0.123)$ | $0.783_{\pm0.010}(0.008)$ | $0.709_{\pm0.003}(0.077)$ | $0.902_{\pm0.003}$ |
| | 0.15 | $0.917_{\pm0.002}(0.076)$ | $0.849_{\pm0.002}(0.001)$ | $1.024_{\pm0.006}(0.068)$ | $1.015_{\pm0.005}(0.059)$ | $0.896_{\pm0.007}(0.016)$ | $0.837_{\pm0.004}(0.053)$ | $0.689_{\pm0.012}$ |
| | 0.2 | $0.866_{\pm0.006}(0.076)$ | $0.802_{\pm0.003}(0.003)$ | $0.916_{\pm0.004}(0.070)$ | $0.892_{\pm0.005}(0.037)$ | $0.946_{\pm0.002}(0.012)$ | $0.899_{\pm0.003}(0.036)$ | $0.426_{\pm0.018}$ |

*Table 17.* Unlearning performance of 9 unlearning methods on **Tiny ImageNet** with **ViT** in **50% random data forgetting** scenario.

| Methods | $\alpha$ | Coverage | | Set Size | | CR | | $\hat{q}$ |
|---|---|---|---|---|---|---|---|---|
| | | $\mathcal{D}_f \downarrow$ | $\mathcal{D}_{test} \uparrow$ | $\mathcal{D}_f \uparrow$ | $\mathcal{D}_{test} \downarrow$ | $\mathcal{D}_f \downarrow$ | $\mathcal{D}_{test} \uparrow$ | |
| **RT** **UA**16.0%, **RA**98.8%, **TA**84.9% | 0.05 | $0.946_{\pm 0.001}(0.000)$ | $0.948_{\pm 0.003}(0.000)$ | $2.146_{\pm 0.006}(0.000)$ | $2.106_{\pm 0.002}(0.000)$ | $0.441_{\pm 0.004}(0.000)$ | $0.450_{\pm 0.005}(0.000)$ | $0.987_{\pm 0.004}$ |
| | 0.1 | $0.892_{\pm 0.007}(0.000)$ | $0.899_{\pm 0.008}(0.000)$ | $1.222_{\pm 0.002}(0.000)$ | $1.211_{\pm 0.007}(0.000)$ | $0.730_{\pm 0.004}(0.000)$ | $0.742_{\pm 0.002}(0.000)$ | $0.889_{\pm 0.009}$ |
| | 0.15 | $0.838_{\pm 0.004}(0.000)$ | $0.847_{\pm 0.001}(0.000)$ | $0.977_{\pm 0.002}(0.000)$ | $0.977_{\pm 0.006}(0.000)$ | $0.858_{\pm 0.008}(0.000)$ | $0.868_{\pm 0.006}(0.000)$ | $0.607_{\pm 0.001}$ |
| | 0.2 | $0.786_{\pm 0.005}(0.000)$ | $0.796_{\pm 0.002}(0.000)$ | $0.856_{\pm 0.007}(0.000)$ | $0.863_{\pm 0.001}(0.000)$ | $0.918_{\pm 0.007}(0.000)$ | $0.922_{\pm 0.008}(0.000)$ | $0.304_{\pm 0.008}$ |
| **FT** **UA**5.4%, **RA**97.1%, **TA**84.4% | 0.05 | $0.995_{\pm 0.013}(0.051)$ | $0.949_{\pm 0.024}(0.000)$ | $1.879_{\pm 0.014}(0.003)$ | $2.216_{\pm 0.003}(0.376)$ | $0.527_{\pm 0.028}(0.024)$ | $0.428_{\pm 0.020}(0.088)$ | $0.992_{\pm 0.019}$ |
| | 0.1 | $0.979_{\pm 0.021}(0.087)$ | $0.901_{\pm 0.014}(0.001)$ | $1.183_{\pm 0.018}(0.032)$ | $1.281_{\pm 0.020}(0.137)$ | $0.828_{\pm 0.029}(0.053)$ | $0.701_{\pm 0.010}(0.085)$ | $0.926_{\pm 0.025}$ |
| | 0.15 | $0.953_{\pm 0.024}(0.112)$ | $0.850_{\pm 0.022}(0.000)$ | $1.014_{\pm 0.011}(0.058)$ | $1.017_{\pm 0.026}(0.061)$ | $0.940_{\pm 0.027}(0.060)$ | $0.839_{\pm 0.004}(0.050)$ | $0.681_{\pm 0.020}$ |
| | 0.2 | $0.910_{\pm 0.029}(0.120)$ | $0.806_{\pm 0.024}(0.007)$ | $0.937_{\pm 0.018}(0.091)$ | $0.895_{\pm 0.001}(0.041)$ | $0.977_{\pm 0.029}(0.043)$ | $0.902_{\pm 0.007}(0.033)$ | $0.345_{\pm 0.016}$ |
| **RL** **UA**22.5%, **RA**93.5%, **TA**77.1% | 0.05 | $0.974_{\pm 0.011}(0.028)$ | $0.953_{\pm 0.001}(0.005)$ | $26.032_{\pm 0.007}(23.886)$ | $23.369_{\pm 0.008}(21.263)$ | $0.038_{\pm 0.015}(0.403)$ | $0.038_{\pm 0.016}(0.412)$ | $0.994_{\pm 0.010}$ |
| | 0.1 | $0.930_{\pm 0.016}(0.038)$ | $0.902_{\pm 0.013}(0.003)$ | $5.277_{\pm 0.001}(4.055)$ | $4.621_{\pm 0.007}(3.410)$ | $0.178_{\pm 0.011}(0.552)$ | $0.197_{\pm 0.001}(0.545)$ | $0.987_{\pm 0.008}$ |
| | 0.15 | $0.875_{\pm 0.011}(0.037)$ | $0.856_{\pm 0.008}(0.009)$ | $1.758_{\pm 0.004}(0.781)$ | $1.657_{\pm 0.005}(0.680)$ | $0.496_{\pm 0.006}(0.362)$ | $0.516_{\pm 0.009}(0.352)$ | $0.970_{\pm 0.017}$ |
| | 0.2 | $0.810_{\pm 0.006}(0.024)$ | $0.805_{\pm 0.013}(0.009)$ | $1.147_{\pm 0.005}(0.291)$ | $1.144_{\pm 0.005}(0.281)$ | $0.707_{\pm 0.004}(0.211)$ | $0.707_{\pm 0.013}(0.215)$ | $0.945_{\pm 0.005}$ |
| **GA** **UA**3.9%, **RA**96.1%, **TA**84.2% | 0.05 | $0.998_{\pm 0.007}(0.052)$ | $0.949_{\pm 0.001}(0.001)$ | $1.807_{\pm 0.001}(0.339)$ | $2.338_{\pm 0.001}(0.232)$ | $0.552_{\pm 0.006}(0.111)$ | $0.407_{\pm 0.006}(0.043)$ | $0.992_{\pm 0.006}$ |
| | 0.1 | $0.986_{\pm 0.009}(0.094)$ | $0.896_{\pm 0.007}(0.003)$ | $1.147_{\pm 0.003}(0.075)$ | $1.278_{\pm 0.007}(0.067)$ | $0.863_{\pm 0.008}(0.133)$ | $0.703_{\pm 0.002}(0.039)$ | $0.918_{\pm 0.010}$ |
| | 0.15 | $0.968_{\pm 0.008}(0.130)$ | $0.850_{\pm 0.002}(0.003)$ | $1.015_{\pm 0.008}(0.038)$ | $1.020_{\pm 0.002}(0.043)$ | $0.954_{\pm 0.009}(0.096)$ | $0.835_{\pm 0.002}(0.033)$ | $0.696_{\pm 0.009}$ |
| | 0.2 | $0.931_{\pm 0.011}(0.145)$ | $0.804_{\pm 0.004}(0.008)$ | $0.948_{\pm 0.000}(0.092)$ | $0.893_{\pm 0.003}(0.030)$ | $0.983_{\pm 0.006}(0.065)$ | $0.900_{\pm 0.004}(0.022)$ | $0.363_{\pm 0.002}$ |
| **Teacher** **UA**22.1%, **RA**85.7%, **TA**76.2% | 0.05 | $0.967_{\pm 0.013}(0.021)$ | $0.950_{\pm 0.017}(0.002)$ | $6.465_{\pm 0.007}(4.319)$ | $6.233_{\pm 0.004}(4.127)$ | $0.151_{\pm 0.002}(0.290)$ | $0.151_{\pm 0.006}(0.299)$ | $0.990_{\pm 0.014}$ |
| | 0.1 | $0.922_{\pm 0.008}(0.030)$ | $0.899_{\pm 0.002}(0.000)$ | $2.202_{\pm 0.012}(0.980)$ | $2.167_{\pm 0.005}(0.956)$ | $0.418_{\pm 0.009}(0.312)$ | $0.419_{\pm 0.024}(0.323)$ | $0.977_{\pm 0.001}$ |
| | 0.15 | $0.869_{\pm 0.025}(0.031)$ | $0.852_{\pm 0.002}(0.005)$ | $1.467_{\pm 0.015}(0.490)$ | $1.459_{\pm 0.004}(0.482)$ | $0.591_{\pm 0.005}(0.267)$ | $0.581_{\pm 0.001}(0.287)$ | $0.958_{\pm 0.021}$ |
| | 0.2 | $0.814_{\pm 0.020}(0.028)$ | $0.801_{\pm 0.017}(0.005)$ | $1.125_{\pm 0.005}(0.269)$ | $1.138_{\pm 0.001}(0.275)$ | $0.718_{\pm 0.017}(0.200)$ | $0.704_{\pm 0.009}(0.218)$ | $0.927_{\pm 0.017}$ |
| **SSD** **UA**1.3%, **RA**98.4%, **TA**86.1% | 0.05 | $0.999_{\pm 0.001}(0.053)$ | $0.952_{\pm 0.001}(0.004)$ | $1.346_{\pm 0.001}(0.800)$ | $1.824_{\pm 0.000}(0.282)$ | $0.742_{\pm 0.000}(0.301)$ | $0.522_{\pm 0.001}(0.072)$ | $0.986_{\pm 0.001}$ |
| | 0.1 | $0.995_{\pm 0.001}(0.103)$ | $0.897_{\pm 0.000}(0.002)$ | $1.033_{\pm 0.001}(0.189)$ | $1.135_{\pm 0.001}(0.076)$ | $0.959_{\pm 0.000}(0.229)$ | $0.790_{\pm 0.000}(0.048)$ | $0.847_{\pm 0.001}$ |
| | 0.15 | $0.982_{\pm 0.001}(0.144)$ | $0.847_{\pm 0.000}(0.000)$ | $0.987_{\pm 0.000}(0.010)$ | $0.956_{\pm 0.000}(0.021)$ | $0.989_{\pm 0.001}(0.131)$ | $0.890_{\pm 0.001}(0.022)$ | $0.517_{\pm 0.001}$ |
| | 0.2 | $0.959_{\pm 0.001}(0.173)$ | $0.804_{\pm 0.001}(0.008)$ | $0.961_{\pm 0.000}(0.105)$ | $0.862_{\pm 0.000}(0.001)$ | $0.995_{\pm 0.001}(0.077)$ | $0.932_{\pm 0.001}(0.010)$ | $0.243_{\pm 0.001}$ |
| **NegGrad+** **UA**11.5%, **RA**98.7%, **TA**83.8% | 0.05 | $0.999_{\pm 0.000}(0.053)$ | $0.979_{\pm 0.001}(0.031)$ | $0.946_{\pm 0.002}(1.200)$ | $1.443_{\pm 0.028}(0.663)$ | $1.056_{\pm 0.002}(0.615)$ | $0.678_{\pm 0.012}(0.228)$ | $0.992_{\pm 0.001}$ |
| | 0.1 | $0.996_{\pm 0.000}(0.104)$ | $0.946_{\pm 0.002}(0.047)$ | $0.900_{\pm 0.003}(0.322)$ | $1.078_{\pm 0.006}(0.134)$ | $1.107_{\pm 0.003}(0.377)$ | $0.877_{\pm 0.003}(0.135)$ | $0.933_{\pm 0.003}$ |
| | 0.15 | $0.990_{\pm 0.000}(0.152)$ | $0.900_{\pm 0.003}(0.052)$ | $0.853_{\pm 0.004}(0.124)$ | $1.008_{\pm 0.002}(0.031)$ | $1.161_{\pm 0.005}(0.303)$ | $0.892_{\pm 0.001}(0.025)$ | $0.712_{\pm 0.015}$ |
| | 0.2 | $0.977_{\pm 0.000}(0.191)$ | $0.848_{\pm 0.003}(0.052)$ | $0.805_{\pm 0.002}(0.052)$ | $0.982_{\pm 0.000}(0.119)$ | $1.214_{\pm 0.003}(0.296)$ | $0.863_{\pm 0.003}(0.058)$ | $0.381_{\pm 0.009}$ |
| **SCRUB** **UA**49.8%, **RA**50.3%, **TA**44.0% | 0.05 | $0.974_{\pm 0.034}(0.028)$ | $0.950_{\pm 0.001}(0.002)$ | $92.143_{\pm 128.417}(89.998)$ | $92.086_{\pm 127.675}(89.980)$ | $0.378_{\pm 0.527}(0.063)$ | $0.265_{\pm 0.368}(0.185)$ | $0.991_{\pm 0.009}$ |
| | 0.1 | $0.948_{\pm 0.063}(0.056)$ | $0.900_{\pm 0.001}(0.001)$ | $84.271_{\pm 117.707}(83.049)$ | $83.935_{\pm 117.094}(82.724)$ | $0.480_{\pm 0.671}(0.250)$ | $0.398_{\pm 0.555}(0.344)$ | $0.926_{\pm 0.101}$ |
| | 0.15 | $0.917_{\pm 0.090}(0.079)$ | $0.848_{\pm 0.003}(0.001)$ | $77.298_{\pm 107.918}(76.322)$ | $76.961_{\pm 107.498}(75.984)$ | $0.499_{\pm 0.698}(0.359)$ | $0.449_{\pm 0.627}(0.419)$ | $0.748_{\pm 0.352}$ |
| | 0.2 | $0.879_{\pm 0.110}(0.093)$ | $0.801_{\pm 0.002}(0.005)$ | $70.976_{\pm 99.018}(70.120)$ | $70.668_{\pm 98.724}(69.804)$ | $0.501_{\pm 0.701}(0.416)$ | $0.470_{\pm 0.656}(0.452)$ | $0.615_{\pm 0.540}$ |
| **LoTUS** **UA**6.9%, **RA**86.4%, **TA**79.9% | 0.05 | $0.994_{\pm 0.000}(0.048)$ | $0.949_{\pm 0.003}(0.001)$ | $2.433_{\pm 0.124}(0.287)$ | $3.187_{\pm 0.213}(1.081)$ | $0.335_{\pm 0.020}(0.106)$ | $0.299_{\pm 0.020}(0.151)$ | $0.989_{\pm 0.001}$ |
| | 0.1 | $0.981_{\pm 0.000}(0.089)$ | $0.900_{\pm 0.001}(0.001)$ | $1.395_{\pm 0.020}(0.173)$ | $1.643_{\pm 0.038}(0.432)$ | $0.611_{\pm 0.011}(0.119)$ | $0.548_{\pm 0.013}(0.194)$ | $0.946_{\pm 0.003}$ |
| | 0.15 | $0.959_{\pm 0.001}(0.121)$ | $0.849_{\pm 0.001}(0.002)$ | $1.113_{\pm 0.007}(0.136)$ | $1.180_{\pm 0.013}(0.203)$ | $0.786_{\pm 0.006}(0.072)$ | $0.719_{\pm 0.008}(0.148)$ | $0.839_{\pm 0.007}$ |
| | 0.2 | $0.930_{\pm 0.002}(0.144)$ | $0.803_{\pm 0.001}(0.007)$ | $0.999_{\pm 0.005}(0.142)$ | $0.982_{\pm 0.005}(0.118)$ | $0.876_{\pm 0.006}(0.042)$ | $0.818_{\pm 0.003}(0.104)$ | $0.674_{\pm 0.014}$ |
| **MUNBa** **UA**2.7%, **RA**97.7%, **TA**85.4% | 0.05 | $0.996_{\pm 0.000}(0.050)$ | $0.948_{\pm 0.001}(0.001)$ | $1.987_{\pm 0.022}(0.159)$ | $2.131_{\pm 0.026}(0.025)$ | $0.516_{\pm 0.007}(0.075)$ | $0.445_{\pm 0.005}(0.005)$ | $0.976_{\pm 0.001}$ |
| | 0.1 | $0.982_{\pm 0.000}(0.090)$ | $0.897_{\pm 0.001}(0.002)$ | $1.240_{\pm 0.006}(0.018)$ | $1.276_{\pm 0.007}(0.064)$ | $0.801_{\pm 0.003}(0.071)$ | $0.703_{\pm 0.003}(0.039)$ | $0.946_{\pm 0.001}$ |
| | 0.15 | $0.955_{\pm 0.001}(0.118)$ | $0.848_{\pm 0.001}(0.000)$ | $1.043_{\pm 0.003}(0.067)$ | $1.018_{\pm 0.006}(0.041)$ | $0.920_{\pm 0.002}(0.062)$ | $0.833_{\pm 0.004}(0.035)$ | $0.904_{\pm 0.002}$ |
| | 0.2 | $0.914_{\pm 0.001}(0.128)$ | $0.797_{\pm 0.003}(0.001)$ | $0.947_{\pm 0.001}(0.091)$ | $0.886_{\pm 0.002}(0.022)$ | $0.967_{\pm 0.001}(0.049)$ | $0.900_{\pm 0.001}(0.022)$ | $0.852_{\pm 0.002}$ |
| **Salun** **UA**9.2%, **RA**95.7%, **TA**81.9% | 0.05 | $0.993_{\pm 0.003}(0.047)$ | $0.962_{\pm 0.026}(0.014)$ | $3.284_{\pm 2.048}(1.138)$ | $4.112_{\pm 0.813}(2.007)$ | $0.500_{\pm 0.478}(0.059)$ | $0.241_{\pm 0.057}(0.209)$ | $0.989_{\pm 0.001}$ |
| | 0.1 | $0.976_{\pm 0.011}(0.084)$ | $0.924_{\pm 0.039}(0.026)$ | $1.386_{\pm 0.423}(0.164)$ | $1.579_{\pm 0.130}(0.368)$ | $0.764_{\pm 0.292}(0.034)$ | $0.590_{\pm 0.077}(0.152)$ | $0.973_{\pm 0.002}$ |
| | 0.15 | $0.944_{\pm 0.024}(0.106)$ | $0.876_{\pm 0.046}(0.029)$ | $1.051_{\pm 0.175}(0.074)$ | $1.139_{\pm 0.017}(0.162)$ | $0.920_{\pm 0.195}(0.062)$ | $0.770_{\pm 0.051}(0.098)$ | $0.942_{\pm 0.002}$ |
| | 0.2 | $0.900_{\pm 0.044}(0.114)$ | $0.825_{\pm 0.049}(0.029)$ | $0.910_{\pm 0.097}(0.054)$ | $0.969_{\pm 0.037}(0.105)$ | $1.000_{\pm 0.164}(0.082)$ | $0.851_{\pm 0.020}(0.071)$ | $0.893_{\pm 0.002}$ |
| **SFRon** **UA**6.3%, **RA**96.8%, **TA**82.9% | 0.05 | $0.994_{\pm 0.001}(0.048)$ | $0.947_{\pm 0.003}(0.001)$ | $2.010_{\pm 0.188}(0.136)$ | $2.327_{\pm 0.087}(0.222)$ | $0.497_{\pm 0.045}(0.057)$ | $0.407_{\pm 0.016}(0.043)$ | $0.983_{\pm 0.002}$ |
| | 0.1 | $0.980_{\pm 0.006}(0.087)$ | $0.900_{\pm 0.003}(0.001)$ | $1.245_{\pm 0.060}(0.023)$ | $1.338_{\pm 0.039}(0.126)$ | $0.788_{\pm 0.041}(0.058)$ | $0.673_{\pm 0.020}(0.069)$ | $0.909_{\pm 0.003}$ |
| | 0.15 | $0.951_{\pm 0.011}(0.113)$ | $0.849_{\pm 0.003}(0.001)$ | $1.041_{\pm 0.020}(0.065)$ | $1.044_{\pm 0.023}(0.067)$ | $0.913_{\pm 0.028}(0.055)$ | $0.813_{\pm 0.016}(0.055)$ | $0.738_{\pm 0.029}$ |
| | 0.2 | $0.910_{\pm 0.011}(0.125)$ | $0.803_{\pm 0.003}(0.008)$ | $0.947_{\pm 0.006}(0.091)$ | $0.910_{\pm 0.022}(0.046)$ | $0.961_{\pm 0.017}(0.044)$ | $0.884_{\pm 0.017}(0.038)$ | $0.523_{\pm 0.068}$ |

*Table 18.* Unlearning performance of 9 unlearning methods on **Tiny ImageNet** with **ViT** in **class-wise forgetting** scenario.

| Methods | $\alpha$ | Coverage $\mathcal{D}_f\downarrow$ | $\mathcal{D}_{tf}\downarrow$ | $\mathcal{D}_{tr}\uparrow$ | Set Size $\mathcal{D}_f\uparrow$ | $\mathcal{D}_{tf}\uparrow$ | $\mathcal{D}_{tr}\downarrow$ | CR $\mathcal{D}_f\downarrow$ | $\mathcal{D}_{tf}\downarrow$ | $\mathcal{D}_{tr}\uparrow$ | $\hat{q}_f$ | $\hat{q}_{test}$ |
|---|---|---|---|---|---|---|---|---|---|---|---|---|
| RT
UA100%, UA$_{tf}$100%,
RA98.7%, TA86.4% | 0.05
0.1
0.15
0.2 | $1.000_{\pm0.000}(0.000)$
$0.936_{\pm0.011}(0.000)$
$0.904_{\pm0.039}(0.000)$
$0.787_{\pm0.061}(0.000)$ | $1.000_{\pm0.000}(0.000)$
$0.960_{\pm0.016}(0.000)$
$0.960_{\pm0.046}(0.000)$
$0.860_{\pm0.024}(0.000)$ | $0.950_{\pm0.003}(0.000)$
$0.903_{\pm0.009}(0.000)$
$0.853_{\pm0.005}(0.000)$
$0.805_{\pm0.003}(0.000)$ | $200.000_{\pm0.000}(0.000)$
$192.882_{\pm0.912}(0.000)$
$186.791_{\pm2.173}(0.000)$
$171.051_{\pm3.183}(0.000)$ | $200.000_{\pm0.000}(0.000)$
$193.340_{\pm2.620}(0.000)$
$188.880_{\pm1.802}(0.000)$
$174.480_{\pm2.311}(0.000)$ | $1.785_{\pm0.056}(0.000)$
$1.146_{\pm0.002}(0.000)$
$0.957_{\pm0.010}(0.000)$
$0.860_{\pm0.010}(0.000)$ | $0.005_{\pm0.000}(0.000)$
$0.005_{\pm0.000}(0.000)$
$0.005_{\pm0.000}(0.000)$
$0.005_{\pm0.000}(0.000)$ | $0.005_{\pm0.000}(0.000)$
$0.005_{\pm0.000}(0.000)$
$0.005_{\pm0.000}(0.000)$
$0.005_{\pm0.000}(0.000)$ | $0.532_{\pm0.009}(0.000)$
$0.788_{\pm0.008}(0.000)$
$0.892_{\pm0.003}(0.000)$
$0.936_{\pm0.002}(0.000)$ | $1.000_{\pm0.000}$
$1.000_{\pm0.000}$
$1.000_{\pm0.000}$
$1.000_{\pm0.000}$ | $0.984_{\pm0.002}$
$0.859_{\pm0.004}$
$0.535_{\pm0.002}$
$0.232_{\pm0.001}$ |
| FT
UA13.8%, UA$_{tf}$22.0%,
RA97.5%, TA84.1% | 0.05
0.1
0.15
0.2 | $0.993_{\pm0.006}(0.007)$
$0.984_{\pm0.009}(0.048)$
$0.902_{\pm0.019}(0.002)$
$0.860_{\pm0.021}(0.073)$ | $0.960_{\pm0.009}(0.040)$
$0.860_{\pm0.013}(0.100)$
$0.800_{\pm0.004}(0.160)$
$0.760_{\pm0.003}(0.100)$ | $0.952_{\pm0.006}(0.002)$
$0.898_{\pm0.005}(0.005)$
$0.852_{\pm0.017}(0.001)$
$0.800_{\pm0.018}(0.005)$ | $8.360_{\pm0.007}(191.640)$
$1.802_{\pm0.009}(191.080)$
$1.120_{\pm0.021}(185.671)$
$0.969_{\pm0.002}(170.082)$ | $8.280_{\pm0.006}(191.720)$
$1.660_{\pm0.018}(191.680)$
$1.040_{\pm0.006}(187.840)$
$0.960_{\pm0.003}(173.520)$ | $2.442_{\pm0.011}(0.657)$
$1.287_{\pm0.009}(0.141)$
$1.021_{\pm0.017}(0.064)$
$0.882_{\pm0.010}(0.022)$ | $0.119_{\pm0.018}(0.114)$
$0.546_{\pm0.008}(0.541)$
$0.806_{\pm0.012}(0.801)$
$0.888_{\pm0.005}(0.883)$ | $0.116_{\pm0.001}(0.111)$
$0.518_{\pm0.004}(0.513)$
$0.769_{\pm0.013}(0.764)$
$0.792_{\pm0.002}(0.787)$ | $0.390_{\pm0.023}(0.142)$
$0.698_{\pm0.019}(0.090)$
$0.835_{\pm0.022}(0.057)$
$0.907_{\pm0.006}(0.029)$ | $0.999_{\pm0.006}$
$0.971_{\pm0.019}$
$0.809_{\pm0.010}$
$0.595_{\pm0.002}$ | $0.993_{\pm0.005}$
$0.924_{\pm0.016}$
$0.686_{\pm0.004}$
$0.338_{\pm0.019}$ |
| RL
UA100%, UA$_{tf}$100%,
RA98.2%, TA84.6% | 0.05
0.1
0.15
0.2 | $0.998_{\pm0.005}(0.002)$
$0.971_{\pm0.013}(0.035)$
$0.922_{\pm0.011}(0.018)$
$0.882_{\pm0.007}(0.095)$ | $0.980_{\pm0.003}(0.020)$
$0.900_{\pm0.017}(0.060)$
$0.900_{\pm0.011}(0.060)$
$0.860_{\pm0.007}(0.000)$ | $0.952_{\pm0.049}(0.002)$
$0.900_{\pm0.002}(0.003)$
$0.852_{\pm0.015}(0.001)$
$0.807_{\pm0.007}(0.002)$ | $199.489_{\pm0.512}(0.511)$
$180.442_{\pm0.710}(12.440)$
$165.884_{\pm2.037}(20.907)$
$154.896_{\pm2.028}(16.155)$ | $195.220_{\pm1.003}(4.780)$
$170.960_{\pm0.948}(22.380)$
$159.980_{\pm1.012}(28.900)$
$149.280_{\pm3.013}(25.200)$ | $2.317_{\pm0.006}(0.532)$
$1.237_{\pm0.050}(0.991)$
$1.001_{\pm0.000}(0.044)$
$0.886_{\pm0.032}(0.026)$ | $0.005_{\pm0.000}(0.000)$
$0.005_{\pm0.000}(0.000)$
$0.006_{\pm0.001}(0.001)$
$0.006_{\pm0.000}(0.001)$ | $0.005_{\pm0.000}(0.000)$
$0.005_{\pm0.000}(0.000)$
$0.006_{\pm0.000}(0.001)$
$0.006_{\pm0.001}(0.001)$ | $0.411_{\pm0.000}(0.121)$
$0.727_{\pm0.016}(0.061)$
$0.851_{\pm0.023}(0.041)$
$0.912_{\pm0.013}(0.024)$ | $1.000_{\pm0.000}$
$1.000_{\pm0.000}$
$1.000_{\pm0.000}$
$1.000_{\pm0.000}$ | $0.995_{\pm0.032}$
$0.925_{\pm0.024}$
$0.641_{\pm0.035}$
$0.262_{\pm0.022}$ |
| GA
UA9.1%, UA$_{tf}$20.0%,
RA98.6%, TA86.1% | 0.05
0.1
0.15
0.2 | $1.000_{\pm0.001}(0.000)$
$0.991_{\pm0.022}(0.055)$
$0.958_{\pm0.002}(0.054)$
$0.880_{\pm0.047}(0.093)$ | $0.980_{\pm0.002}(0.020)$
$0.900_{\pm0.014}(0.060)$
$0.820_{\pm0.010}(0.140)$
$0.800_{\pm0.051}(0.060)$ | $0.948_{\pm0.026}(0.002)$
$0.897_{\pm0.016}(0.006)$
$0.850_{\pm0.006}(0.003)$
$0.803_{\pm0.025}(0.002)$ | $22.836_{\pm0.045}(177.164)$
$1.631_{\pm0.031}(191.251)$
$1.151_{\pm0.039}(185.640)$
$0.929_{\pm0.002}(170.122)$ | $20.600_{\pm0.011}(179.400)$
$1.720_{\pm0.005}(191.620)$
$1.140_{\pm0.042}(187.740)$
$0.900_{\pm0.009}(173.580)$ | $1.781_{\pm0.017}(0.004)$
$1.133_{\pm0.044}(0.013)$
$0.958_{\pm0.026}(0.001)$
$0.861_{\pm0.000}(0.001)$ | $0.044_{\pm0.017}(0.019)$
$0.608_{\pm0.006}(0.603)$
$0.832_{\pm0.003}(0.827)$
$0.947_{\pm0.036}(0.942)$ | $0.048_{\pm0.028}(0.043)$
$0.523_{\pm0.007}(0.518)$
$0.719_{\pm0.021}(0.714)$
$0.889_{\pm0.029}(0.884)$ | $0.532_{\pm0.013}(0.000)$
$0.792_{\pm0.037}(0.004)$
$0.887_{\pm0.044}(0.005)$
$0.933_{\pm0.027}(0.003)$ | $1.000_{\pm0.000}$
$0.972_{\pm0.033}$
$0.868_{\pm0.023}$
$0.473_{\pm0.016}$ | $0.984_{\pm0.033}$
$0.849_{\pm0.039}$
$0.535_{\pm0.011}$
$0.238_{\pm0.000}$ |
| Teacher
UA100%, UA$_{tf}$100%,
RA88.8%, TA78.6% | 0.05
0.1
0.15
0.2 | $0.982_{\pm0.014}(0.018)$
$0.909_{\pm0.013}(0.027)$
$0.887_{\pm0.014}(0.017)$
$0.838_{\pm0.022}(0.051)$ | $1.000_{\pm0.007}(0.000)$
$0.940_{\pm0.015}(0.020)$
$0.880_{\pm0.011}(0.080)$
$0.840_{\pm0.002}(0.020)$ | $0.952_{\pm0.025}(0.002)$
$0.903_{\pm0.032}(0.000)$
$0.854_{\pm0.003}(0.001)$
$0.799_{\pm0.017}(0.006)$ | $199.971_{\pm0.009}(0.029)$
$199.813_{\pm0.009}(6.931)$
$199.667_{\pm0.030}(12.876)$
$199.413_{\pm0.024}(28.362)$ | $200.000_{\pm0.000}(0.000)$
$199.900_{\pm0.013}(6.560)$
$199.760_{\pm0.026}(10.880)$
$199.620_{\pm0.030}(25.140)$ | $5.095_{\pm0.020}(3.310)$
$2.033_{\pm0.031}(0.887)$
$1.331_{\pm0.012}(0.374)$
$1.022_{\pm0.017}(0.162)$ | $0.005_{\pm0.000}(0.000)$
$0.005_{\pm0.000}(0.000)$
$0.004_{\pm0.000}(0.001)$
$0.004_{\pm0.001}(0.001)$ | $0.005_{\pm0.000}(0.000)$
$0.005_{\pm0.000}(0.000)$
$0.004_{\pm0.001}(0.001)$
$0.004_{\pm0.001}(0.001)$ | $0.187_{\pm0.008}(0.345)$
$0.444_{\pm0.006}(0.344)$
$0.641_{\pm0.010}(0.251)$
$0.781_{\pm0.019}(0.155)$ | $1.000_{\pm0.000}$
$1.000_{\pm0.000}$
$1.000_{\pm0.000}$
$1.000_{\pm0.000}$ | $0.989_{\pm0.001}$
$0.965_{\pm0.003}$
$0.919_{\pm0.001}$
$0.825_{\pm0.002}$ |
| SSD
UA100%, UA$_{tf}$100%,
RA98.4%, TA86.1% | 0.05
0.1
0.15
0.2 | $1.000_{\pm0.000}(0.000)$
$0.949_{\pm0.017}(0.013)$
$0.913_{\pm0.007}(0.009)$
$0.833_{\pm0.007}(0.046)$ | $1.000_{\pm0.000}(0.000)$
$0.900_{\pm0.012}(0.060)$
$0.880_{\pm0.020}(0.080)$
$0.800_{\pm0.013}(0.060)$ | $0.950_{\pm0.017}(0.000)$
$0.897_{\pm0.007}(0.006)$
$0.852_{\pm0.015}(0.001)$
$0.806_{\pm0.022}(0.001)$ | $198.769_{\pm0.052}(1.231)$
$171.073_{\pm0.209}(21.809)$
$157.140_{\pm1.209}(29.651)$
$136.502_{\pm3.022}(34.549)$ | $197.320_{\pm1.010}(2.680)$
$169.360_{\pm2.002}(23.980)$
$154.960_{\pm0.907}(33.920)$
$136.420_{\pm2.422}(38.060)$ | $1.866_{\pm0.019}(0.081)$
$1.141_{\pm0.014}(0.005)$
$0.959_{\pm0.011}(0.002)$
$0.864_{\pm0.002}(0.004)$ | $0.005_{\pm0.000}(0.000)$
$0.006_{\pm0.000}(0.001)$
$0.006_{\pm0.001}(0.001)$
$0.006_{\pm0.000}(0.001)$ | $0.005_{\pm0.000}(0.000)$
$0.005_{\pm0.000}(0.000)$
$0.006_{\pm0.000}(0.001)$
$0.006_{\pm0.000}(0.001)$ | $0.509_{\pm0.013}(0.023)$
$0.786_{\pm0.021}(0.002)$
$0.888_{\pm0.012}(0.004)$
$0.932_{\pm0.015}(0.001)$ | $1.000_{\pm0.000}$
$1.000_{\pm0.000}$
$1.000_{\pm0.000}$
$1.000_{\pm0.000}$ | $0.986_{\pm0.006}$
$0.854_{\pm0.006}$
$0.538_{\pm0.007}$
$0.254_{\pm0.005}$ |
| NegGrad+
UA100%, UA$_{tf}$100%,
RA99.0%, TA85.8% | 0.05
0.1
0.15
0.2 | $1.000_{\pm0.000}(0.000)$
$0.927_{\pm0.104}(0.009)$
$0.862_{\pm0.013}(0.042)$
$0.830_{\pm0.027}(0.043)$ | $1.000_{\pm0.000}(0.000)$
$0.950_{\pm0.071}(0.010)$
$0.870_{\pm0.042}(0.090)$
$0.840_{\pm0.085}(0.020)$ | $0.947_{\pm0.002}(0.003)$
$0.894_{\pm0.001}(0.009)$
$0.849_{\pm0.000}(0.004)$
$0.802_{\pm0.002}(0.003)$ | $200.000_{\pm0.000}(0.000)$
$193.994_{\pm8.493}(1.112)$
$188.686_{\pm0.954}(1.894)$
$187.219_{\pm0.064}(16.168)$ | $200.000_{\pm0.000}(0.000)$
$197.490_{\pm3.550}(4.150)$
$195.590_{\pm0.863}(6.710)$
$194.310_{\pm0.948}(19.830)$ | $1.850_{\pm0.036}(0.065)$
$1.140_{\pm0.007}(0.006)$
$0.961_{\pm0.000}(0.004)$
$0.861_{\pm0.001}(0.002)$ | $0.005_{\pm0.000}(0.000)$
$0.005_{\pm0.000}(0.000)$
$0.005_{\pm0.000}(0.000)$
$0.004_{\pm0.000}(0.000)$ | $0.005_{\pm0.000}(0.000)$
$0.005_{\pm0.000}(0.000)$
$0.004_{\pm0.000}(0.001)$
$0.004_{\pm0.000}(0.001)$ | $0.512_{\pm0.009}(0.020)$
$0.784_{\pm0.004}(0.004)$
$0.884_{\pm0.000}(0.008)$
$0.931_{\pm0.001}(0.005)$ | $1.000_{\pm0.000}$
$1.000_{\pm0.000}$
$1.000_{\pm0.000}$
$1.000_{\pm0.000}$ | $0.987_{\pm0.001}$
$0.859_{\pm0.003}$
$0.537_{\pm0.003}$
$0.220_{\pm0.002}$ |
| SCRUB
UA100.0%, UA$_{tf}$100.0%,
RA98.6%, TA86.1% | 0.05
0.1
0.15
0.2 | $1.000_{\pm0.000}(0.000)$
$1.000_{\pm0.000}(0.064)$
$1.000_{\pm0.000}(0.096)$
$1.000_{\pm0.000}(0.213)$ | $1.000_{\pm0.000}(0.000)$
$1.000_{\pm0.000}(0.040)$
$1.000_{\pm0.000}(0.040)$
$1.000_{\pm0.000}(0.140)$ | $0.950_{\pm0.001}(0.001)$
$0.897_{\pm0.001}(0.005)$
$0.847_{\pm0.001}(0.006)$
$0.803_{\pm0.001}(0.002)$ | $200.000_{\pm0.000}(0.000)$
$200.000_{\pm0.000}(7.118)$
$200.000_{\pm0.000}(13.209)$
$200.000_{\pm0.000}(28.949)$ | $200.000_{\pm0.000}(0.000)$
$200.000_{\pm0.000}(6.660)$
$200.000_{\pm0.000}(11.120)$
$200.000_{\pm0.000}(25.520)$ | $1.793_{\pm0.021}(0.009)$
$1.125_{\pm0.003}(0.021)$
$0.949_{\pm0.001}(0.008)$
$0.859_{\pm0.002}(0.000)$ | $0.005_{\pm0.000}(0.000)$
$0.005_{\pm0.000}(0.000)$
$0.005_{\pm0.000}(0.000)$
$0.005_{\pm0.000}(0.000)$ | $0.005_{\pm0.000}(0.000)$
$0.005_{\pm0.000}(0.000)$
$0.005_{\pm0.000}(0.000)$
$0.005_{\pm0.000}(0.000)$ | $0.530_{\pm0.006}(0.003)$
$0.798_{\pm0.002}(0.010)$
$0.893_{\pm0.000}(0.001)$
$0.934_{\pm0.001}(0.002)$ | $1.000_{\pm0.000}$
$1.000_{\pm0.000}$
$1.000_{\pm0.000}$
$1.000_{\pm0.000}$ | $0.984_{\pm0.001}$
$0.844_{\pm0.004}$
$0.502_{\pm0.002}$
$0.232_{\pm0.002}$ |
| LoTUS
UA8.8%, UA$_{tf}$6.0%,
RA86.1%, TA79.0% | 0.05
0.1
0.15
0.2 | $1.000_{\pm0.000}(0.000)$
$0.939_{\pm0.011}(0.003)$
$0.897_{\pm0.035}(0.007)$
$0.831_{\pm0.026}(0.044)$ | $1.000_{\pm0.000}(0.000)$
$0.947_{\pm0.023}(0.013)$
$0.913_{\pm0.042}(0.047)$
$0.860_{\pm0.040}(0.000)$ | $0.949_{\pm0.002}(0.001)$
$0.901_{\pm0.004}(0.002)$
$0.850_{\pm0.002}(0.004)$
$0.800_{\pm0.004}(0.005)$ | $199.071_{\pm1.253}(0.983)$
$1.297_{\pm0.325}(191.585)$
$1.184_{\pm0.216}(185.607)$
$1.059_{\pm0.202}(169.993)$ | $197.960_{\pm0.295}(2.040)$
$1.193_{\pm0.248}(192.147)$
$1.107_{\pm0.136}(187.773)$
$1.027_{\pm0.103}(173.453)$ | $3.771_{\pm0.236}(1.987)$
$1.834_{\pm0.074}(0.688)$
$1.261_{\pm0.023}(0.304)$
$1.011_{\pm0.009}(0.152)$ | $0.005_{\pm0.000}(0.000)$
$0.750_{\pm0.159}(0.745)$
$0.775_{\pm0.139}(0.770)$
$0.802_{\pm0.129}(0.797)$ | $0.005_{\pm0.000}(0.000)$
$0.813_{\pm0.143}(0.808)$
$0.835_{\pm0.118}(0.830)$
$0.845_{\pm0.109}(0.840)$ | $0.252_{\pm0.015}(0.280)$
$0.492_{\pm0.018}(0.296)$
$0.674_{\pm0.010}(0.218)$
$0.791_{\pm0.003}(0.145)$ | $1.000_{\pm0.000}$
$0.967_{\pm0.009}$
$0.962_{\pm0.010}$
$0.959_{\pm0.011}$ | $0.994_{\pm0.001}$
$0.965_{\pm0.004}$
$0.874_{\pm0.007}$
$0.693_{\pm0.011}$ |
| MUNBa
UA7.9%, UA$_{tf}$8.0%,
RA99.0%, TA86.1% | 0.05
0.1
0.15
0.2 | $1.000_{\pm0.000}(0.000)$
$0.945_{\pm0.027}(0.010)$
$0.907_{\pm0.013}(0.003)$
$0.802_{\pm0.025}(0.016)$ | $1.000_{\pm0.000}(0.000)$
$0.967_{\pm0.023}(0.007)$
$0.927_{\pm0.012}(0.033)$
$0.867_{\pm0.031}(0.007)$ | $0.949_{\pm0.001}(0.001)$
$0.899_{\pm0.001}(0.003)$
$0.850_{\pm0.001}(0.003)$
$0.803_{\pm0.001}(0.002)$ | $120.767_{\pm41.426}(79.233)$
$1.236_{\pm0.172}(191.646)$
$0.993_{\pm0.072}(185.798)$
$0.830_{\pm0.035}(170.221)$ | $120.287_{\pm43.180}(79.713)$
$1.160_{\pm0.140}(192.180)$
$0.993_{\pm0.058}(187.887)$
$0.880_{\pm0.040}(173.600)$ | $1.818_{\pm0.016}(0.034)$
$1.141_{\pm0.000}(0.004)$
$0.959_{\pm0.003}(0.002)$
$0.859_{\pm0.002}(0.000)$ | $0.009_{\pm0.004}(0.004)$
$0.773_{\pm0.092}(0.768)$
$0.916_{\pm0.051}(0.911)$
$0.966_{\pm0.014}(0.962)$ | $0.009_{\pm0.004}(0.004)$
$0.840_{\pm0.084}(0.835)$
$0.934_{\pm0.041}(0.929)$
$0.985_{\pm0.013}(0.980)$ | $0.522_{\pm0.004}(0.011)$
$0.788_{\pm0.002}(0.000)$
$0.887_{\pm0.002}(0.005)$
$0.935_{\pm0.001}(0.001)$ | $1.000_{\pm0.000}$
$0.883_{\pm0.097}$
$0.648_{\pm0.145}$
$0.239_{\pm0.077}$ | $0.986_{\pm0.000}$
$0.860_{\pm0.004}$
$0.525_{\pm0.008}$
$0.221_{\pm0.004}$ |
| Salun
UA100%, UA$_{tf}$100%,
RA98.4%, TA86.1% | 0.05
0.1
0.15
0.2 | $0.997_{\pm0.003}(0.003)$
$0.975_{\pm0.019}(0.039)$
$0.961_{\pm0.022}(0.057)$
$0.960_{\pm0.015}(0.173)$ | $0.993_{\pm0.012}(0.007)$
$0.927_{\pm0.023}(0.033)$
$0.860_{\pm0.040}(0.100)$
$0.840_{\pm0.020}(0.020)$ | $0.949_{\pm0.001}(0.001)$
$0.899_{\pm0.001}(0.003)$
$0.850_{\pm0.001}(0.004)$
$0.801_{\pm0.001}(0.004)$ | $199.599_{\pm0.207}(0.401)$
$191.973_{\pm1.616}(0.910)$
$187.825_{\pm3.461}(1.034)$
$184.838_{\pm3.478}(13.787)$ | $197.440_{\pm1.244}(2.560)$
$185.220_{\pm0.918}(8.120)$
$180.307_{\pm2.908}(8.573)$
$177.647_{\pm2.627}(3.167)$ | $1.980_{\pm0.050}(0.196)$
$1.169_{\pm0.002}(0.023)$
$0.969_{\pm0.002}(0.012)$
$0.863_{\pm0.004}(0.003)$ | $0.005_{\pm0.000}(0.000)$
$0.005_{\pm0.000}(0.000)$
$0.005_{\pm0.000}(0.000)$
$0.005_{\pm0.000}(0.001)$ | $0.005_{\pm0.000}(0.000)$
$0.005_{\pm0.000}(0.000)$
$0.005_{\pm0.000}(0.000)$
$0.005_{\pm0.000}(0.000)$ | $0.479_{\pm0.012}(0.053)$
$0.769_{\pm0.001}(0.019)$
$0.877_{\pm0.002}(0.015)$
$0.928_{\pm0.003}(0.008)$ | $1.000_{\pm0.000}$
$1.000_{\pm0.000}$
$1.000_{\pm0.000}$
$1.000_{\pm0.000}$ | $0.989_{\pm0.001}$
$0.884_{\pm0.001}$
$0.562_{\pm0.003}$
$0.230_{\pm0.009}$ |
| SFRon
UA100%, UA$_{tf}$100%,
RA96.1%, TA84.3% | 0.05
0.1
0.15
0.2 | $1.000_{\pm0.000}(0.000)$
$1.000_{\pm0.000}(0.064)$
$1.000_{\pm0.000}(0.096)$
$1.000_{\pm0.000}(0.213)$ | $1.000_{\pm0.000}(0.000)$
$1.000_{\pm0.000}(0.040)$
$1.000_{\pm0.000}(0.040)$
$1.000_{\pm0.000}(0.140)$ | $0.948_{\pm0.001}(0.002)$
$0.900_{\pm0.001}(0.003)$
$0.850_{\pm0.002}(0.003)$
$0.802_{\pm0.003}(0.003)$ | $200.000_{\pm0.000}(0.000)$
$200.000_{\pm0.000}(7.118)$
$200.000_{\pm0.000}(13.209)$
$200.000_{\pm0.000}(28.949)$ | $200.000_{\pm0.000}(0.000)$
$200.000_{\pm0.000}(6.660)$
$200.000_{\pm0.000}(11.120)$
$200.000_{\pm0.000}(25.520)$ | $2.264_{\pm0.254}(0.479)$
$1.266_{\pm0.044}(0.120)$
$1.009_{\pm0.012}(0.051)$
$0.886_{\pm0.006}(0.026)$ | $0.005_{\pm0.000}(0.000)$
$0.005_{\pm0.000}(0.000)$
$0.005_{\pm0.000}(0.000)$
$0.005_{\pm0.000}(0.000)$ | $0.005_{\pm0.000}(0.000)$
$0.005_{\pm0.000}(0.000)$
$0.005_{\pm0.000}(0.000)$
$0.005_{\pm0.000}(0.000)$ | $0.423_{\pm0.050}(0.110)$
$0.711_{\pm0.026}(0.077)$
$0.843_{\pm0.011}(0.049)$
$0.905_{\pm0.007}(0.031)$ | $1.000_{\pm0.000}$
$1.000_{\pm0.000}$
$1.000_{\pm0.000}$
$1.000_{\pm0.000}$ | $0.990_{\pm0.003}$
$0.912_{\pm0.017}$
$0.665_{\pm0.029}$
$0.358_{\pm0.017}$ |

*Table 19.* Performance of our unlearning framework. We show the unlearning performance on **CIFAR-10** with **ResNet-18** and **Tiny ImageNet** with **ViT** in **10% random data forgetting** scenario.

| Methods | α | | | $\lambda = 0.2$ | | | | | $\lambda = 0.5$ | | | | | $\lambda = 1$ | | |
|---|---|---|---|---|---|---|---|---|---|---|---|---|---|---|---|---|
| | | UA↑ | RA↑ | TA↑ | $CR_{\mathcal{D}_f}\downarrow$ | $CR_{\mathcal{D}_{test}}\uparrow$ | UA↑ | RA↑ | TA↑ | $CR_{\mathcal{D}_f}\downarrow$ | $CR_{\mathcal{D}_{test}}\uparrow$ | UA↑ | RA↑ | TA↑ | $CR_{\mathcal{D}_f}\downarrow$ | $CR_{\mathcal{D}_{test}}\uparrow$ |
| **CIFAR-10 with ResNet-18** | | | | | | | | | | | | | | | | |
| RT | 0.05 | 10.8%(2.2) | 98.3%(1.4) | 91.0%(0.8) | 0.788(0.076) | 0.824(0.055) | 14.0%(5.4) | 97.8%(1.9) | 90.4%(0.4) | 0.763(0.101) | 0.825(0.054) | 17.7%(9.1) | 96.8%(2.9) | 90.5%(1.3) | 0.719(0.145) | 0.820(0.059) |
| | 0.1 | | | | 0.914(0.029) | 0.924(0.021) | | | | 0.879(0.064) | 0.912(0.033) | | | | 0.838(0.105) | 0.911(0.034) |
| | 0.15 | | | | 0.956(0.019) | 0.959(0.009) | | | | 0.936(0.039) | 0.954(0.014) | | | | 0.906(0.069) | 0.951(0.017) |
| | 0.2 | | | | 0.977(0.011) | 0.976(0.005) | | | | 0.963(0.025) | 0.966(0.015) | | | | 0.932(0.056) | 0.965(0.016) |
| FT | 0.05 | 6.8%(1.8) | 97.0%(2.7) | 90.8%(1.0) | 0.844(0.020) | 0.829(0.050) | 7.9%(0.7) | 96.9%(2.8) | 90.9%(0.9) | 0.853(0.011) | 0.843(0.036) | 9.2%(0.6) | 97.9%(1.8) | 91.2%(0.6) | 0.835(0.029) | 0.854(0.025) |
| | 0.1 | | | | 0.948(0.005) | 0.924(0.021) | | | | 0.940(0.003) | 0.927(0.018) | | | | 0.938(0.005) | 0.936(0.009) |
| | 0.15 | | | | 0.983(0.008) | 0.959(0.009) | | | | 0.975(0.000) | 0.961(0.007) | | | | 0.976(0.001) | 0.970(0.002) |
| | 0.2 | | | | 0.989(0.001) | 0.974(0.007) | | | | 0.983(0.005) | 0.975(0.006) | | | | 0.986(0.002) | 0.984(0.003) |
| RL | 0.05 | 9.7%(1.1) | 96.6%(3.1) | 89.4%(2.4) | 0.709(0.155) | 0.736(0.143) | 9.9%(1.3) | 96.9%(2.8) | 89.7%(2.1) | 0.708(0.156) | 0.731(0.148) | 12.6%(4.0) | 95.3%(4.4) | 88.1%(3.7) | 0.629(0.235) | 0.669(0.210) |
| | 0.1 | | | | 0.896(0.047) | 0.887(0.058) | | | | 0.902(0.041) | 0.896(0.049) | | | | 0.845(0.098) | 0.858(0.087) |
| | 0.15 | | | | 0.946(0.029) | 0.931(0.037) | | | | 0.939(0.036) | 0.932(0.036) | | | | 0.911(0.064) | 0.913(0.055) |
| | 0.2 | | | | 0.964(0.024) | 0.949(0.032) | | | | 0.959(0.029) | 0.950(0.031) | | | | 0.936(0.052) | 0.938(0.043) |
| **Tiny ImageNet with ViT** | | | | | | | | | | | | | | | | |
| RT | 0.05 | 19.3%(4.6) | 98.8%(0.0) | 86.0%(0.0) | 0.458(0.045) | 0.516(0.000) | 26.4%(11.7) | 98.7%(0.1) | 85.8%(0.2) | 0.396(0.107) | 0.489(0.027) | 35.7%(21.0) | 98.6%(0.2) | 85.2%(0.8) | 0.346(0.157) | 0.481(0.035) |
| | 0.1 | | | | 0.729(0.046) | 0.786(0.000) | | | | 0.649(0.126) | 0.765(0.021) | | | | 0.549(0.226) | 0.739(0.047) |
| | 0.15 | | | | 0.841(0.039) | 0.889(0.000) | | | | 0.768(0.112) | 0.880(0.009) | | | | 0.658(0.222) | 0.861(0.028) |
| | 0.2 | | | | 0.898(0.036) | 0.932(0.003) | | | | 0.839(0.095) | 0.929(0.006) | | | | 0.743(0.191) | 0.918(0.017) |
| FT | 0.05 | 9.8%(4.9) | 97.4%(1.4) | 83.6%(2.4) | 0.441(0.062) | 0.399(0.117) | 13.6%(0.9) | 97.2%(1.6) | 83.6%(2.4) | 0.413(0.090) | 0.401(0.115) | 20.0%(5.3) | 96.4%(2.4) | 82.9%(3.1) | 0.342(0.161) | 0.363(0.153) |
| | 0.1 | | | | 0.753(0.022) | 0.683(0.103) | | | | 0.718(0.057) | 0.683(0.103) | | | | 0.627(0.148) | 0.652(0.134) |
| | 0.15 | | | | 0.884(0.004) | 0.823(0.066) | | | | 0.848(0.032) | 0.819(0.070) | | | | 0.772(0.108) | 0.802(0.087) |
| | 0.2 | | | | 0.942(0.008) | 0.893(0.042) | | | | 0.914(0.020) | 0.890(0.045) | | | | 0.856(0.078) | 0.877(0.058) |
| RL | 0.05 | 31.8%(17.1) | 95.3%(17.9) | 80.9%(5.1) | 0.051(0.452) | 0.111(0.405) | 36.2%(21.5) | 95.3%(3.5) | 80.4%(5.6) | 0.051(0.452) | 0.121(0.395) | 40.2%(25.5) | 94.5%(4.3) | 79.5%(6.5) | 0.048(0.455) | 0.119(0.397) |
| | 0.1 | | | | 0.278(0.497) | 0.451(0.335) | | | | 0.254(0.521) | 0.449(0.337) | | | | 0.236(0.539) | 0.436(0.350) |
| | 0.15 | | | | 0.579(0.301) | 0.710(0.179) | | | | 0.541(0.339) | 0.708(0.181) | | | | 0.480(0.400) | 0.673(0.216) |
| | 0.2 | | | | 0.752(0.182) | 0.825(0.110) | | | | 0.718(0.216) | 0.827(0.108) | | | | 0.642(0.292) | 0.793(0.142) |

