# OpenReview forum: "Tackling Fake Forgetting through Uncertainty Quantification"
_ICML.cc/2026/Conference — ICML 2026 regular_

### Official Review · Reviewer_SkLp · 2026-02-18

**Soundness:** 3
**Presentation:** 3
**Significance:** 3
**Originality:** 3
**Overall Recommendation:** 5
**Confidence:** 3

**Summary:**

This paper focuses on a novel perspective of uncertainty quantification on machine unlearning. Authors discover that utilizing the unlearning accuracy (UA) metric alone does not consider confidence level and yields fake forgetting, where the true label is included in the conformal prediction set. To quantify this uncertainty, this paper proposes a new metric, Conformal Ratio (CR), and integrates it with the standard unlearning objective for optimization. Evaluations are mainly conducted on CIFAR-10 and Tiny ImageNet to demonstrate the efficacy of the proposed method.

**Compliance With Llm Reviewing Policy:**

Affirmed.

**Final Justification:**

I appreciate the authors’ response on the connection between Grad-CAM and conformal prediction, the rationale of C\&W loss, and the additional results on the high-resolution datasets Oxford-Pets with a Swin Transformer. The rebuttal addressed my concerns, and I decide to raise the recommendation score.

**Key Questions For Authors:**

Please see the weakness.

**Limitations:**

yes

**Strengths And Weaknesses:**

## Strength

1. The observation on uncertainty inadequacy is interesting and of practical significance. I agree that if real labels are included in the conformal prediction set, the forgotten data is likely to be easily recovered.

2. The introduced Conformal Ratio metric is technically solid in design.

3. This paper is generally well-written.


## Weakness

1. Some concerns about Figure 1.

    - Although the proposed CPU framework shifts the activation away from the object's key features, the attention appears to concentrate on other seemingly irrelevant regions (e.g., the food in the wok or a person’s lower leg). Could the authors provide an explanation for this activation shift phenomenon?

    - The authors state that conformal prediction is used to demonstrate that unlearning accuracy is insufficient for evaluating forgetting reliability, noting that more than 50% of misclassified samples still appear in the conformal prediction set. However, Figure 1 primarily presents Grad-CAM visualizations as qualitative evidence. The connection between the Grad-CAM maps and the claimed verification via conformal prediction (e.g., the corresponding non-conformity scores) is unclear. Could the authors clarify how these two analyses are related?

2. What is the rationale for drawing insight from the C&W loss? This paper does not provide a sufficient explanation.

3. According to Tables 3 and 5, several recent unlearning methods already demonstrate a certain degree of robustness under conformal prediction, despite not being explicitly optimized for this objective. For instance, for some methods, fewer than 10% of misclassified samples can be recovered by conformal prediction. This observation appears to weaken the necessity of the proposed CR metric.

4. Some statements are given without context and may introduce ambiguity. For example, in my understanding, "*This emphasizes that the non-conformity scores of ground truth labels should be pushed beyond the conformal prediction threshold $\hat{q}$*" in line 231 refers to the forget set, not the retain set, but no contexts are given.

5. Experiments are not sufficient enough, which are conducted only on CIFAR-10 and tiny-imagenet, lacking evaluations on other widely used datasets like CIFAR-100 and high-resolution datasets like Oxford Pets. Moreover, more network architectures besides ResNet-18 and ViT are encouraged, such as VGG, Swin Transformer, etc.

---

> ### Author Rebuttal · Authors · 2026-03-30
>
> # Response to Reviewer SkLp
>
> Thank you for recognizing the contributions of our work. We appreciate your constructive feedback and address your concerns below.
>
> > **W1: Grad-CAM visualization**
> >
>
> **Concern 1: Activation shift to irrelevant regions.**
>
> This is expected and actually indicates successful forgetting.
>
> Mechanistically, Grad-CAM highlights regions that contribute most to the predicted class score. When the model has not truly forgotten (i.e., fake forgetting), the true label still retains relatively high probability, and the gradients remain aligned with class-discriminative features, leading to focused activations on the object.
>
> In contrast, under our CPU framework, the true label's probability is explicitly suppressed below the conformal threshold. As a result, the model no longer relies on class-specific features, and the gradient signal becomes diffuse or shifts toward non-discriminative regions.
>
> Therefore, the observed activation shift reflects the loss of class-specific feature alignment, which is precisely the intended effect of successful unlearning.
>
> **Concern 2: The connection between Grad-CAM and conformal prediction.**
>
> Grad-CAM serves as qualitative evidence of fake forgetting, while in-set ratios in Table 2 provide quantitative support. The two are connected through predicted probabilities: when fake forgetting occurs, the true label retains high softmax probability, keeping it in the prediction set and causing Grad-CAM to focus on discriminative features. When CPU successfully removes the true label from the prediction set, its probability drops and Grad-CAM activations shift away from key features accordingly. The two analyses thus jointly demonstrate the same underlying phenomenon from complementary perspectives.
>
> > **W2: rationale of C&W loss**
> >
>
> The adoption of C&W loss is motivated by its natural structural alignment with our unlearning objective. The original C&W loss is designed to ensure that the model's predicted probability for the true label $p_t(x)$ falls below the highest competing class probability $\max_{i\neq t} p_i(x)$, which aligns well with the unlearning goal of suppressing the model's confidence in the true label of forget data.
>
> However, C&W loss alone only guarantees misclassification — it does not ensure the true label is excluded from the conformal prediction set. Our CPU loss (Eq. 8) is a natural extension: we replace $\max_{i\neq t} p_i(x)$ with the conformal threshold $\bar{q}$, and $p_t(x)$ with the non-conformity score $S(x, y_t)$. This directly optimizes for the true label's exclusion from the prediction set, upgrading the guarantee from mere misclassification to reliable forgetting under our CR metric. The C&W loss thus serves as a principled and structurally compatible starting point for our framework design.
>
> > **W3: For some methods, fewer than 10% of misclassified samples can be recovered by conformal prediction**
> >
>
> **Explanation for the example**
>
> We clarify that Table 5 is intended as a preliminary exploration of the fake forgetting phenomenon for UA metric, illustrating how conformal prediction can recover data that UA considers forgotten. The low in-set ratios (<10%) of GA, Teacher, and SSD in Table 5 are simply a consequence of their low misclassification counts. When we evaluate these methods using our proposed CR metric in Table 3, GA, Teacher, and SSD all perform poorly.
>
> **The different purposes of in-set ratio and CR**
>
> The key point is that cited examples (e.g., low in-set ratios) do **not negate the need for CR** but expose the limits of partial diagnostics. Low in-set ratios in Table 5 mainly result from these methods producing **few misclassified samples**, not strong forgetting; evaluated over the full dataset via CR (Table 3), they (e.g., GA, Teacher, SSD) show **poor forgetting quality**. The in-set ratio is a **conditional diagnostic on misclassified samples**, whereas CR evaluates **all forget data**, capturing both insufficient forgetting (low UA) and fake forgetting (high coverage). Thus, CR offers a **unified, stricter evaluation**, while the in-set ratio shows only one aspect, explaining why CR remains necessary even if some methods seem robust under partial analysis.
>
> > **W4: Some statements are given without context and may introduce ambiguity.**
> >
>
> Thank you for pointing this out. The statement indeed refers to the forget set D_f, and we will make this explicit in the revision.
>
> > **W5: More architecture and datasets**
> >
>
> We note that CIFAR-100 results are already in Table 9 (Appendix G) for the subclass-wise forgetting scenario (i.e., CIFAR-20). Additional experiments on Oxford-Pets with a Swin Transformer (Table 19, Supplementary Material: https://anonymous.4open.science/r/CPU-54CC/material.pdf) show that CPU consistently outperforms FT, NegGrad+, and Salun, demonstrating the CPU generalizability across datasets and architectures.

---

> > ### Author Rebuttal · Reviewer_SkLp · 2026-03-31
> >
> > I appreciate the authors’ response, which has addressed my concerns. I am willing to increase my score, and I look forward to seeing the corresponding updates in the revised version of the paper.

---

> > > ### Author Response · Authors · 2026-03-31
> > >
> > > Thank you again for your constructive feedback and for reconsidering the score. We will incorporate the corresponding updates in the revised version of the paper.
> > >
> > > The Authors

---

### Official Review · Reviewer_T91b · 2026-03-13

**Soundness:** 2
**Presentation:** 2
**Significance:** 3
**Originality:** 2
**Overall Recommendation:** 4
**Confidence:** 4

**Summary:**

This paper tackles the issue of fake forgetting in machine unlearning. Many unlearning approaches simply try to make the ML model misclassify data in the forget set without considering that uncertainty measures computed on those examples might act as a side-channel for identifying previously trained-on examples (and therefore retention patterns in the model). The paper proposes to use a conformal prediction measure to determine the degree of fake forgetting in models. Building on top of that metric, the paper goes on and proposes a new unlearning technique based on conformal prediction and the Carlini & Wagner attack to remove the true label from the prediction set. The paper then shows the effectiveness of the proposed unlearning and evaluation technique on multiple data sets and with varying ablations.

**Compliance With Llm Reviewing Policy:**

Affirmed.

**Final Justification:**

I thank the authors for their responses to my concerns. I think the paper has improved during the rebuttal. I encourage the authors to make sure that the proposed improvements are incorporated into the final paper draft. As it pertains to Subsection 3.3.2, I would strongly recommend to include a forward pointer at the end of 3.1 (potentially bundled with point 3) or at the start of 3.2. The right column of page 3 already introduces a bunch of results that are hard to interpret without having a feeling for what $\alpha$ looks like.

I have updated my recommendation to a weak accept. Since proposed changes are not verifiable, I am however unable to strongly advocate for the paper. I have faith that the authors will do a good job with this.

**Key Questions For Authors:**

1. Can you comment on the interplay between the unlearning method and the evaluation metrics? Is there a dependence between the two?
2. The paper basically argues that confidence can be a side-channel to understand if a data point was truly unlearned or not. But in many cases all the user might see is a hard-thresholded black box prediction without a confidence or softmax score. How would you deal with that setup?
3. Can you comment on the positioning of the work in the data removal vs privacy space?

**Limitations:**

The paper does not include an explicit acknowledgment of current limitations. However, as described above, there definitely exist some flaws in the soundness and presentation of this work.

**Strengths And Weaknesses:**

My biggest concern with the paper are related to soundness and presentation.

### __Soundness__:
- I am skeptical of statements along the lines of: Inclusion of a class in the conformal prediction set is a sign of fake forgetting. This is especially true since the paper assumes $\alpha=0.05$ in general. This means that 95% coverage is desired which, depending on the confidence distribution can result in a large set that might contain almost all classes. I think more nuance is needed when making such statements.
- Unlearning accuracy (i.e. accuracy on the unlearn set) seems to be an obviously flawed metric. I found it very surprising that the paper suggests that this is the standard way unlearning is evaluated. To the best of my understanding, membership inference attacks, distributional distances to untrained models, and retrain-from-scratch appear to be much superior in terms of measuring the goal of unlearning.
- In fact, I am not sure that what the paper suggests in terms of conformal prediction machinery is even necessary. It seems like regularizing the output distribuition of data points in the forget set towards a uniform distribution (or the marginal label distribution in case classes are not balanced) while at the same time continuing to maximize predictive performance on the retain set gives the desired outcome of unlearning. Performance evaluation would then simply consider how close the retain set matches that marginal distribution while still performing well on the test set. Since the goal in unlearning is the make the forget set give the same outputs as data on which we never trained on, why not directly optimize for that output distribution? It seems like that would side-step the problem of just wanting to misclassify the data (which I iterate again is obviously flawed). See comments below on originality.
- I wonder whether the paper is using the fact that a conformal prediction based loss is used for unlearning and then it is evaluated using a conformal prediction based evaluation metric. It might be the fact that the approach is essentially gaming the evaluation as a result since it is evaluated on a property that it is explicitly optimized for. I do see that the proposed metric also works reasonably well on other evaluation metrics, but this still leaves a bit of a bitter taste in my mouth.

### __Presentation__:
- The method section mixes some methodological proposals with well established results and formalism, especially related to CP. This makes it hard to understand where the key new contribution of the paper lies.
- Subsection 3.3.2 appears too late in the paper. By that point, I already asked myself multiple times how $\alpha$ affects the proposed methods. It would be better pro-actively address the dependence on the confidence level and how this interplays with the proposed approach.
- It is also a bit unclear to me how the paper fits into the broader privacy community. It is not necessarily clear based on how the work is positioned whether the goal is to remove bias from the model or to enable user privacy.
- The authors use some clear vspace tricks to make the paper fit in the page limit. This is not a good look.
- Results are only shown as aggregate performance numbers. It would have been great to see a few more in-depth examples of interesting differences between unlearning methods (and evaluation metrics) in line of what Figure 1 shows. This seems like a missed opportunity.

### __Significance__:
- Machine unlearning has suffered for a while from misleading metrics and evaluation techniques [1]. As such, accounting for the fact that an uncertainty side channel can reveal additional information about data points is an important topic to study.
- While the paper raises awareness to this issue, I am still skeptical that a somewhat natural application of conformal prediction pushed the field forward in a significant manner.
- The paper performs a large amount of experimentation on different baselines and evaluation metrics, including plenty of ablations. This experimental coverage is commendable.

### __Originality__:
- I don't think that there are other significant works on approaching the issue of fake unlearning from an uncertainty point of view so there is definitely some degree of novelty here.
- Still, to the best of my understanding, there exists prior work that has already moved away from evaluating accuracy on the forget set. Moreover, other works has also considered the role of uncertainty in unlearning [4], especially regularizing outputs to a particular distribution [2,3,5]. Especially LoTUS and Bad Teacher, which the paper already cites, should be highlighted in that regard.

__References__

[1]: Cooper, A. Feder, et al. "Machine Unlearning Doesn't Do What You Think: Lessons for Generative AI Policy and Research." arXiv preprint arXiv:2412.06966 (2024).

[2]: Chien, Eli, et al. "Langevin unlearning: A new perspective of noisy gradient descent for machine unlearning." Advances in neural information processing systems 37 (2024): 79666-79703.

[3]: Rabanser, Stephan, et al. "Confidential guardian: Cryptographically prohibiting the abuse of model abstention." arXiv preprint arXiv:2505.23968 (2025).

[4]: Graves, Laura, Vineel Nagisetty, and Vijay Ganesh. "Amnesiac machine learning." Proceedings of the AAAI conference on artificial intelligence. Vol. 35. No. 13. 2021.

[5]: Scholten, Yan, Stephan Günnemann, and Leo Schwinn. "A probabilistic perspective on unlearning and alignment for large language models." arXiv preprint arXiv:2410.03523 (2024).

---

> ### Author Rebuttal · Authors · 2026-03-28
>
> # Response to Reviewer T91b
>
> We really appreciate your feedback and address your concerns as follows.
>
> > **Soundness1: Prediction set size concern.**
>
> As shown in the Set Size columns of Table 3, average set sizes are around 1 for CIFAR-10 and 2 for Tiny ImageNet, far smaller than the total class count. This empirically demonstrates that the 95% coverage requirement does not lead to trivially large prediction sets.
>
> More importantly, our CR metric is  explicitly designed to account for this concern by jointly considering both **Coverage and Set Size**. A method that achieves low Coverage only by inflating Set Size (i.e., high uncertainty) will not be rewarded by CR, as the denominator increases accordingly. This design ensures CR reflects genuine forgetting quality rather than mere prediction uncertainty.
>
> > **Soundness2: UA concern.**
>
> “UA is an imperfect metric” is exactly the starting point of our paper. We identify a concrete and **previously overlooked failure mode of UA**, namely fake forgetting, and propose CR as a more reliable complement.
>
> We also note that MIA, distributional distances, and retrain-from-scratch metrics are already included in our evaluation. However, none of these metrics directly captures whether the true label remains recoverable under uncertainty quantification, which is the specific gap our CR metric addresses. In this sense, CR is an important addition that reveals a blind spot they all share.
>
> We further show in Table 17 in https://anonymous.4open.science/r/CPU-54CC/material.pdf that fake forgetting also appears under MIA, supporting the broader value of our uncertainty-based perspective. The key insight in our work not only reveals limitations in UA, but can also be extended to other evaluation metrics.
>
> > **Soundness3 & Originality: uniform distribution regularization**
>
> CPU differs fundamentally from uniform distribution regularization: rather than matching a predefined target distribution, CPU enforces a data-adaptive, calibrated threshold derived from unseen calibration data, explicitly targeting exclusion of the true label from the conformal prediction set. Importantly, the reviewer's suggestion about the uniform distribution method is closely related to LoTUS [1], which we include as a baseline. As shown in Table 3, LoTUS still exhibits notable fake forgetting under CR, demonstrating that uniform entropy regularization is insufficient. This further motivates the necessity of our CPU.
>
> > **Soundness4: Circular evaluation**
>
> We respectfully note that proposing a metric to identify a problem and then designing a method to address it is a standard and well-accepted research paradigm, also followed by prior unlearning works [1,2], which evaluate their proposed methods under their own metrics. Our workflow follows the same logic: (1) identify a failure mode (fake forgetting), (2) design a metric to quantify it (CR), and (3) develop a method to address it (CPU).
>
> Crucially, CPU is not evaluated solely on CR. As shown in Table 4, it consistently improves across multiple independent metrics, including UA, RA, TA, and MIA, as well as qualitative analyses such as Grad-CAM and output distributions.
>
> **We therefore argue that the effectiveness of CPU is well-supported by comprehensive cross-metric evidence, and should not be dismissed simply because it shares a similar underlying idea with CR.**
>
> > **Presentation.**
>
> Thanks for your suggestions.
>
> - Section 3.1 covers CP preliminaries; remaining Section 3 proposes CR; Section 4 proposes CPU. We believe this clearly separates background from contributions.
> - Placing Section 3.3.2 after CR is defined is intentional, introducing confidence level before CR would confuse readers unfamiliar with the metric. We note Reviewer mJGj even suggests moving it further to the experiments section.
> - We will clarify the positioning of our work in both privacy (data removal) and bias mitigation contexts.
> - Thanks for pointing out this vspace issue and we will fix it.
> - We will add more qualitative case studies in the revision.
>
> > **Significance.**
>
> We respectfully believe that the significance of our work lies not in the technical complexity of CP itself, but in the following contributions:
>
> - identifying and formalizing fake forgetting from uncertainty perspective, a concrete blind spot in UA;
> - proposing CR, a principled metric capturing forgetting quality beyond existing metrics;
> - proposing CPU, a general plug-in framework.
>
> **We believe identifying an important problem, measuring it reliably, and solving it together constitute a significant contribution regardless of the simplicity of individual components.**
>
> > **Originality**
>
> Thank you for recognizing the novelty of our work. We will provide a more detailed discussion of these cited works.
>
> ---
>
> **Reference**
>
> [1] Spartalis et al. Lotus: Large-scale machine unlearning with a taste of uncertainty
>
> [2] Jeon et al. An information theoretic evaluation metric for strong unlearning

---

> > ### Author Rebuttal · Reviewer_T91b · 2026-04-03
> >
> > I thank the authors for their response. It has addressed some but not all of my concerns.
> >
> > I think I am coming around to the idea that the paper has value in terms of raising awareness on evaluation issues of unlearning and how uncertainty can be one way of achieving that goal. As a result, I have updated my score. But there are still some outstanding issues that prevent me from recommending the paper for acceptance.
> >
> > On the prediction set size: if many prediction sets only include a single example, then misclassification accuracy on the forget set should be sufficient, right? There is no way for the true class to still leak since the goal is to misclassify the top class. Unless I am missing something obvious then the conformal prediction machinery does not really add anything useful in these cases.
> >
> > Also, while I understand that this was hard to do with the 5000 char limit, I did not see my explicit questions answered in the rebuttal.
> >
> > Finally, it seems like the authors are not generally in favor of many of my suggestions on improving the presentation of the paper. I still maintain that the presentation of this work can be improved.

---

> > > ### Author Response · Authors · 2026-04-03
> > >
> > > We sincerely thank the reviewer for the follow-up and for reconsidering the score. We are glad that our contribution is becoming clearer, and we address the remaining concerns below.
> > >
> > > > **Prediction set equals 1**
> > >
> > > **Assuming prediction set size of 1 for forget samples does not hold consistently**. Conformal prediction produces sample-dependent set sizes that grow with uncertainty. Even if the average set size is around 1, individual points can vary. **The set size cannot be manually controlled.** This uncertainty-adaptive behavior adds value beyond misclassification accuracy: misclassified points with set size >1 may still include the ground truth, which UA cannot detect as shown in Table 20 in our **link** (https://anonymous.4open.science/r/CPU-54CC/material.pdf). Our CR metric captures this by jointly considering Coverage and Set Size, accounting for such variations across samples and methods.
> > >
> > > ---
> > >
> > > We thank the reviewer for the questions and apologize for not addressing them clearly in the initial rebuttal due to space limitations. We provide detailed responses below.
> > >
> > > > **Q1: interplay between CR and CPU**
> > >
> > > While CPU and CR are conceptually aligned, they are not directly coupled. CPU increases the non-conformity score of the ground-truth label beyond the threshold to exclude it from the conformal set, without explicitly optimizing Coverage or Set Size. CR, however, is computed from Coverage and Set Size. Thus, CR improvements after CPU occur naturally, rather than from directly optimizing the metric.
> > >
> > > Beyond this distinction, proposing a metric to identify a problem and then designing a method to address it is a standard paradigm [1,2]. Importantly, CPU’s improvements are consistently reflected across other metrics (UA, RA, TA, MIA), suggesting that the gains are not specific to CR. Therefore, we consider CPU and CR to be aligned but not dependent.
> > >
> > > > **Q2: blackbox assumption**
> > >
> > > Machine unlearning is typically performed by the model owner, who has full access to internal outputs such as softmax probabilities. Thus, a strict black-box setting with only hard predictions is not a realistic constraint. Moreover, without access to confidence scores, widely used evaluation methods (e.g., MIA, LiRA, and output distribution analysis) would also be inapplicable, indicating that assuming access to confidence is reasonable in practice.
> > >
> > > > **Q3: the positioning of the work in the data removal vs privacy space**
> > >
> > > We clarify how our work relates to both **bias removal vs user privacy** and **data removal vs privacy space**.
> > >
> > > **bias removal vs. user privacy**
> > >
> > > Our work is applicable to both scenarios, as reflected in the 2 evaluation criteria in Section 3.3.1.
> > >
> > > Criterion ➊ (Gap to R_T) targets user privacy, aiming for the unlearned model to behave as if the forget data was never seen, with RT as the gold-standard baseline.
> > >
> > > Criterion ➋ (Limit-Based) targets **bias removal**, following [3], which includes removing biases, outdated, or incorrect information. Here, a lower CR or higher UA on D_f is desirable.
> > >
> > > **data removal vs. privacy space**
> > >
> > > Separately, one can view unlearning from (1) a behavioral data removal perspective (does the model’s output match retraining?), and (2) a privacy perspective (does the model still leak information about the forgotten data, e.g., via MIA), which we include in our experimental results in Tables 3 and 4.
> > >
> > > Our work connects these perspectives through the notion of label recoverability under uncertainty: if the true label remains within the prediction set, this indicates both incomplete behavioral removal and potential information leakage.
> > >
> > > ---
> > >
> > > > **presentation suggestions**
> > >
> > > We sincerely appreciate the reviewer's suggestions and wish to clarify that we are willing to improve the presentation of our work. There are only two suggestions where we would like to further discuss our rationale.
> > >
> > > **Mixing some methodological proposals … makes it hard...**
> > >
> > > We wish to clarify that only Section 3.1 covers standard CP preliminaries as background. The rest of Section 3 focuses on fake forgetting and our proposed metric CR, while Section 4 presents the adaptation of the C&W loss to unlearning and its integration with CP. Thus, the key contributions span Sections 3 and 4. But we are happy to make this distinction more visually explicit in the revision.
> > >
> > > **Subsection 3.3.2 appears too late**
> > >
> > > Introducing the ablation of confidence level and calibration set size before defining CR in Section 3.3.1 would leave readers without the necessary context. Since these factors pertain to CP unlearning and CR, the ablation is meaningful only after CR is formally defined. We welcome further suggestions if the reviewer has alternative organization.
> > >
> > > ---
> > >
> > > **References**
> > >
> > > [1] Spartalis et al. Lotus: Large-scale machine unlearning with a taste of uncertainty
> > >
> > > [2] Jeon et al. An information theoretic evaluation metric for strong unlearning
> > >
> > > [3] Kurmanji et al. Towards unbounded machine unlearning

---

### Official Review · Reviewer_YzP3 · 2026-03-13

**Soundness:** 4
**Presentation:** 3
**Significance:** 4
**Originality:** 4
**Overall Recommendation:** 5
**Confidence:** 3

**Summary:**

The paper identifies "fake forgetting" in machine unlearning: examples misclassified by unlearning accuracy may still retain their true label in a conformal prediction set. To address this, the authors propose the CR metric and the CPU framework, and show empirically that UA can overestimate forgetting quality while CPU improves forgetting behavior on image classification benchmarks.

**Compliance With Llm Reviewing Policy:**

Affirmed.

**Final Justification:**

The rebuttal fully addressed my concerns. I maintain the Accept recommendation accordingly.

**Key Questions For Authors:**

1. Practical overhead of CPU. Since CPU augments training-based unlearning methods, can the authors provide a summary of the additional computational overhead and implementation complexity compared with the underlying baseline methods? This would help readers assess the practical tradeoff between improved forgetting quality and extra cost.
2. Generality beyond image classification. The paper focuses on image classification, which is a reasonable first step. Do the authors expect the fake-forgetting phenomenon and the CR/CPU framework to transfer directly to other modalities or tasks, and if so, what parts would likely need modification?
3. Reproducibility. The submission states that code is available, but the provided anonymous repository was inaccessible at review time. Can the authors restore the repository and clarify what code, configs, and scripts will be released?

**Limitations:**

Yes

**Strengths And Weaknesses:**

Strengths: important and well-motivated problem, novel use of conformal prediction for unlearning evaluation, useful method beyond diagnosis, and solid empirical evidence. Weaknesses: practical overhead of CPU deserves clarification, the code repository was inaccessible, and some abbreviations should be spelled out more consistently at first use. Overall I find the contribution significant and technically solid.

---

> ### Author Rebuttal · Authors · 2026-03-28
>
> # Response to Reviewer YzP3
>
> Thank you for acknowledging the contributions of our work. We sincerely appreciate your feedback and provide our detailed responses below.
>
> > **W3: Some abbreviations should be spelled out more consistently at first use**
> >
>
> Thank you for pointing this out. We have carefully proofread the paper and will incorporate the corrections in the revised version.
>
> > **Q1 & W1: Practical overhead of CPU**
> >
>
> We have provided a detailed runtime comparison in Appendix E.2. As shown in Table 6, CPU introduces only negligible overhead. For example, on the FT baseline, training time increases from 6.3s to 6.8s (<8% increase). This is because the additional computation in CPU consists only of a lightweight non-conformity score and quantile update, both of which are computationally inexpensive.
>
> > **Q2: Generality beyond image classification**
> >
>
> We thank the reviewer for this insightful question about broader applicability. While our current work focuses on establishing the conceptual framework and methodology for image classification, we believe the core principles can indeed extend to other domains:
>
> **For Image Generation (e.g., Diffusion Models)**
>
> Instead of class labels, conformal prediction can be applied to latent feature spaces or generation quality metrics. For example, when unlearning a specific concept (e.g., a person's face), we can construct conformal sets in CLIP embedding space. Fake forgetting would manifest as the forgotten concept's features still falling within high-confidence conformal regions.
>
> **For Large Language Models**
>
> Our framework can be adapted to **token-level conformal prediction** during sequence generation. For factual unlearning (e.g., forgetting specific knowledge), we can examine whether tokens related to forgotten facts remain in high-probability conformal sets. Metrics like perplexity combined with conformal intervals can indicate fake forgetting: truly forgotten knowledge should exhibit both high perplexity and high uncertainty.
>
> **Unified Principle**
>
> The core insight of our work, that models may appear to forget through misclassification while retaining confidence in ground truth, applies across modalities. The key is adapting the notion of "prediction set" to domain-specific output spaces.
>
> **Overall, we view these extensions as exciting directions for future work, building upon the key idea established in this paper.**
>
> > **Q3 & W2: Reproducibility**
> >
>
> Thank you for pointing this out. We have restored the visibility of the repository. The code is available in the original repository mentioned in our paper.

---

> > ### Author Rebuttal · Reviewer_YzP3 · 2026-04-01
> >
> > I appreciate the authors' comprehensive rebuttal; they have fully addressed all my comments. I also agree that the suggested future work represents an exciting and promising direction for the field.

---

> > > ### Author Response · Authors · 2026-04-01
> > >
> > > We are sincerely grateful to the reviewer for the highly positive feedback and the meticulous time and effort dedicated to evaluating our work. We also appreciate the reviewer’s insightful suggestions for future work outlined in the comments, which we believe will be valuable to the unlearning community.

---

### Official Review · Reviewer_mJGj · 2026-03-13

**Soundness:** 3
**Presentation:** 3
**Significance:** 3
**Originality:** 3
**Overall Recommendation:** 5
**Confidence:** 4

**Summary:**

This paper identifies an interesting limitation of the standard unlearning accuracy (UA) metric — namely that misclassifying forget data does not guarantee true forgetting, as the true label may still appear in the conformal prediction set, a phenomenon the authors term "fake forgetting." To address this, the authors propose a new metric (CR) inspired by conformal prediction, as well as an unlearning framework (CPU) that incorporates a modified C&W loss to explicitly push true labels out of the conformal prediction set.

**Compliance With Llm Reviewing Policy:**

Affirmed.

**Final Justification:**

Thank you for the additional explanation! Your response has addressed my questions. I have changed my score to Accept. Congratulations!

**Key Questions For Authors:**

please see above

**Limitations:**

It would be interesting to hear from the authors what they think the main limitation of this method is.

**Strengths And Weaknesses:**

**Strengths**:

1. The observation that misclassification does not imply true forgetting is genuine and well-motivated. The empirical finding that over 50% of UA-misclassified points remain recoverable via conformal prediction is striking and I found it quite convincing.

2. The breadth of the empirical evaluation is commendable — 12 unlearning methods across multiple datasets, architectures, and forgetting scenarios is a thorough sweep.

3. The idea of incorporating conformal prediction thresholds into the unlearning loss is an interesting design choice, and the negligible computational overhead makes it practically appealing.


**Weaknesses**:

1. The CR metric is an interesting idea, but there is a fundamental theoretical concern worth addressing. As the authors themselves acknowledge, the exchangeability property required for conformal prediction to provide meaningful coverage guarantees may not always hold in the machine unlearning setting which can make it a less reliable metric — is this a fair reading, or am I missing something? If my understanding is correct, then arguably the most valuable contribution of this paper is not the metric itself, but rather the insight that an unlearning loss inspired by conformal prediction can meaningfully improve forgetting quality. However, the evaluation of the CPU framework is limited in two important ways. First, results are only reported for three of the most basic unlearning methods (RT, FT, and RL), leaving it unclear whether the framework generalizes to the stronger and more competitive methods evaluated elsewhere in the paper. Second, evaluating CPU using CR is somewhat circular, as the framework is explicitly optimized to push true labels out of the conformal prediction set — the same thing CR measures. The results will naturally look favorable, but this limits how much we can read into them.
Therefore, to meaningfully evaluate the proposed unlearning framework, I found myself having to compare against the oracle using traditional metrics. This however left me genuinely puzzled: for CPU-RT specifically, λ=0 is RT itself — the true gold standard that never saw the forget data. Yet CPU-RT achieves UA of 10.8%, 14.0%, and 17.7% for λ=0.2, 0.5, and 1.0 respectively, compared to RT's 8.6%. Since the goal of unlearning is to match RT rather than exceed it, this warrants a much clearer explanation from the authors.

2. The requirement of a held-out calibration dataset adds an additional practical constraint that may be restrictive in real-world unlearning scenarios.

3. I am struggling to understand the claim that "RT is no longer a gold-standard baseline" under the CR metric. Retrain-from-scratch is the oracle by definition — it should always serve as the gold standard regardless of which metric is used. From what I can tell, the gap relative to RT is still reported in the tables under CR, so it is unclear what the authors mean by this statement. A clearer explanation would be appreciated.

**Minor Comments:**

1. The goal of unlearning is not to maximize UA or minimize MIA — it is to match the behavior of the oracle, i.e., achieve UA and MIA close to those of RT (with MIA ideally around 0.5). The arrows in the tables indicating "larger is better" or "smaller is better" for these metrics are therefore misleading and should be corrected accordingly.

2. I would recommend some restructuring of the paper. Specifically, adding a dedicated ablation studies section within the experiments and moving the calibration set size analysis there would improve the overall flow and clarity of the paper.

3. Throughout the paper, "forgetting data" should be replaced with "forget data." Additionally, there are several grammatical mistakes throughout that should be addressed in a careful proofread.

---

> ### Author Rebuttal · Authors · 2026-03-30
>
> # Response to Reviewer **mJGj**
>
> We sincerely appreciate your feedback, and our detailed responses are provided below.
>
> > **W1**
> >
>
> **Exchangeability**
>
> The reviewer's reading is correct, and we acknowledge this in the paper. However. our use of CR is not to claim distribution-free coverage guarantees for every unlearned model. Rather, CR is a practical evaluation tool. In the RT baseline, the calibration data and forget/test examples are drawn from the same post-unlearning data world, thus exchangeability is well-motivated for RT.  CR is then used to measure how closely a given unlearning method matches this gold-standard RT. Since all methods are evaluated under the same setting, CR remains a meaningful and fair relative comparison metric even when exchangeability does not strictly hold.
>
> **CPU with other baselines**
>
> We conducted additional experiments applying CPU to more baselines, and the results, presented in Table 18 in https://anonymous.4open.science/r/CPU-54CC/material.pdf, consistently show that CPU enhances forgetting quality across all methods.
>
> **Circular evaluation concern**
>
> Please kindly refer to our response to **Soundness 4** for Reviewer T91b.
>
> **CPU-RT exceeding RT’s UA.**
>
> Our paper considers two evaluation criteria [1] (Section 3.3.1) corresponding to two unlearning scenarios (Section 3.3.1): Criterion ➊ (Gap to RT) for user privacy, and Criterion ➋ (Limit-Based) for bias removal. Under Criterion ➊, RT is a gold standard baseline. Under Criterion ➋, a higher UA on the forget set is desirable, thus the results are reasonable.
>
> For Criterion ➊, the question of how to choose hyperparameter λ of CPU to best approximate RT has already been addressed in prior work [7], which proposes adaptively adjusting hyperparameter during training. This strategy can be straightforwardly extended to CPU. As this adaptive approach lies outside the primary scope of our work, we do not elaborate on it further.
>
> > **W2: The requirement of a held-out calibration dataset**
>
> **It is a worthwhile trade-off.** The calibration set is the key enabler for introducing uncertainty quantification into machine unlearning, which provides a more reliable assessment of forgetting quality and leads to improved unlearning performance. We believe this small additional requirement is well justified by the gains it brings.
>
> **It is common in current works.** A held-out validation set is widely used in prior unlearning works [1,2], so this requirement is not a novel burden introduced by our method.
>
> **It is a reasonable assumption in practice.** Since the unlearning process is controlled by the data owner, it is reasonable to assume access to a small held-out validation set.
>
> > **W3: RT is no longer a gold-standard baseline**
>
> The statement applies **only to our Evaluation Criterion ➋ (Limit-Based)**, not to **Criterion** ➊. As noted in Section 3.3.1 and Footnote 2, the two criteria correspond to two distinct scenarios following [1]:
>
> - **Criterion ➊ (Gap to RT)** targets the **user privacy scenario**, where the goal is to make the unlearned model behave as if the forget data was never seen — here, RT remains the gold-standard oracle, and a smaller gap to RT is better.
> - **Criterion ➋ (Limit-Based)** targets the **bias removal scenario**, where the goal is to remove the influence of certain data as completely as possible. In this case, RT — which simply retrains on the remaining data — does not necessarily achieve the strongest forgetting, and is therefore not the gold standard. A lower CR on Df is directly desirable regardless of RT's value.
>
> > **W4: Minor comments**
>
> **Arrows in tables**: As explained in our responses to W3 above, the arrows are intentionally kept to facilitate evaluation under Criterion ➋.
>
> **Restructuring of the paper**: We appreciate the suggestion and will restructure the paper accordingly in the revision.
>
> **forgetting data → forget data**: We use the term "forgetting data" following several prior works [3-7]. Nevertheless, to avoid any potential confusion, we will replace it with "forget data" throughout the paper. We will also proofread the paper and address all grammatical issues in the revision.
>
> ---
>
> **References**
>
> [1] Kurmanji et al. Towards unbounded machine unlearning. NeurIPS 2023
>
> [2] Spartalis et al. Lotus: Large-scale machine unlearning with a taste of uncertainty. CVPR 2025
>
> [3] Fan et al. Salun: Empowering machine unlearning via gradient-based weight saliency in both image classification and generation. ICLR 2024
>
> [4] Jia et al. Model sparsity can simplify machine unlearning[J]. NeurIPS 2023
>
> [5] Chen et al. Boundary unlearning: Rapid forgetting of deep networks via shifting the decision boundary. CVPR 2023
>
> [6] Wu et al. Munba: Machine unlearning via nash bargaining. CVPR 2025
>
> [7] Shi et al. MCU: Improving Machine Unlearning through Mode Connectivity. 2025.

---

> > ### Author Rebuttal · Reviewer_mJGj · 2026-04-02
> >
> > I appreciate the explanation — things are much clearer after rereading the paper. However, I now have questions about the way the results are being reported:
> > The experiments in Table 3 are based on random data forgetting, which is inherently a privacy scenario. Criterion ➋ however is described as being relevant for bias removal. Could the authors clarify what random data forgetting even means in the context of bias removal? What am I missing here?
> > Table 3 reports results for 12 unlearning methods and discusses them under both Criterion ➊ (privacy/user data removal) and Criterion ➋ (bias removal). Yet these two criteria correspond to fundamentally different use cases. Could the authors clarify how the same set of results in Table 3 can simultaneously and rigorously validate both use cases? Would not each use case require its own separate hyperparameter tuning to ensure a fair and valid evaluation under each criterion?

---

> > > ### Author Response · Authors · 2026-04-03
> > >
> > > We are glad that our explanation was helpful, and we sincerely appreciate the reviewer’s insightful questions. The response to the follow-up questions is as follows.
> > >
> > > > **The relevance of random data forgetting to bias removal**
> > >
> > > As defined in [1] (Section 5.2), "In bias removal ... example scenarios include removing biases, **outdated** or **incorrect information**." We clarify that outdated or incorrect information can naturally manifest as a subset with different classes, which aligns well with the **random data forgetting** scenario. In other words, Criterion ➋ evaluates the strength of information removal, which is independent of whether the removed data corresponds to a semantic class or an arbitrary subset (e.g., outdated or undesired information). Therefore, random forgetting can still meaningfully evaluate the core objective underlying bias removal.  We will highlight this connection more explicitly in the revision.
> > >
> > > > **Different criteria using the same hyperparameter tuning**
> > >
> > > The reviewer's understanding is correct, but we want to clarify as follows.
> > >
> > > Most existing unlearning works focus exclusively on Criterion ➊ (Gap to RT), and do not provide explicit strategy or guidance for tuning toward Criterion ➋. Due to **this practical constraint,** we adopt the default settings in their original papers for our experiments to ensure a **fair and consistent comparison**.
> > >
> > > Nevertheless, we intentionally evaluate the results under both criteria to **highlight the fact that** **Criterion ➊ should not be the sole consideration when designing and evaluating unlearning methods**. Criterion ➋ also reflects real and important application scenarios such as bias mitigation or suppression of undesired information.
> > >
> > > Importantly, **the primary purpose of Table 3 is to validate our proposed metric (CR) across a diverse set of unlearning methods.** By evaluating the same trained models under multiple criteria, we can demonstrate how different evaluation perspectives reveal distinct aspects of forgetting behavior, which would not be visible under a single criterion alone. Importantly, we are not claiming any method should be simultaneously optimal under both criteria.

---

### Decision · Program_Chairs · 2026-04-30

**Decision:**

Accept (regular)

**Comment:**

This paper identifies the issue of fake forgetting in machine unlearning and proposes an uncertainty-aware metric (CR) along with a plug-in framework (CPU) to improve forgetting reliability. Reviewers generally agreed that the problem is important and the approach is novel and practically useful, with solid empirical validation. After the rebuttal, most reviewers (three “Accept” and one “Weak Accept”) were satisfied with the clarifications and supported acceptance. While some concerns remain regarding theoretical assumptions and the coupling between metric and method, I believe these are reasonably addressed and do not outweigh the overall contribution. Therefore, I follow the majority consensus and recommend acceptance.